# A brain-wide map of neural activity during complex behaviour

International Brain Laboratory*[✉], Dora Angelaki[1], Brandon Benson[2], Julius Benson[1], Daniel Birman[3], Niccolò Bonacchi[4], Kcénia Bougrova[5], Sebastian A. Bruijns[6], Matteo Carandini[7], Joana A. Catarino[5], Gaelle A. Chapuis[8], Anne K. Churchland[9], Yang Dan[10], Felicia Davatolhagh[9], Peter Dayan[6], Eric EJ DeWitt[5], Tatiana A. Engel[11], Michele Fabbri[5], Mayo Faulkner[7], Ila Rani Fiete[12], Charles Findling[8], Laura Freitas-Silva[5], Berk Gerçek[8], Kenneth D. Harris[7], Michael Häusser[7,13], Sonja B. Hofer[14], Fei Hu[10], Félix Hubert[8], Julia M. Huntenburg[6], Anup Khanal[9], Christopher S. Krasniak[11], Christopher Langdon[15], Christopher Langfield[16], Petrina Y. P. Lau[7], Zachary F. Mainen[5], Guido T. Meijer[5], Nathaniel J. Miska[14], Thomas D. Mrsic-Flogel[14], Jean-Paul Noel[1], Kai Nylund[3], Alejandro Pan-Vazquez[15], Liam Paninski[16], Alexandre Pouget[8], Cyrille Rossant[7], Noam Roth[3], Rylan Schaeffer[2], Michael Schartner[5], Yanliang Shi[15], Karolina Z. Socha[7], Nicholas A. Steinmetz[3], Karel Svoboda[17], Anne E. Urai[18], Miles J. Wells[7], Steven J. West[14], Matthew R. Whiteway[16], Olivier Winter[5] & Ilana B. Witten[15]

A key challenge in neuroscience is understanding how neurons in hundreds of interconnected brain regions integrate sensory inputs with previous expectations to initiate movements and make decisions[1]. It is difficult to meet this challenge if different laboratories apply different analyses to different recordings in different regions during different behaviours. Here we report a comprehensive set of recordings from 621,733 neurons recorded with 699 Neuropixels probes across 139 mice in 12 laboratories. The data were obtained from mice performing a decision-making task with sensory, motor and cognitive components. The probes covered 279 brain areas in the left forebrain and midbrain and the right hindbrain and cerebellum. We provide an initial appraisal of this brain-wide map and assess how neural activity encodes key task variables. Representations of visual stimuli transiently appeared in classical visual areas after stimulus onset and then spread to ramp-like activity in a collection of midbrain and hindbrain regions that also encoded choices. Neural responses correlated with impending motor action almost everywhere in the brain. Responses to reward delivery and consumption were also widespread. This publicly available dataset represents a resource for understanding how computations distributed across and within brain areas drive behaviour.

It is unclear how hundreds of interconnected brain areas that are processing information related to sensation, decisions, action and behaviour lead to coherent and effective outputs[2–4]. To answer this question, we need to know how the activities of individual neurons and populations of neurons across the brain reflect variables such as stimuli, expectations, choices, actions, rewards and punishments[5]. Electrophysiological recordings from animals have been instrumental in this exploration. Until recently, however, technical limitations have restricted these recordings to a limited number of brain areas, which leaves much of the mammalian brain uncharted or described by fragmentary maps. For example, the mouse brain comprises over 300 identified regions[6], of which only a minority has been systematically recorded in comparable behavioural settings. The regions studied were typically chosen on the basis of a priori hypotheses derived from previous recordings and anatomical connectivity. This approach can identify a localization of function and reveal brain regions that are engaged in computations such as the accumulation of sensory evidence in favour of a decision[7]. Nevertheless, studies have shown that such regions can sometimes be silenced without substantial behavioural consequences[8–12], which suggests that other regions are involved. Overall, it has proven difficult to obtain a comprehensive picture of neural processing based on different reports from different laboratories recording in different brain regions during different behaviours and analysing the data with different methods.

[1]New York University, New York, NY, USA. [2]Stanford University, Stanford, CA, USA. [3]University of Washington, Seattle, WA, USA. [4]William James Center for Research, ISPA–Instituto Universitario, Lisbon, Portugal. [5]Champalimaud Foundation, Lisboa, Portugal. [6]Max Planck Institut, University of Tübingen, Tübingen, Germany. [7]University College London, London, UK. [8]University of Geneva, Geneva, Switzerland. [9]University of California Los Angeles, Los Angeles, CA, USA. [10]University of California Berkeley, Berkeley, CA, USA. [11]Cold Spring Harbor Laboratory, Cold Spring Harbor, NY, USA. [12]Massachusetts Institute of Technology, Cambridge, MA, USA. [13]The University of Hong Kong, Hong Kong, China. [14]Sainsbury Wellcome Centre, University College London, London, UK. [15]Princeton University, Princeton, NJ, USA. [16]Columbia University, New York, NY, USA. [17]Allen Institute for Neural Dynamics, Seattle, WA, USA. [18]Leiden University, Leiden, The Netherlands. *A list of authors and their affiliations appears at the end of the paper. ✉e-mail: info+brainwidemap@internationalbrainlab.org

Nature | Vol 645 | 4 September 2025 | **177**

A broader search for the neuronal correlates of variables such as sensation and decision-making therefore requires the systematic recording of brain regions at a wider scale using a single task with sufficient behavioural complexity. Moreover, the data should be analysed using the same methods. Obtaining such a comprehensive dataset has recently become possible with advances in recording technology. In a species with a small brain such as the mouse, Neuropixels probes[13,14] have enabled larger-scale recordings, such as sampling activity across eight visual areas[15] or across tens of brain regions in mice performing behavioural tasks[1,16,17] or experiencing changes in physiological state[18]. Modern imaging techniques also provide a wider view of activity across dorsal cortical regions[19–21]. Results from these broad surveys suggest that the encoding of task variables varies substantially. Some variables have correlates only in a few brain areas, whereas others are encoded in sparse sets of cells or distributed much more broadly. It is critical to obtain more comprehensive recordings because past recordings may have missed essential regions that are focused on the coding of certain variables and have not fully characterized the nature of distributed coding.

Here we present a publicly available dataset[22] of recordings from 699 Neuropixels probe insertions spaced across an entire hemisphere of the brain in mice performing a behavioural task that requires sensory, cognitive and motor processing[23]. This approach enabled the detection of brain-wide correlates of sensation, choice, action and reward, as well as internal cognitive states, including stimulus expectation or priors (this 'block' prior is described in the companion paper[24]). We also describe initial analyses of these data. Neural correlates of some variables, such as reward and action, were found in many neurons across essentially the whole brain. By contrast, correlates of other variables, such as the input stimulus, could be decoded from a narrower range of regions and significantly influenced the activity of fewer individual neurons. These data, which can be examined online (viz.internationalbrainlab.org) and downloaded from GitHub (https://int-brain-lab.github.io/iblenv/notebooks_external/data_release_brainwidemap.html), are intended to be the starting point for a detailed examination of decision-making processes across the brain and represent a valuable resource to enable the community to perform a broad range of further analyses with single-neuron resolution at a brain-wide scale.

First, we describe the task, the recording strategy and the analysis methods used to provide different views of this rich and complex dataset. Further details of how we ensured reproducible behaviour, electrophysiology and videography are available separately[23,25] and are summarized in the Methods. Then we report the neural correlates of the main events and variables in the task: visual stimulus, choice, feedback and wheel movement.

## Behavioural task

We trained 139 mice (94 male and 45 female) on the International Brain Laboratory (IBL) decision-making task[23] (Fig. 1a,b). On each trial, a visual stimulus appeared to the left or right on a screen, and the mouse had to move it to the centre by turning a wheel with its front paws within 60 s (Fig. 1c). After an initial 90 unbiased trials, the prior probability for the stimulus to appear on the left or right side was constant over a block of trials at a ratio of 20:80% (right block) or 80:20% (left block). Blocks lasted for between 20 and 100 trials, which were drawn from a truncated geometric distribution (empirical mean of 51 trials). Block changes were not cued. Stimulus contrast was uniformly sampled from 5 possible values (100, 25, 12.5, 6.25 and 0%). The 0% contrast trials, when no stimulus was presented, were assigned to a left or right side following the probability distribution of the block. This allowed mice to perform above chance by incorporating this prior in their choices. Following a wheel turn, mice received positive feedback in the form of a water reward, or negative feedback in the form of a white-noise pulse and a 2-s time out. The next trial began

after a delay, followed by a quiescence period during which the mice had to hold the wheel still.

As previously shown[23], mice learnt to both indicate the position of the stimulus and to exploit the block structure of the task. After training, they made correct choices on 81.4 ± 0.4% (mean ± s.d.) of the trials, performing better and faster on trials with high visual stimulus contrast (Fig. 1d). Recorded sessions lasted on average 645 trials (median of 602, range of 401–1,525). Towards the end of the sessions, performance decreased and first wheel-movement times increased (Fig. 1e,f). On 0% contrast trials, in which no visual information was provided, mice gained rewards on 58.7 ± 0.4% (mean ± s.e.m) of trials, significantly better than chance (t-test $t_{138}$ = 20.18, $P$ = 5.2 × 10^{-43}). After a block switch, mice took around 5–10 trials to adjust their behaviour to the new block, as revealed by the fraction of correct choices made on 0% contrast trials after the switch (Fig. 1g). Mice were also influenced by their previous estimate in the presence of visual stimuli. That is, for all contrast values, mice tended to answer left more often on left blocks than right blocks (Fig. 1h).

## Recordings

In these mice, we inserted 699 Neuropixels probes (see an example of one recording of three trials in Fig. 1i), following a grid that covered the left hemisphere of the forebrain and midbrain, which typically represent stimuli or actions on the contralateral side, and the right hemisphere of the cerebellum and hindbrain, which typically represent the ipsilateral side (Fig. 2a). Recordings were collected by 12 laboratories in Europe and the USA, with most recordings using 2 simultaneous probe insertions. To ensure reproducibility, one brain location was targeted in every mouse in every laboratory, as described elsewhere[25]. Only sessions with at least 400 trials were retained for further analyses. Data were uploaded to a central server, preprocessed and shared through a standardized interface[26]. To perform spike sorting on the recordings, we used a version of Kilosort[27] with custom additions[28]. This process produced 621,733 units (including multineuron activity), averaging 889 per probe. To separate individual neurons from clusters of multineuron activity, we then applied stringent quality-control metrics (based on those described in ref. 15), which identified 75,708 well-isolated neurons, averaging 108 per probe.

After recordings, probe tracks were reconstructed using serial-section two-photon microscopy[29], and each recording site and neuron was assigned a region in the Allen Common Coordinate Framework[6] (a table of regions is available online on GitHub (https://github.com/int-brain-lab/paper-brain-wide-map/blob/main/brainwidemap/meta/region_info.csv); statistics are shown in Fig. 2b). Our main analyses were restricted to regions with 20 or more neurons assigned to them in at least 2 sessions, with at least 5 neurons per session. Owing to the grid-based insertion strategy of the probes, more recordings were made in larger regions, which typically led to more neurons being analysed in such regions. Note that it was harder to extract well-isolated neurons from some regions than others; therefore, the yield substantially differed. Although information about molecular cell types can sometimes be gleaned from spike waveforms, we did not attempt to do so for the analyses here. For example, although we performed recordings in some of the main neuromodulatory regions, we do not make specific claims about which neurons release which neurotransmitter.

To illustrate our main results, we plotted them into a flatmap of the brain[30] (Fig. 2c). The Extended Data figures present some of the results on more conventional two-dimensional (2D) sections (which are detailed in Extended Data Fig. 1). For reference, the average activity across all regions aligned to the major task events—stimulus onset, first wheel-movement time and feedback—is shown in Supplementary Fig. 1. To visualize continuous temporal dynamics across different task epochs, that figure also shows time-warped average activity, which was simultaneously aligned to stimulus, movement and feedback onsets.

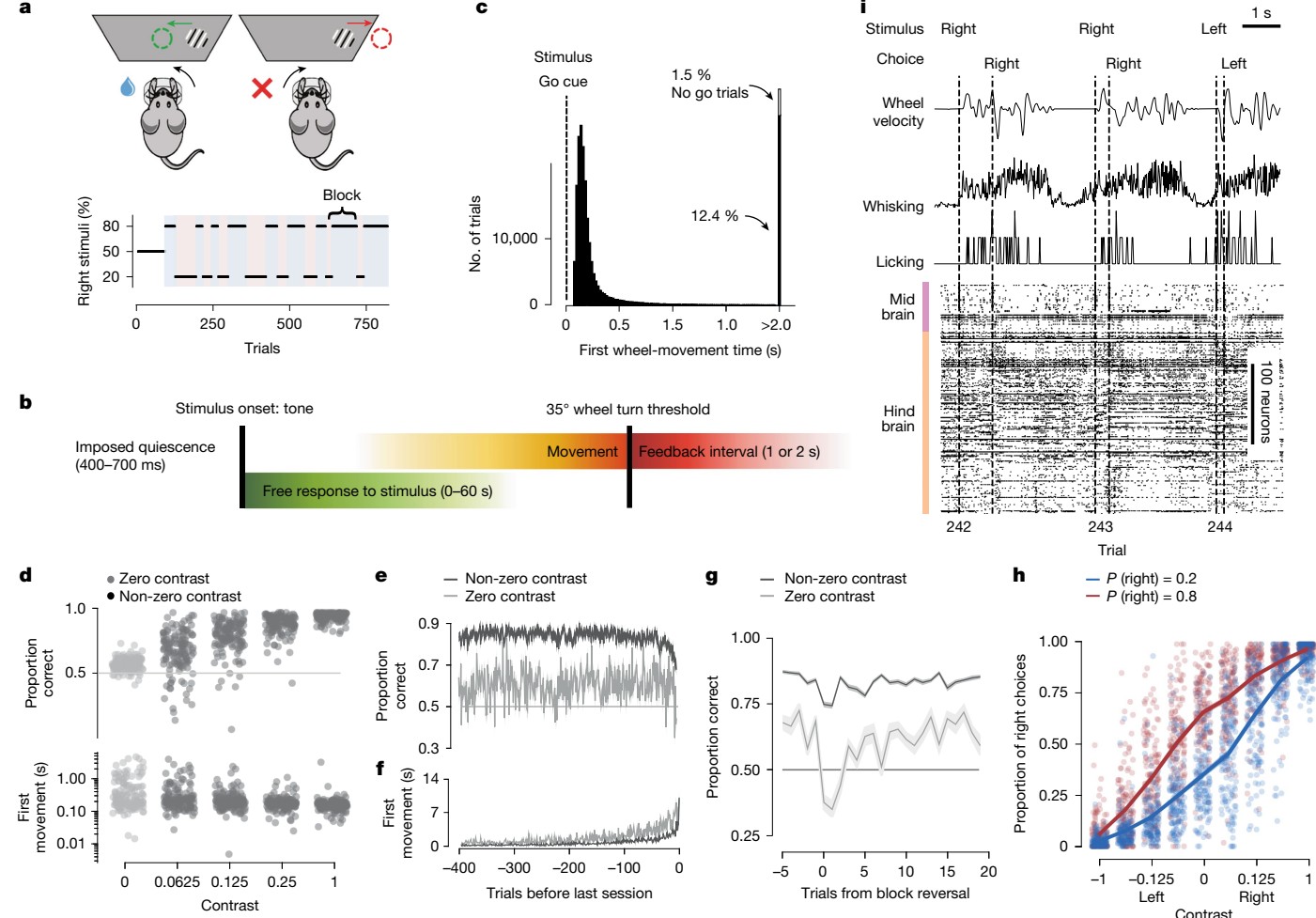

**Fig. 1 | The IBL task, data types and behaviour. a**, Schematic of the IBL task and the block structure of an example session. **b**, Timeline of the events and analysed variables. Colours are used in other figures. **c**, Distribution of the times between stimulus onset and first wheel-movement time (interpreted as a reaction time) from 459 sessions. The distribution is truncated at 80 ms and 2 s. A total of 22.8% first wheel-movement times occurred under 80 ms (not shown). **d**, Proportions of correct choices (top) and first wheel-movement time (bottom) given a stimulus contrast (one point per mouse per contrast). Performance on 0% contrast trials (grey) was 58.7 ± 0.4% correct (mean ± s.e.m. across mice). Data are for 139 mice and 454 sessions. **e**, Proportions of correct choices given the number of trials before the end of the session for 0% (grey) and non-0% (black) contrast trials (mean ± s.e.m. across mice). **f**, The same

analysis for first wheel-movement times. **g**, Reversal curves. Proportions of correct choices around a block change for trials with 0% contrast and >0% contrast (mean ± s.e.m. across mice; excluding the first 90 unbiased trials). **h**, Psychometric curves. Fraction of correct choices given a signed contrast (positive or negative for right or left stimuli, respectively) for all mice (one dot per contrast per mouse). Right choices were more or less common in right (red) or left (blue) blocks, with $P_{right} = 0.8$ and $P_{right} = 0.2$, respectively. **i**, Time series and trial information for three example trials: rotary encoder output of the wheel, video analysis and spike-time rasters across multiple brain regions. Figures are organized according to the IBL style (https://github.com/int-brain-lab/ibl-style/tree/main). Schematic in **a** was adapted from ref. 23, eLife, under a Creative Commons licence, CC BY 4.0.

The processed data for each trial consisted of a set of spike trains from multiple brain regions together with continuous behavioural traces and discrete behavioural events (Fig. 1i). These were recorded using a variety of sensors, including three video cameras and a rotary encoder on the wheel. The data were processed using custom scripts and DeepLabCut[31] to generate the times of major events in each trial along with wheel velocity, whisker motion energy, lick timing and the positions of body parts. We only analysed trials in which the first wheel-movement time (which is our operational definition of a reaction time) was between 80 ms and 2 s (Fig. 1c).

Instructions for accessing the data[22], together with an online browser, are available at https://data.internationalbrainlab.org.

## Neural analyses

To obtain an initial appraisal of the brain-wide map, we performed single-cell and population analyses to assess how neural activity

encodes task variables and how it can be analysed to decode these variables. We considered four key task variables (Fig. 3a): visual stimulus; choice (left or right turning of the wheel); feedback (reward or time out); and wheel movement (speed and velocity). The main figures show the results of these analyses in a canonical dataset of 201 regions for which we had at least 2 sessions with 5 well-isolated neurons each and at least 20 well-isolated neurons after pooling all sessions (for a total of 62,857 neurons; Supplementary Table 1). Supplementary information shows results for a wider range of neurons and regions appropriate for each analysis.

To provide complementary views on how these task variables are represented in each brain region, we used four analysis techniques (Fig. 3b–e; see Supplementary Fig. 2 for a fuller picture). The details of each technique are provided in the Methods, along with a discussion of the corresponding null distributions, permutation tests and false discovery rate (FDR$_q$ at the level $q$ using the Benjamini–Hochberg procedure) that we used to limit statistical artefacts.

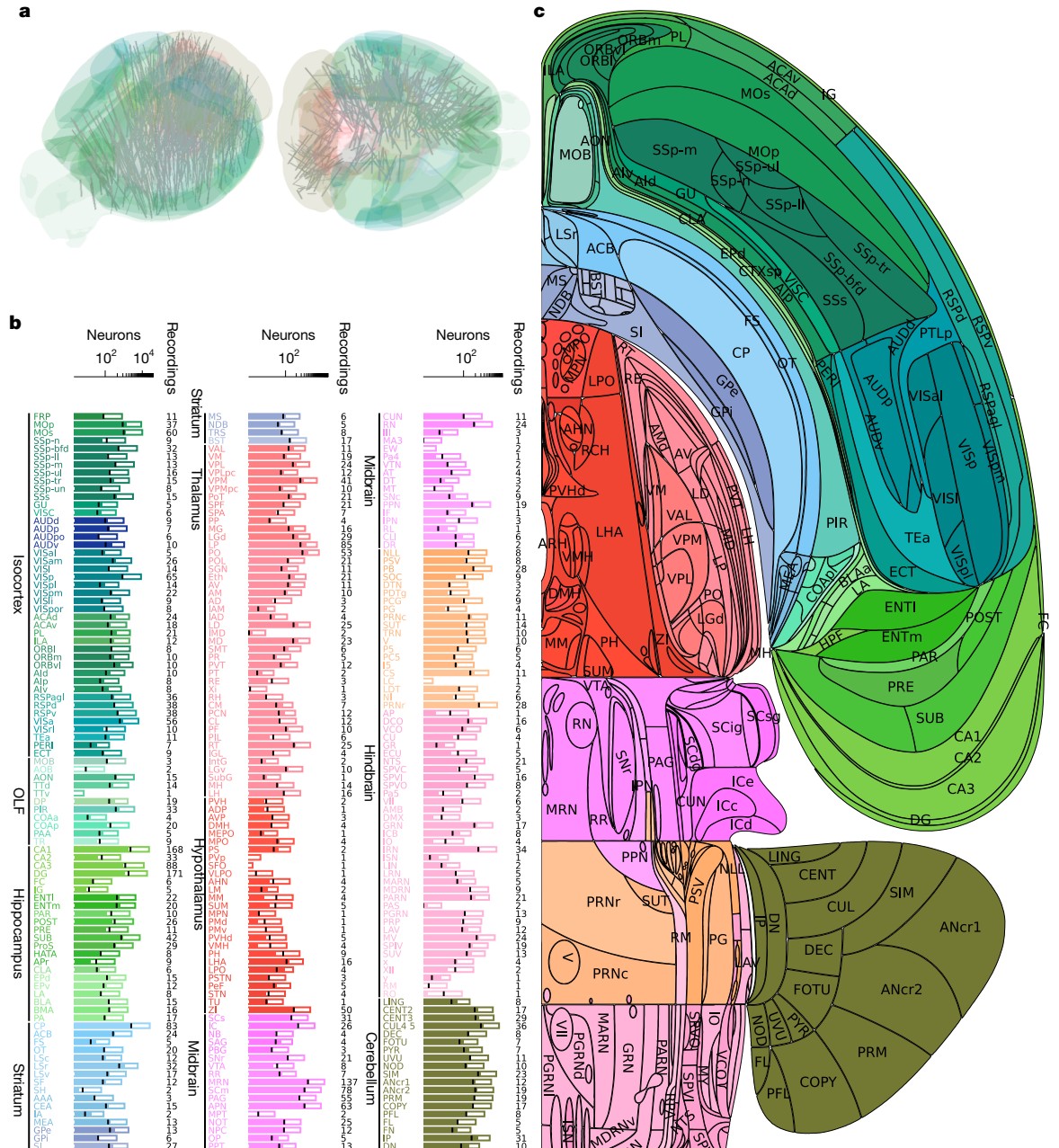

**Fig. 2 | Brain-wide recordings during behaviour. a**, Neuropixels probe trajectories shown in a 3D brain schematic. A total of 699 insertions were performed across 139 mice. **b**, For each region, the number of neurons recorded (full bar length) and the number of well-isolated neurons used for analysis (filled portion; for reference, the black line on each bar shows 10% of the number of recorded neurons, which is the average number of neurons that were well-isolated) are shown. Additional information and definitions for brain regions are provided in a table on GitHub (https://github.com/int-brain-lab/

paper-brain-wide-map/blob/main/brainwidemap/meta/region_info.csv). The same table reports the so-called Cosmos hierarchical grouping of the regions, which distinguishes the isocortex, the olfactory areas (OLF), the hippocampal formation (HPF), the cortical subplate (CTXsp), the cerebral nuclei (CNU), the thalamus (TH), the (hypothalamus (HY), the midbrain (MB), the hindbrain (HB) and the cerebellum (CB). The coloured text labels of brain regions are used in other figures. **c**, Flatmap of one hemisphere showing the acronyms used for the regions.

First, we used a decoding model to predict the value of the task variable on each trial from the neural population activity using regularized logistic or linear regression (Fig. 3b and Supplementary Fig. 2b). This analysis can detect situations in which a variable is robustly encoded but only in a sparse subset of neurons. We assessed decoding for each variable separately without considering the correlations between the task variables. This quantifies what downstream neurons would be able to determine from the activity, but does not disentangle factors that are related such as the stimulus side and choice. We performed decoding in each region and then corrected the $R^2$ of the fit to a variable by the

$R^2$ of the fit to a suitable null distribution. We then used Fisher's combined probability test[32,33] to combine decoding results across sessions. To correct for the comparisons over multiple regions, we chose a level of 0.01 for the FDR (FDR$_{0.01}$).

Second, we computed single-cell statistics. For this, we tested whether the firing rates of single neurons correlated with three task variables (visual stimulus side, choice side and feedback) in the appropriate epochs of each trial (Fig. 3c and Supplementary Fig. 2c). As the task variables of interest are themselves correlated, we used a condition combined Mann–Whitney $U$-test[1] for analysis, which compares

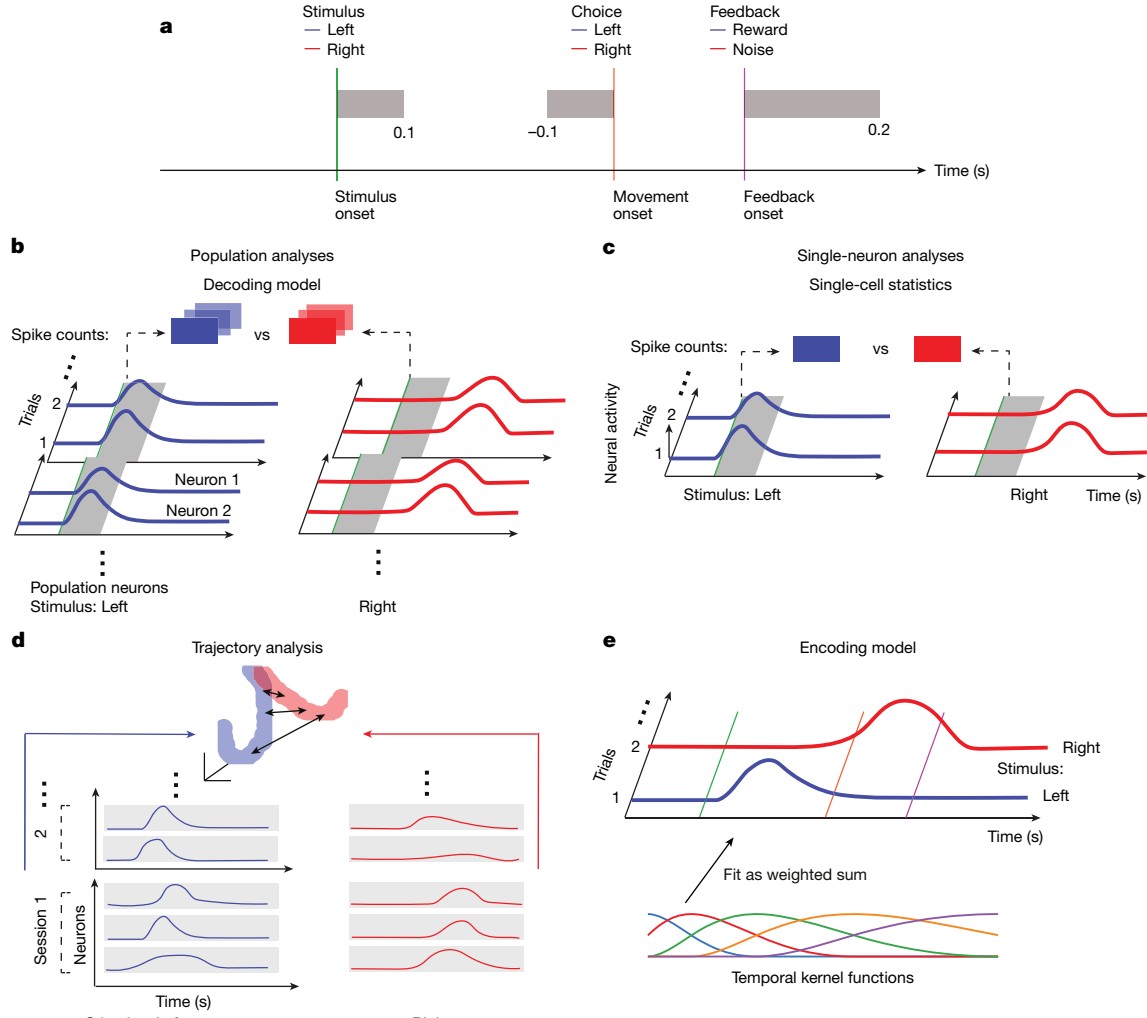

**Fig. 3 | Illustration of neural analyses.** See also Supplementary Fig. 2. **a**, Schematic of the task structure, with the time windows used for analysis in grey. In **b**–**d**, left (blue) and right (red) stimuli are used as example task variables, and the neural traces are coloured accordingly. **b**, Schematic of the decoding model, which quantifies neural population correlates with task variables. Regularized logistic or linear regression is used to map spike counts in the relevant time windows (grey zone) from cells in each area into predictions of the values of the variables. **c**, Schematic of the single-cell analysis, which quantifies single-cell neural correlates with task variables. A conditioned combined Mann-Whitney *U* statistic is used to compute how sensitive the activities (in the grey zone) of single cells are to individual task variables, controlling for the values of other variables. **d**, Schematic of the population trajectory analysis, which describes the time evolution of the across-session neural population response, pooling cells across all recordings per region. The mean activities of every cell across their entire session for the different values of task variables are segmented into short bins, and used to define trajectories in a high-dimensional state space (projected, purely for visualization, into 3D). The distances between the trajectories for the different values of the task variables (arrows) define the separation. **e**, Schematic of the encoding model, which uses multiple linear regression of task-defined and behaviourally defined temporal kernels (the multicoloured traces) to fit the activity of single neurons.

spike counts between trials differing in just one discrete task variable with all others held constant. Using a permutation test, we determined the fraction of individual neurons in a region that were significantly selective to a variable, using a threshold specific to each variable. For each session with recordings in a specific region, we computed the significance score of the proportion of significant neurons by using the binomial distribution to estimate false-positive events. We then combined significance scores across sessions with Fisher's combined probability test[32,33] to obtain a combined *P* value for each region. We report a region as being responsive to a variable if this combined *P* value was below the chance level, correcting for multiple comparisons using $FDR_{0.01}$. This method has lower statistical power than decoding as it only examines noisy single neurons; therefore, it may miss areas that have weak but distributed correlates of a variable. However, it controls for correlated variables in a way that the decoding method does not. Therefore, it is able to exclude neurons that appear to represent a variable by virtue of the correlation of that variable with a confound.

Third, we performed a population trajectory analysis (Fig. 3d and Supplementary Fig. 2d). To that end, we averaged firing rates of single neurons in a session across all trials in 20-ms bins and then aggregated all neurons across sessions and mice per brain region. We examined how trajectories in the high-dimensional neural spaces reflected task variables. We did this by measuring the time-varying Euclidean distance between trajectories, $d(t)$, in the interval of interest, normalized by the square root of the number of recorded cells in the given region to obtain a distance in units of spikes per second. From this time-varying distance, we extracted differential response amplitude and latency statistics. For significance testing, we permuted trials for the variable of interest while keeping the other variables fixed to minimize the effect of correlations among the variables (as done for single-cell statistics), using $FDR_{0.01}$ to control for multiple comparisons. For visualization, we projected the

trajectories into a three-dimensional principal components space. This analysis combined all recordings into one supersession before computing the effect of a variable for each brain region, weighting each cell equally, thus combining recordings at the cell level for each region. This approach can produce a strong signal-to-noise ratio, but it cannot distinguish results obtained in individual sessions. It further stands out by providing the temporal evolution of the sensitivity of a region to a variable during the interval of interest.

Finally, we used an encoding model[34] to fit the activity of each cell on each trial as a linear combination of a set of temporal kernels locked to each task event (Fig. 3e and Supplementary Fig. 2e). This generalized linear model quantifies the dependency at a temporally fine scale at the cost of a potentially low signal-to-noise ratio. We measured the impact of a variable by removing its temporal kernels and quantifying the reduction in the fit of the activity of a neuron (typically assessed using $\Delta R^2$). This method lacks a convenient null distribution; therefore, we report effect sizes rather than significance.

The results from the different analysis methods are not expected to agree perfectly, because they focus on different aspects of the responsivity of individual neurons and populations thereof. Moreover, for the population trajectory analysis, information across multiple sessions rather than within single sessions was combined. However, this strategy allowed us to test the robustness of our results by comparing findings based on subsets of the data (Supplementary Fig. 3). The results from the methods therefore should be collectively interpreted. For a direct comparison of analysis scores, see flatmaps in Extended Data Fig. 2 and scatter plots of scores for analysis pairs in Extended Data Fig. 3.

On the basis of the four main analyses, we also performed a basic, inter-area analysis using Granger interactions for simultaneously recorded brain areas (Extended Data Fig. 4). High Granger interaction scores were obtained for region pairs from all major brain regions, which were weakly correlated with anatomical distance and mostly bidirectional. A similar analysis was performed for the prior over the block in a companion paper[24].

Below, we describe the results of our four analyses applied to the coding of each of the four task variables.

## Representation of visual stimulus

We first considered neural activity related to the visual stimulus. Classical brain regions in which visual responses are expected include the superior colliculus[35,36], the visual thalamus[37–39] and visual cortical areas[40–43], with latencies reflecting successive stages in the visual pathway[15,44]. Correlates of visual stimuli have also been observed in other regions implicated in visual performance, such as parietal[45] and frontal[46–49] cortical areas and the striatum[50,51]. Substantial encoding of visual stimuli may also be present beyond these classical pathways, as the retina sends outputs to a large number of brain regions[52]. Indeed, an initial survey[1] of regions involved in a similar task uncovered visual responses in areas such as the MRN (information and definitions for brain regions are provided in a table on GitHub: https://github.com/int-brain-lab/paper-brain-wide-map/blob/main/brainwidemap/meta/region_info.csv). Thus, we proposed that visual coding would extend to diverse regions beyond those classically described.

Consistent with this hypothesis, a decoding analysis based on the first 100 ms after stimulus onset revealed correlates of the visual stimulus side in many cortical and subcortical regions. Strong signatures were observed in the visual cortex (VISam, VISl, VISa and VISp), the prefrontal cortex (MOs), the thalamus (LGd, LGv and CL), the midbrain (NOT, SNr, MRN, SCm and APN) and the hindbrain (GRN and PGRN) (Fig. 4a,f). For example, the activity of neurons in the primary visual cortex (VISp) could be used to predict the stimulus side (Fig. 4i). Note that among the analyses we performed, the decoding analysis is distinct because it does not control for variables correlated with

the visual stimulus, such as choice and block. Therefore, some of the regions with significant results from this analysis might instead encode these variables.

Decoding performance varied across sessions; therefore, in Extended Data Fig. 5, we show performance across sessions for all regions, even those that are not significant after the $FDR_{0.01}$ correction for multiple comparisons. Supplementary Figure 4 presents data for a subset of these results split by sex. These results did not reveal differences between male and female mice. Given that some regions may represent visual information in localized sites that were only occasionally covered by our probes, we also report the fraction of sessions in which we were able to decode the stimulus from a region significantly to assess the spatial distribution (Extended Data Fig. 6a).

To distinguish the possible contributions of variables correlated with the visual stimulus, we next analysed responses in the same 100-ms window using single-cell analysis, which controlled for other variables by holding them constant in each comparison of stimulus side. This analysis produced a consistent picture but revealed fewer significant areas, with 0.5% of all neurons correlated with a stimulus side (Fig. 4b). Significant regions included visual cortical areas (VISp, VISpm, VISam and VISl) and the visual thalamus (LGd, LGv and LP), but also other structures such as the auditory cortex (AUDv), the dorsal thalamus (PF), parts of the midbrain (SCm, APN and NOT) and the hindbrain (CS, PRNr, GRN, IP and ANcr1). However, even in the regions that contained the largest fractions of responsive neurons (such as the visual cortex), this fraction did not exceed about 10%. Given our grid-based approach to probe insertion, this low percentage of neurons could be the result of neurons having receptive fields (RFs) that did not overlap with the stimulus position.

To provide an overview of the variability across sessions, Extended Data Fig. 7 presents the fraction of significant neurons broken down by sessions without applying the $FDR_{0.01}$ correction.

The results of the population trajectory analysis were consistent with the decoding analysis (Fig. 4c) and provided further information about the time course during which visual signals were encoded (Fig. 4d). For example, the responses in the visual area VISp to right visual stimuli compared with left visual stimuli showed early divergence shortly after stimulus onset, followed by rapid convergence (Fig. 4j,k). The shuffled control trajectories (shown in grey) are close to the true trajectories (Fig. 4j) because this analysis controls for choice, which is tightly correlated with the stimulus. Altogether, this analysis indicated that distance was a significant result in 104 regions (Fig. 4c,f).

The evolution of trajectories over time could be distilled into two numbers (Fig. 4l,m): the maximal response (maximum distance or $d_{max}$) and the response latency (first time to reach 70% of $d_{max}$; mapped across the brain in Fig. 4d). This characterization of the dynamics of visual representations revealed that some areas had short latencies and early peaks, including classical areas (LGd, LP, VISp, VISam and VIpm). A spatiotemporally finely resolved view of latency differences was obtained, such that LGd < VISp ≈ LP < VISpm < VISam (latencies of around 34, 42, 42, 57 and 78 ms, respectively; Fig. 4d,l,m). This early wave of activity was followed substantially later by significant visual encoding in other areas, including the MRN, SCm, PRNr, IRN and GRN (latencies of about 100–120 ms; Fig. 4d,l,m).

The encoding analysis characterized visual encoding in individual neurons across the brain (Fig. 4e). We asked whether a prediction of single-trial activity can be improved by adding a temporally structured kernel that unfolds over 400 ms after stimulus onset, on top of activity related to feedback, wheel movement speed and block identity. As there is no convenient null distribution that could be used to test the significance of this improvement, we only report effect sizes. For instance, as expected, an example VISp neuron showed large differences between stimuli on the left and right (Fig. 4g) such that removing the visual kernel resulted in a poor fit of the firing rate of the neuron (Fig. 4h). This analysis indicates that the visual stimulus variable improved fits

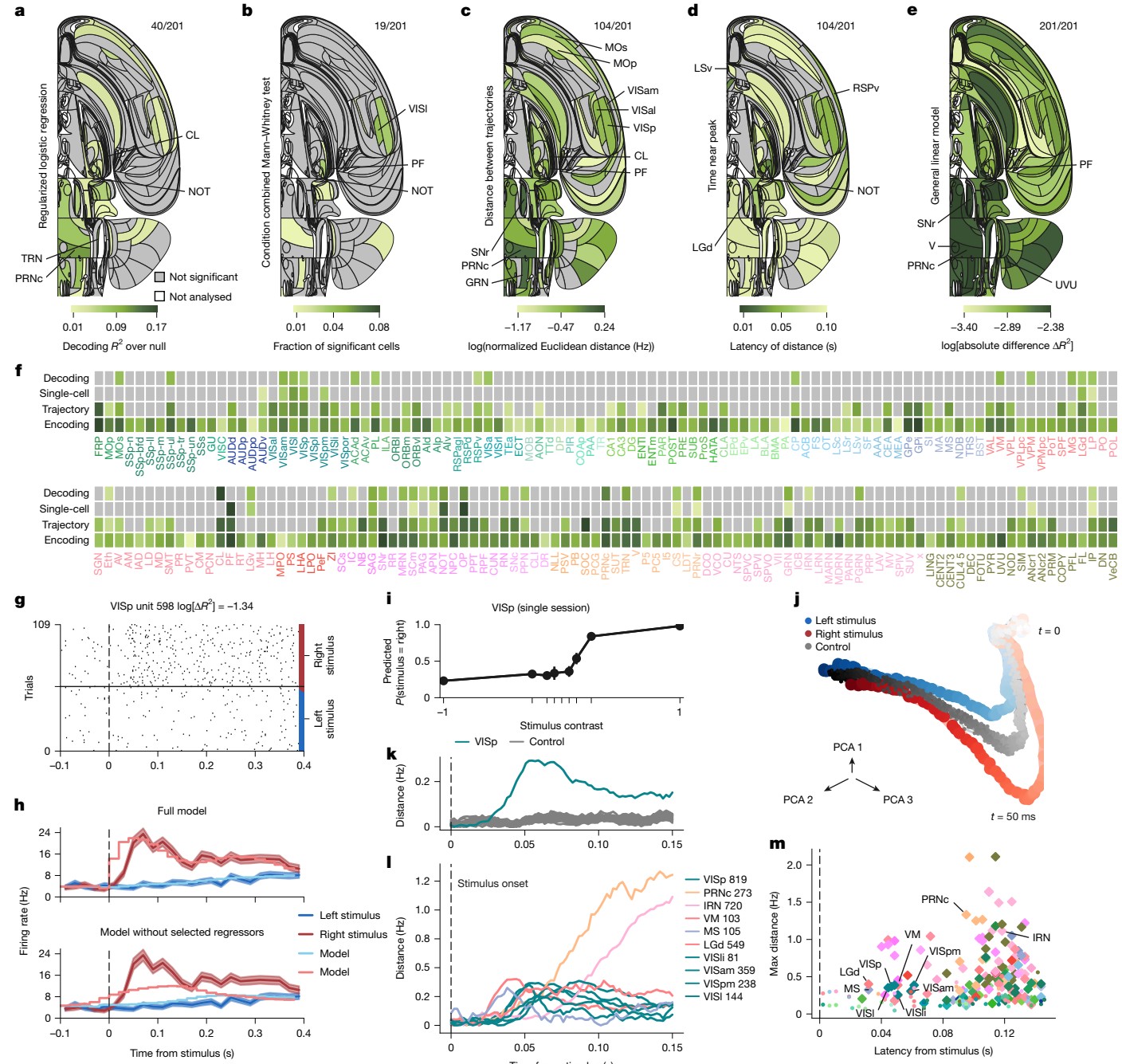

**Fig. 4 | Representation of the visual stimulus.** See also Extended Data Figs. 5–7 and data in the IBL Brain Atlas (https://atlas.internationalbrainlab.org/?alias= bwm_stimulus). For **a**–**e**, colour indicates the effect size; grey not significant at FDR$_{0.01}$; ratios on the upper right indicate significant/total regions; white, regions not analysed. The analysed stimulus interval is 0 ms to 100 ms following stimulus onset. **a**, Decoding. Null-corrected median balanced accuracy. **b**, Single-cell statistics. Fraction of neurons significantly modulated by stimulus side. Mann–Whitney and condition combined Mann–Whitney tests at $P < 0.001$ and $P < 0.05$, respectively. Significance was based on the binomial distribution of false-positive events and FDR$_{0.01}$. **c**, Population trajectory distance. Time-resolved maximum Euclidean distance (in spikes per s, for dimension = number of cells per region, log$_{10}$) between trajectories following left versus right stimuli. Significance was relative to a shuffle control and FDR$_{0.01}$. **d**, Population trajectory latency. First time crossing 70% of the $d_{max}$ for significant regions. **e**, Encoding. Mean absolute difference $|\Delta R^2|$ in improvements over 400 ms after stimulus-onset from causal right or left stimulus kernels for all neurons. Extended Data Figure 9 shows the median split by first wheel-movement time. **f**, Effect significance (grey, not significant; **a**–**c**) and size (by darkness; **a**–**c**,**e**)

by region. **g**, Spike raster for an example VISp neuron (Supplementary Table 3). Trials per condition are shown in temporal order, with every third trial shown. **h**, Top, peri-event time histogram (PETH; shading indicates ±1 s.e.m.) and full encoding model prediction for left or right stimuli aligned to stimulus onset for the neuron in **g**. Bottom, same PETHs, but for predictions from a model that omitted stimulus kernels. **i**, Decoded probability (with 95% confidence intervals across 474 trials) of a stimulus side given contrast from 40 VISp neurons (Supplementary Table 3). **j**, Trial-averaged population trajectories from left and right stimulus trial-averaged activity across the VISp (all sessions), in three principal component analysis (PCA) dimensions. Dots, single time bins; darker colours are later times. Grey, pseudo-trajectories (control) from randomly selected trials matched for block and choice but not stimuli. **k**, Trajectory distance for VISp neurons. Grey, pseudo-trajectory distances. **l**, Trajectory distances across regions (with neuron numbers indicated). Early responses are observed across visual areas, with ramping modulation in others. **m**, Maximal population trajectory distance and modulation latency (diamonds, significant regions; dots, not significant regions). Extended Data Figure 10a,d,g shows a longer time window and more neurons.

of encoding models for neurons across a wide range of brain regions (Fig. 4e,f).

These results were broadly consistent with RF measurements. At the end of the decision-making task, we performed RF mapping in most recording sessions (504 out of 699 sessions). We thereby computed the visual RFs of neurons across the brain (204 regions covered), including classical visual areas and beyond. We estimated the significance of the RF of each neuron by fitting the RF to a 2D Gaussian function and comparing the variance explained to the fitting of 200 random shuffles of each RF. Overall, we found a relatively small fraction of cells with significant RFs (Supplementary Fig. 5). The regions with large fractions tended to be classical visual regions (VISp, VISl and SCs). We also observed non-zero fractions in diverse areas beyond classical visual regions, including the auditory cortex (AUDv), the auditory thalamus (MG), parts of the midbrain (MRN, SCm, APN and NOT) and the hindbrain (ANcr1) (Supplementary Fig. 5 and Extended Data Fig. 8). These results provide further support for the findings from the neural analysis of coding of visual stimuli during the task.

To determine whether trials with rapid responses were associated with distinct patterns of activity, we separated effect sizes according to a median split of the first wheel-movement time (Extended Data Fig. 9). Regions with high explained variance from stimulus onset on all trials also mostly appeared in the early first wheel-movement time model (for example, RSPv, VISl, PAG and RN). By contrast, a handful of new cortical regions in motor areas (namely MOs, and ORBm and ILA to a lesser extent) seemed to be explained only when fitting early-response trials. Late-response trials showed fewer regions with well-explained (>0.03 change) variance, but in those trials, significant variance in the subiculum and post-subiculum was explained, which was not true when considering all trials.

Taken together, the decoding, single-cell statistic and population trajectory analyses reveal a largely consistent picture of visual responsiveness. That is, it includes large and short-latency responses in classical areas but also extends to other diverse regions, even when controlling for correlated variables, particularly at later times relative to the stimulus onset.

## Representation of choice

Next, we examined which regions of the brain represented the mouse's choice and in which order. Choice-related activity has been observed in parietal, frontal and premotor regions of the primate cortex[7,53,54] where many neurons show ramping activity consistent with evidence accumulation[7,55]. These choice signals develop across frontoparietal regions and appear later in frontal eye fields[56]. Similar responses were found in rodent parietal[57], frontal[58,59] and premotor[60,61] cortical regions. However, in both rodents and primates, choice-related activity has also been found in the hippocampal formation[62] and subcortical areas, in particular in the striatum[1,63], the superior colliculus[1,64,65] and other midbrain structures[1]. Subcortical regions show choice signals with similar timings as the cortex[1,17] and have a causal role in making choices[65]. This evidence indicates that decision formation engages a distributed network of cortical and subcortical brain regions. Our recordings enabled us to determine in detail where and when choice-related activity emerges across the brain.

The decoding analysis suggested a representation of choice (left versus right upcoming action) in a larger number of brain regions than the representation of the visual stimulus (Fig. 5a,f). The animal's choice could be decoded from neural population activity during a 100-ms time window before the first wheel-movement time in many analysed regions. The strongest effect sizes were observed in the hindbrain (GRN, VII, PRNr and MARN), the thalamus (CL), the midbrain (SNr, RPF and MRN), the hypothalamus (LPO) and the cerebellum (CENT2). For example, the activity of neurons in the GRN of the medulla could be readily decoded to predict choice in an example session (Fig. 5g,i).

Choice was also significantly decodable from somatosensory (SSp-ul), prelimbic (PL), motor (MOp, MOs), orbital (ORBvl) and visual (VISp) cortical areas.

Some of the decodable choice information, however, could be due to responses to the visual stimulus or block, which correlate with choice. We therefore performed single-cell analyses that controlled for correlations between all these task variables. More single neurons significantly responded to choice than to the visual stimulus (Fig. 5b,f). That is, firing rates of 4% of all neurons recorded across all brain regions correlated with choice direction during the 100 ms before the first wheel-movement time when controlling for correlations with the stimulus and block. The largest fractions of neurons that significantly responded to choice were in the hindbrain, cerebellar, midbrain and thalamic regions, consistent with the results of the decoding analysis. Neurons with significant responses to choice were highly prevalent in the thalamus (CL and SPF), the midbrain (SCm, MRN, SNr, RPF and NPC), the pons and the medulla and cerebellar nuclei (GRN, IRN, SOC, VII, TRN and FOTU), most of which did not show visual responses. The prevalence of choice-selective neurons in these subcortical regions was further confirmed by the single-cell encoding model (Fig. 5e). For instance, an example neuron in GRN showed stronger responses for right choices than left choices (Fig. 5g). The encoding model captured this preference but only in the presence of the kernel associated with choice (Fig. 5h), thereby indicating choice selectivity.

The population trajectory analysis enabled us to compare the magnitude of choice representation across brain regions on the population level. This parameter was measured as the distance between trajectories in the neural population state space on left versus right choice trials (Fig. 5c,f). The population-level choice representation was evident in many regions across the brain, with the strongest separation of neural trajectories in the hindbrain (IRN, GRN and PRNc) and the midbrain (APN, MRN and SCm). A similar magnitude of population-level choice encoding was observed in many other areas (Fig. 5c,f). In our example region, the GRN, the trajectories for left and right choice trials separated significantly more than in the shuffled control (Fig. 5j,k; controlling for correlations with stimulus and block), and the magnitude of this separation was greatest across all brain regions (Fig. 5l). Thus, all our analyses consistently point to a distributed choice representation, with some of the strongest choice signals in the midbrain, the hindbrain and the cerebellum, and relatively weaker encoding of choice across many cortical areas.

Next, we analysed when the choice signals emerged across the brain by measuring the latency at which neural population trajectories separated on the left versus right choice trials during the time preceding the first wheel-movement time (Methods). Some of the earliest choice signals developed nearly simultaneously in the thalamus (LD) and the cortex (VISl, VISam and ECT), and later appeared in a larger distributed set of brain areas (Fig. 5d,l,m). The regions GRN and MRN in the reticular formation showed moderate choice latencies and some of the strongest magnitude of population-level choice representations (Fig. 5j,l,m). This result suggests that these brainstem structures have a role in decision formation or movement preparation.

For the encoding model, we also separated out the differences in variance explained according to a median split of the first wheel-movement time (Extended Data Fig. 9). Regions that showed a high degree of variance explained by rightward movement onset in the RSPv again appeared when fitting all trials and early-response trials, but not late-response trials. In the case of movement onset, however, the secondary visual areas VISam and VISal were consistently involved in all trials along with motor areas. Notably, the high variance explained in the subiculum extended to the hippocampal CA1 and the post-subiculum only in late-response trials, and did not appear at all in early-response trials. Subcortical involvement seemed to be limited to early-response trials in some regions like the PAG, which did not appear in the model fit to all trials.

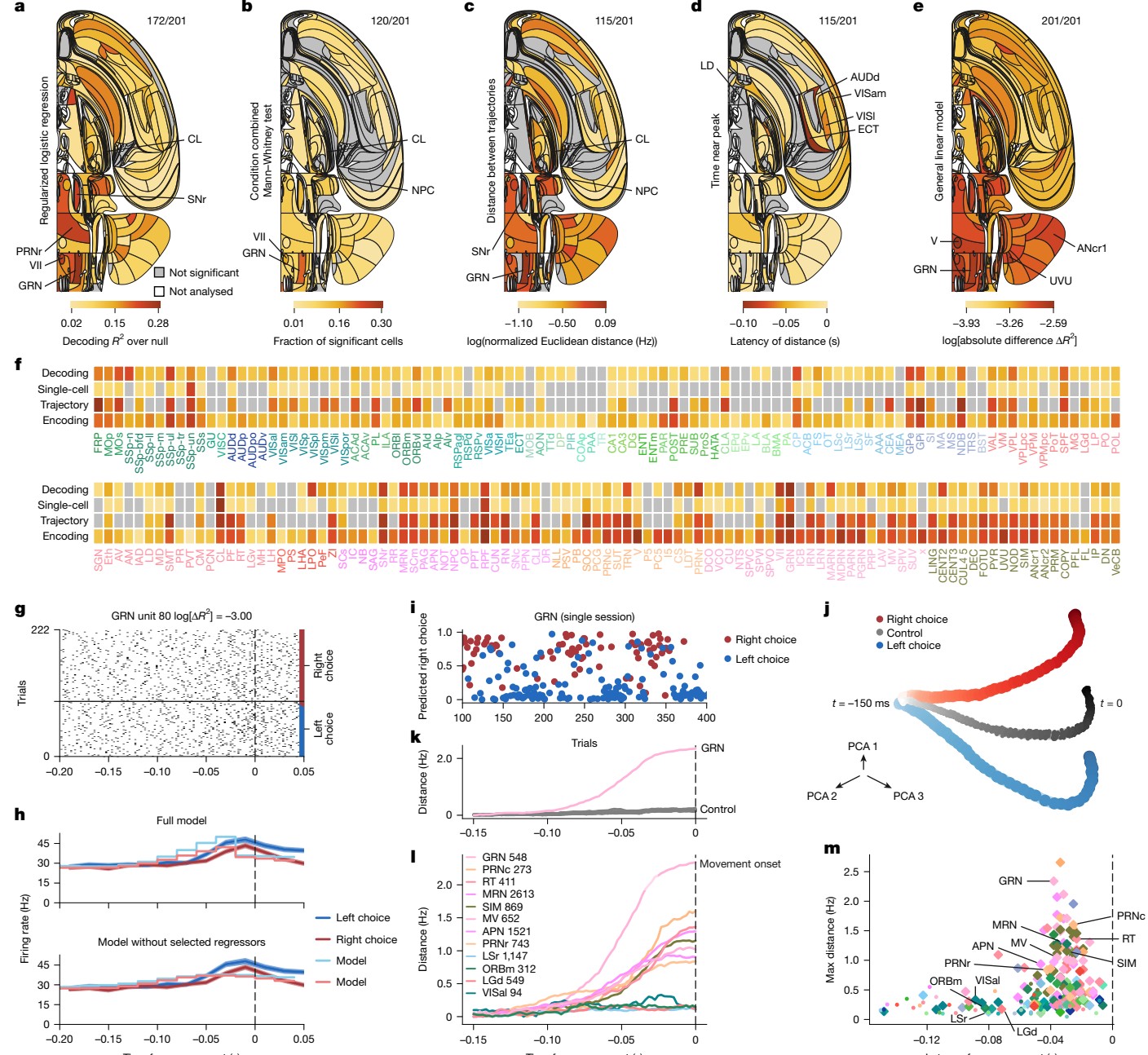

**Fig. 5 | Representation of choice.** See also Extended Data Figs. 5, 7 and 11 and data in the IBL Brain Atlas (https://atlas.internationalbrainlab.org/?alias=bwm_choice). Figure parts and statistics are as for Fig. 4. The analysed choice interval is −100 ms to 0 ms relative to the first wheel-movement time. **a**, Decoding. Null-corrected median balanced accuracy. **b**, Single-cell statistics. Fraction of neurons significantly modulated by choice side in −100 to 0 ms relative to the first wheel-movement time. Mann–Whitney and condition combined Mann–Whitney tests at $P < 0.001$ and $P < 0.05$, respectively. Significance was based on the binomial distribution of false-positive events and $FDR_{0.01}$. **c**, Population trajectory distance. Time-resolved maximum Euclidean distance ($d_{max}$ in spikes per s, for dimension = number of cells per region, $\log_{10}$) between trajectories for left versus right choices. Significance is relative to a shuffle control and $FDR_{0.01}$. **d**, Population trajectory latency. First time crossing 70% of the $d_{max}$ for significant regions relative to movement onset. **e**, Encoding. Mean absolute difference $|\Delta R^2|$ in improvements from 200-ms anticausal kernels aligned to the right and left first-movement times. See Extended Data Fig. 9 for separation by the median first wheel-movement time. **f**, Effect significance (grey, not significant; **a**–**c**) and size (by darkness; **a**–**c**,**e**) by region. **g**, Spike raster of an example GRN neuron (Supplementary Table 3). Trials per condition are shown in temporal order, with every third trial shown. **h**, Top, PETHs (shading indicates ±1 s.e.m.) aligned to the first wheel-movement time on left (blue) and right (red) choice trials and full encoding model prediction for an example neuron in **g**. Bottom, the same PETHs but with predictions from a model that omitted left and right first-movement regressors. **i**, Decoded choice probability from 68 neurons in the GRN (Supplementary Table 3). **j**, Trial-averaged population trajectories in GRN neurons from left and right choice trials in three PCA dimensions. Dots, single time bins; darker colours indicate times closer to the first wheel-movement time. Grey (control), pseudo-trajectories from trials with a randomized choice, controlling for correlations with stimulus and block. **k**, Trajectory distance between left and right choice for GRN neurons, showing ramping activity. Grey, pseudo-trajectory distances. **l**, Trajectory distances across regions (with neuron numbers indicated) showing ramping choice-modulation with time. **m**, Maximal population trajectory distance and modulation latency (diamonds, significant regions; dots, not significant regions). Extended Data Figure 10b,e,h shows a longer time window and more neurons.

## Representation of feedback

At the end of each trial, the mouse received feedback for correct or incorrect responses: a liquid reward at the lick port or a noise-burst stimulus with a time-out period. These positive or negative reinforcers influence the learning of the task[66–70]. The liquid is consumed through licking, an activity that probably involves prominent neural representations that, in this study, we were not able to distinguish from the more abstract representation of reward. Feedback also activates neuromodulatory systems such as dopamine[71,72], which have widespread connections throughout cortical and subcortical regions. Nevertheless, it is unclear whether direct encoding of feedback signals in the brain is widespread.

The decoding analysis revealed nearly ubiquitous neural responses associated with reward delivery on correct versus incorrect trials, and probably the motor responses associated with its consumption (Fig. 6a,f). Using the neural responses in the 200 ms after feedback onset, we were able to decode whether the trial was correct from nearly all recorded brain regions (Fig. 6a,f). In many regions, decoding was almost perfect. An example is the activity of the IRN in a selected session shown in Fig. 6i.

Our single-cell statistics applied to the same trial interval confirmed the decoding results. Neurons with significant response changes to correct versus incorrect feedback or reward consumption were widespread (Fig. 6b,f), with only a small handful of regions not significant for feedback type. The same was true for feedback versus the inter-trial interval baseline (Supplementary Fig. 6).

Population trajectory analysis also showed significant response differences for correct versus incorrect responses across every recorded brain region, which was predominantly consistent with the other analyses (Fig. 6c,f). It confirmed the relative strength of hindbrain, midbrain and thalamic responses to feedback seen across analyses. Population trajectory analysis also revealed asymmetries in response to negative versus positive feedback. For positive feedback, the response was overall stronger, and multiple brain areas exhibited coherent approximately 10 Hz oscillatory dynamics during reward delivery that was phase-locked across brain areas (Fig. 6j,k,l) and sessions (Extended Data Fig. 12). Across-session coherence was visible as a large oscillatory signal in an example area: the IRN (Fig. 6j,k). These oscillatory dynamics were missing during negative feedback and were closely related to licking behaviour[73–76] (Extended Data Fig. 12), being stronger when reward was delivered. This result suggests that motor-related activity is the dominant factor over more abstract influences of reward on neural activity.

Assessing response latencies on the basis of the divergence of the trajectories over time showed that the saccade-reorienting and gaze-reorienting brainstem region the PRNr and the primary auditory region the AUDp exhibited the earliest and strongest responses (Fig. 6l,m). Some early responsivity is probably a carry-over from choice-related activity because the latencies were short and several identified areas exhibited high choice responsivity. The responses from auditory areas probably reflect responses to the error tone and the click from the reward delivery valve. This was particularly clear for the IC (a region that is known to relay auditory signals), which had a peak at the start and end of the 0.5-s-long error noise burst (Fig. 6l). After these initial responses, latencies across other brain regions appeared roughly similar, which suggests that there is a common signal broadcast across the brain (Fig. 6d). More detailed trajectory distance and latency scatterplots are provided in Extended Data Fig. 10.

We then applied the encoding model to the responses measured in the 400 ms after stimulus onset. The kernel for correct feedback was the largest single contributor to neural response variance across each trial (mean $\Delta R^2$ of $8.6 \times 10^{-3}$ averaged across all neurons; Extended Data Fig. 13), which exceeded all other kernels (left or right stimulus, left or right wheel movement, incorrect feedback, block probability and wheel speed). This high variance-explaining response to reward

delivery or consumption held across both wide regions of the cortex and subcortical areas. Midbrain and hindbrain areas exhibited particularly strong responses to reward delivery, with many additional regions, including the thalamus and the sensory (AUDp and SSs) and motor (MOp) cortices, also showing sensitivity (Fig. 6e). Removing the regression kernel for correct feedback then refitting the encoding model of an IRN neuron confirmed the large influence of correct feedback on activity (Fig. 6g,h).

In summary, we found that feedback signals are present across nearly all recorded brain regions, with a stronger response to positive feedback (that is, to reward delivery and consumption) and with particularly strong responses in the thalamus, the midbrain and the hindbrain. Further research will be needed to distinguish between responses for an internal expectation of feedback or the initiation of choice-related action versus responses to external feedback.

## Representation of wheel movement

A consistent finding from previous large-scale recordings in mice has been the macroscopic impact of movement on neural activity. That is, both task-related and task-unrelated movements influence activity beyond premotor, motor and somatosensory cortical areas[19,20,77,78]. Here we started from the task-dependent component of movement, namely the movement of the wheel to register a response. We observed that different mice (and potentially the same mouse on different sessions) adopted different strategies for moving the wheel. For instance, some used both front paws, whereas others used only one paw. Turning the wheel is also a relatively complex operation, rather than being a simple, ballistic movement. Thus, one should not expect a simple relationship between these movements and activity in laterally specific motor areas. For simplicity, we restricted our analyses to the activity associated with wheel velocity (signed to distinguish left from right movements) and its absolute value, wheel speed. Furthermore, unlike the other task variables, movement trajectories change relatively quickly, which necessitates different analysis and null control strategies. Accordingly, we only report simple decoding and encoding analyses.

Wheel speed was decodable from 81% of the reportable areas (163 out of 201), with the strongest effect sizes in the hypothalamus (LPO), the hindbrain (MARN and GRN), the midbrain (CLI), the thalamus (CL, PF), the cortex (ORBvl, VISC and VISrl) and the cerebellum (VeCB) (Fig. 7a,e). For example, we could readily decode wheel speed from single trials of activity in the GRN (Fig. 7f).

The encoding analysis confirmed that many regions across the brain were sensitive to the wheel speed during the task, with $\Delta R^2$ showing values up to several times larger than for the other variables considered besides feedback (Fig. 7b,e and Extended Data Fig. 13). The PRNc and GRN in particular stood out in our analysis for the mean $\Delta R^2$ for neurons in these regions (mean $\Delta R^2 = 9.4 \times 10^{-3}$ in the PRNc and $\Delta R^2 = 179 \times 10^{-3}$ in the GRN). Many other cortical (for example, MOs) and subcortical (for example, GPe, GPi and CP) regions had less substantial, but still above-average, correlations with the wheel speed relative to other regressors (Fig. 7b,e).

Wheel velocity was also significantly decodable from a similar collection of areas as wheel speed (Fig. 7c,e,g and Supplementary Fig. 7a) and was also duly encoded (Fig. 7d,e), albeit with generally smaller values of $\Delta R^2$ (Fig. 7h). The apparently high decodability of velocity was unexpected given the complexities of wheel movement (as mentioned above). Indeed, the uncorrected values of $R^2$ for decoding speed were substantially larger than those for velocity in most regions (Supplementary Fig. 7). However, the null distribution based on imposter sessions (that is, wheel movements from other sessions, including from other mice; Methods) could be decoded more accurately for speed than for velocity (Supplementary Fig. 7c), which reduced the significance of the decoding of speed. We attributed this excess decodability of the null

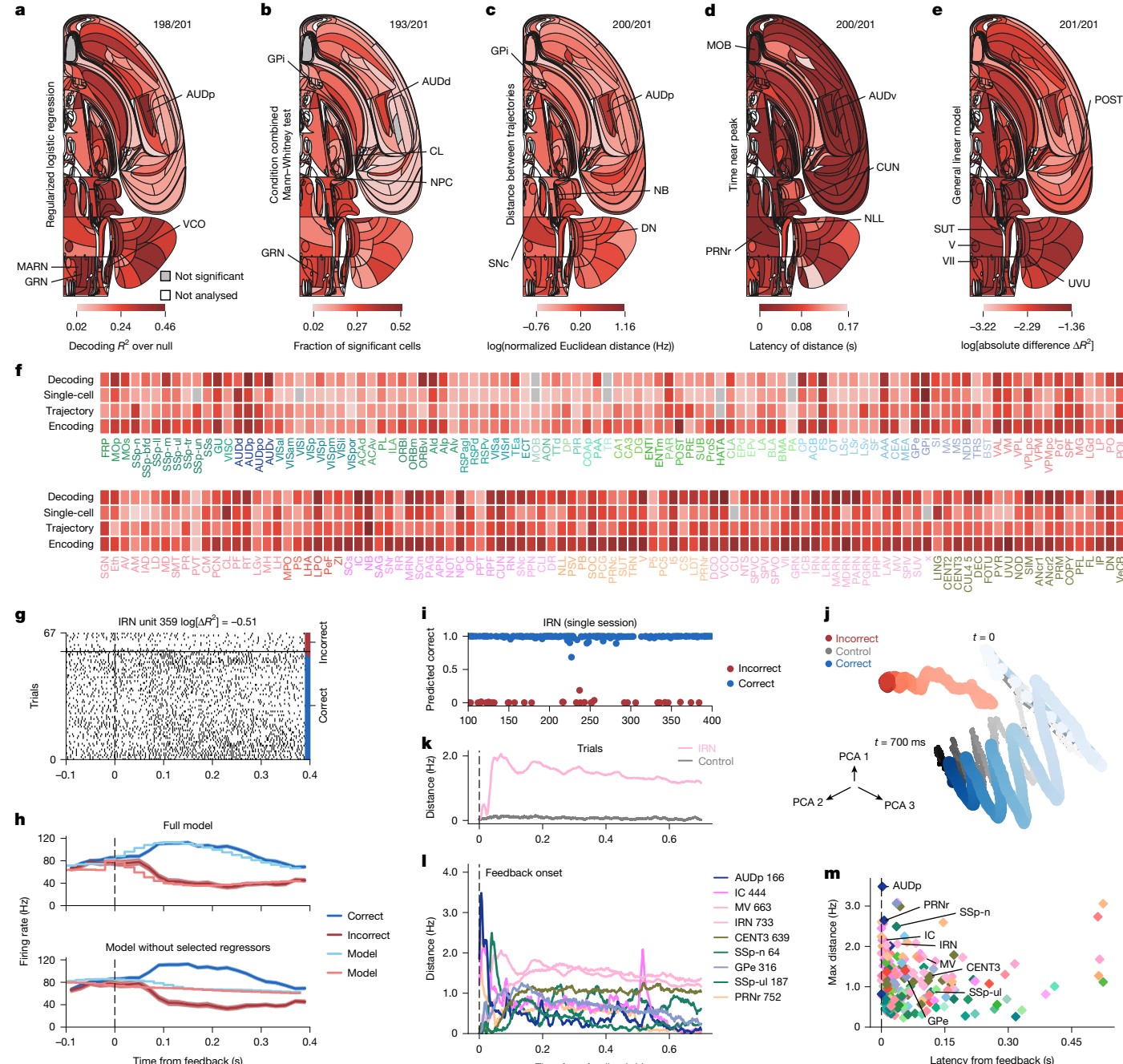

**Fig. 6 | Representation of feedback.** See also Extended Data Figs. 5, 7 and 14 and data in the IBL Brain Atlas (https://atlas.internationalbrainlab.org/?alias=bwm_feedback). Figure parts and statistics are as described in Fig. 4. The analysed feedback interval is 0–200 ms following feedback onset. **a**, Decoding. Null-corrected median balanced accuracy. **b**, Single-cell statistics. Fraction of neurons modulated by feedback compared with activity during baseline (−200 to 0 ms aligned to the stimulus onset). Mann–Whitney and condition combined Mann–Whitney tests at $P < 0.001$ and $P < 0.05$, respectively. Significance was based on the binomial distribution of false positive-events and $FDR_{0.01}$. **c**, Population trajectory distance. Time-resolved maximum Euclidean distance ($d_{max}$ in spikes per s for dimension = number of cells per region, $\log_{10}$) between trajectories for correct versus incorrect choices. Significance is relative to a shuffle control and $FDR_{0.01}$. **d**, Population trajectory latency. First time crossing 70% of the $d_{max}$ for significant regions. **e**, Encoding. Mean absolute difference $|\Delta R^2|$ in improvements from 400-ms causal kernels for correct and incorrect feedback aligned to the feedback time. **f**, Effect significance (grey, not significant; **a**–**c**) and size (by darkness; **a**–**c**,**e**). **g**, Spike raster for an example IRN neuron (Supplementary Table 3). Trials per condition are shown in temporal order, with every third trial shown. **h**, Top,

PETHs (shading indicates ±1 s.e.m.) aligned to first wheel-movement time on correct (blue) and incorrect (red) trials and the full encoding model prediction for an example neuron in **g**. Bottom, the same PETHs but for predictions from a model that omitted correct and incorrect feedback regressors. **i**, Decoded probability of reward receipt coloured by true feedback from 39 neurons in the IRN (Supplementary Table 3). **j**, Trial-averaged population trajectories from incorrect and correct trial-averaged activity across the IRN in three PCA dimensions. Dots, single time bins; darker colours indicate later times. The oscillation of the blue trajectory correlated with licking. Grey (control) pseudo-trajectories from averaging randomized trials, with shuffling choice types within classes of stimulus side and block. **k**, Trajectory distance between correct and incorrect trials in the IRN. Grey, pseudo-trajectory distances. **l**, Trajectory distances across (with neuron numbers indicated) regions showing early response in, for example, auditory areas and prolonged feedback type modulation with time in others. The IC relays auditory signals, which explains the peaks at onset (0.5 s), when the noise burst starts and ends on incorrect trials. **m**, Maximal population trajectory distance and modulation latency (diamonds, significant regions; dots, not significant regions). Extended Data Figure 10c,f,i shows a longer time window and more neurons.

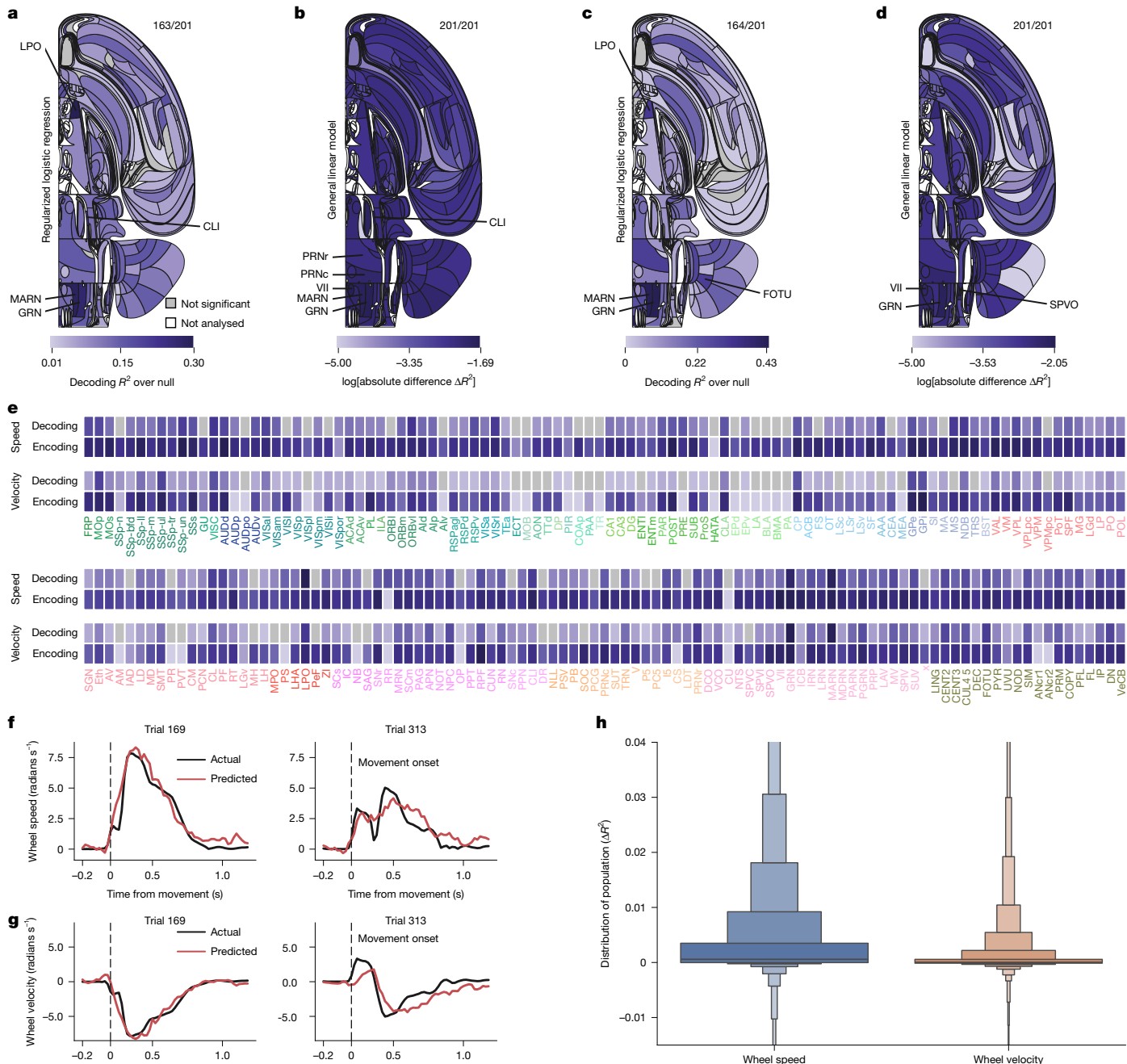

**Fig. 7 | Representation of wheel movement.** See also Extended Data Fig. 5, Supplementary Fig. 7 and the IBL Brain Atlas website for speed data (https://atlas.internationalbrainlab.org/?alias=bwm_wheel_speed) and velocity data (https://atlas.internationalbrainlab.org/?alias=bwm_wheel_velocity). For **a**–**d**, colour indicates the effect size; grey not significant at $FDR_{0.01}$; ratios on the upper right indicate significant/total regions; white, regions not analysed. Wheel movement was decoded based on 20-ms bins from 200 ms before, until 1,000 ms after, first wheel-movement onset. **a**, Wheel speed decoding. Null-corrected median balanced accuracy. **b**, Wheel speed encoding. Mean improvement ($\Delta R^2$) per region from including, as the regressor, 200-ms anticausal temporal kernels convolved with the trace of wheel speed. **c**,**d**, As for **a** and **b** but for velocity. Encoding results were based on a completely separate model fit. **e**, Effect significance (grey, not significant; white, not analysed; **a**,**c**) and size (by darkness; **a**–**d**) by region. **f**, Actual and decoded wheel speed for an example trial from 68 GRN neurons in the GRN (Supplementary Table 3). **g**, Same as **f** but for velocity. **h**, Truncated distributions of additional variance $\Delta R^2_{\text{wheel}}$ explained across all neurons for speed or velocity as base signals.

distribution to the more stereotyped, that is, less variable, trajectory of speed (Supplementary Fig. 7d).

We also correlated neural activity with behavioural movement traces extracted from videos (nose, paw, pupil and tongue). To test for significance, we used the linear-shift method to compare the correlation of spiking activity with behavioural movement variables against a null distribution in which the movement variables were shifted in time[79]

(Methods and Extended Data Fig. 15a). More than half of the neurons in most brain regions were significantly correlated with at least one behavioural variable (Extended Data Fig. 15b).

The widespread relationship between neural activity and motion has various potential sources. These include the specific details of motor planning and execution, efference copy[80], somatosensory feedback and the suppression of input associated with self-motion[81]. More subtle

effects such as the change in other sensory inputs caused by the movement[19] or even prediction errors associated with incompetent execution that can fine-tune future performance[82], may also be involved. Others are more general, including arousal and the calculation and processing of the costs of movement (which would then be balanced against future gain)[83]. More generally, of the components that are indeed specific, only a fraction is likely to be associated with the wheel movement that we monitored compared with other task-related motor actions. This is especially true given the results of previous studies[19,20] showing how important uninstructed movements are in modulating a swathe of neural activity.

## Discussion

Building on previous efforts to build large-scale maps of activity in the mouse at neuronal resolution using Neuropixels probes[1,16,17] and in other species using imaging (for example, refs. 84–86), we assembled a brain-wide electrophysiological map by pooling data from numerous laboratories that used the same standardized and reproducible perceptual decision-making task for mice[23]. Rigorous statistical analyses of this brain-wide consensus map of neural activity showed that neural activity throughout the entire brain correlates with some aspects of the task but with large differences in the ubiquity of representation of different task variables (Figs. 4–7; see Extended Data Figs. 2, 5 and 7 for side by side comparisons).

The neural representations of feedback (Fig. 6) and movement[1,19,20] (Fig. 7) were particularly widespread. The former may also partially or primarily reflect the licking movements required for reward consumption rather than the hedonic aspects of reward as such. Distinguishing these possibilities would require experiments that involve recording activity during the presentation of reinforcement with no motor correlates, such as optogenetic stimulation of dopamine systems[87,88]. Alternatively, correlates of rewards with a motor correlate to the same movements when their hedonic reward is devalued, for example, by satiation, can be compared. The brain-wide correlates of movement could potentially reflect a brain-wide change in the state of neural processing during movement periods along with specific encoding of motor features. The hypothesis of a brain-wide state change is consistent with findings that the neural representation of upcoming movements in the cortex is widespread, although not all of this activity is causally related to the performance of the movements[47]. Our strategy of reducing individual differences in performance impedes a complete analysis of the relationship between neural activity in particular regions and factors such as the reward rate.

The upcoming choice for a mouse was represented in the activity of neurons across brain systems that included the cortex, the basal ganglia, the thalamus, the midbrain, the hindbrain and the cerebellum (Fig. 5). These representations cannot reflect sensory reafference (that is, responses related to sensory stimuli that occur as part of the movements, such as pressure on the paws and movements of the visual stimuli across the screen). This is because we only analysed the time period before the earliest detectable first wheel-movement time. Moreover, our carefully controlled task design and pseudo-session statistical methods meant that choice coding reported in the single-cell and population trajectory analyses cannot reflect processing of the visual stimulus or nonspecific brain states such as arousal. Instead, these responses reflect aspects of decision formation or motor preparation, which potentially include corollary discharge specific to the chosen action[89,90]. Although many studies have focused on the role of the cortex, the basal ganglia or the midbrain in visual perceptual decisions[1,7,55,56,63–65,91,92], here we discovered that parts of the medulla, the pons and the cerebellum are all selectively responsive with similar timing to those areas. Our data were not able to determine whether these different systems make specific contributions to decision formation and execution. However, they rule out a model in which only a limited set of systems subserve a given behaviour according to specific task demands.

The visual stimulus (Fig. 4) was represented (before movement) in a more restricted manner. Its processing followed a temporal sequence through traditional visual areas from the visual thalamus to the cortex and to midbrain and hindbrain regions, the activity of which also correlated with choices. Notably, the temporal structure of activity in these two groups of regions differed. Visual representations in classical visual regions showed a transient representation of the stimuli, whereas activity in the midbrain and hindbrain showed later, ramping activity, consistent with a role of this activity in decision-making. The fact that visual information was found in hindbrain regions such as the GRN and the PRNr, even after accounting for correlates of choice, suggests that these regions have a role in all phases of the cognitive decision-making process rather than simply low-level motor control.

Although more than half of the recorded neurons in most brain regions were significantly modulated by at least some aspect of the task, our ability to explain the total variance of single neurons was limited (Extended Data Fig. 13). This finding indicates that the bulk of activity in the brain is not modulated by the task. It may instead be related to uninstructed movements[19,20] or other processes that are not timed to the task events. Even for the activity that is modulated by the task, it is notable that external cue-driven responses were consistently smaller than internally generated signals, such as those arising in relation to the integration of the stimulus and movement planning. However, the absence of evidence for a neural representation of a task variable in a given region cannot be taken to indicate evidence of absence. This is particularly important to keep in mind because here, for robustness, we used simple variants of analysis methods rather than, for instance, extensively parameterized deep neural networks. Furthermore, our recordings may include implicit biases; for example, spike sorting may be more challenging where cell bodies are more densely packed. Nevertheless, our freely available dataset provides a rich resource for in-depth investigations of brain-wide neural computations. Such studies may include detailed analyses at the level of subregions (for example, cortical layers or functional zones of the striatum) and cell types (as identifiable from extracellular waveforms, such as broad versus narrow spike shapes in the cortex).

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

**International Brain Laboratory**

Leenoy Meshulam[19], Dora Angelaki[20], Julius Benson[20], Isaiah McRoberts[20], Jean-Paul Noel[20], Jaime Arlandis[21], Niccolò Bonacchi[21], Kcenia Bougrova[21], Joana A. Catarino[21], Fanny Cazettes[21], Davide Crombie[21], Eric EJ DeWitt[21], Laura Freitas-Silva[21], Inês C. Laranjeira[21], Zachary F. Mainen[21], Guido T. Meijer[21], Pranav Rai[21], Georg Raiser[21], Florian Rau[21], Michael M. Schartner[21], Olivier Winter[21], Anne E. Urai[22], Valeria Aguillon-Rodriguez[11], Cristian Soitu[11], Anthony M. Zador[11], Christopher S. Krasniak[11,23], Yang Dan[24], Fei Hu[24], Brandon Benson[25], Surya Ganguli[25], Luigi Acerbi[26], Gaelle A. Chapuis[26], Charles Findling[26], Berk Gercek[26], Felix Huber[26], Alexandre Pouget[26], Hailey Barrell[27], Dan Birman[27], Kim Miller[27], Kai Nylund[27], Noam Roth[27], Nicholas A. Steinmetz[27], Matthew Tucker[27], Kenneth Yang[27], Ila Rani Fiete[28], Ari Liu[28], Rylan Schaeffer[28], Anne K. Churchland[29], Felicia Davatolhagh[29], Anup Khanal[29], Maxwell Melin[29], Masayoshi Murakami[30], Sophie Denève[31], Ivan Gordeliy[31], Mandana Ahmadi[32], Jaweria Amjad[32], Naoki Hiratani[32], Sanjukta Krishnagopal[32], Peter Latham[32], Alberto Pezzotta[32], Zekai Xu[32], Kush Banga[33], Jai Bhagat[33], Mayo Faulkner[33], Kenneth D. Harris[33], Michael Krumin[33], Samuel Picard[33], Carolina Quadrado[33], Cyrille Rossant[33], Miles J. Wells[33], Lauren E. Wool[33], Matteo Carandini[34], Agnès Landemard[34], Karolina Z. Socha[34], Sebastian A. Bruijns[35], Peter Dayan[35], Julia M. Huntenburg[35], Debottam Kundu[35], Farideh Oloomi[35], Charline Tessereau[35], Zoe C. Ashwood[36], Tatiana Engel[36], Robert Fetcho[36], Laura M. Haetzel[36], Christopher Langdon[36], Brenna McMannon[36], Zeinab Mohammadi[36], Alejandro Pan Vazquez[36], Jonathan W. Pillow[36], Nicholas A. Roy[36], Yanliang Shi[36], Ilana B. Witten[36], Robert Campbell[14], Naureen Ghani[14], Sonja B. Hofer[14], Hernando Martinez-Vergara[14], Nathaniel J. Miska[14], Thomas Mrsic-Flogel[14], Steven J. West[14], Yaxuan Yang[14], Karel Svoboda[37], Marsa Taheri[9], Michael Häusser[7,13], Petrina Y. P. Lau[7], Amalia Makri-Cottington[7], Sabrina Perrenoud[7], Larry Abbot[38], Hannah M. Bayer[38], Julien Boussard[38], E. Kelly Buchanan[38], Michele Fabbri[38], Cole Hurwitz[38], Christopher Langfield[38], Hyun Dong Lee[38], Catalin Mitelut[38], Liam Paninski[38], Kamron Saniee[38], Erdem Varol[38], Shuqi Wang[38], Matthew R. Whiteway[38], Charles Windolf[38], Han Yu[38] & Yizi Zhang[38]

[19]Center for Computational Neuroscience, University of Washington, Seattle, WA, USA. [20]Center for Neural Science, New York University, New York, NY, USA. [21]Champalimaud Center for the Unknown, Lisboa, Portugal. [22]Cognitive Psychology Unit, Institute of Psychology and Leiden Institute for Brain and Cognition, Leiden University, Leiden, The Netherlands. [23]Watson School of Biological Science, Cold Spring Harbor, NY, USA. [24]Department of Molecular and Cell Biology, University of California, Berkeley, CA, USA. [25]Department of Applied Physics, Stanford University, Stanford, CA, USA. [26]Department of Basic Neuroscience, University of Geneva, Geneva, Switzerland. [27]Department of Biological Structure, University of Washington, Seattle, WA, USA. [28]Department of Brain and Cognitive Sciences, Massachusetts Institute of Technology, Cambridge, MA, USA. [29]Department of Neurobiology, University of California, Los Angeles, Los Angeles, CA, USA. [30]Department of Physiology, University of Yamanashi, Yamanashi, Japan. [31]Département D'études Cognitives, École Normale Supérieure, Paris, France. [32]Gatsby Computational Neuroscience Unit, University College London, London, UK. [33]Institute of Neurology, University College London, London, UK. [34]Institute of Opthalmology, University College London, London, UK. [35]Max Planck Institute for Biological Cybernetics, Tübingen, Germany. [36]Princeton Neuroscience Institute, Princeton University, Princeton, NJ, USA. [37]The Allen Institute for Neural Dynamics, Seattle, WA, USA. [38]Zuckerman Institute, Columbia University, New York, NY, USA.

## Methods

All experimental procedures involving animals were conducted in accordance with local laws and approved by the relevant institutional ethics committees. Approvals were granted by the Animal Welfare Ethical Review Body of University College London, under licences P1DB285D8, PCC4A4ECE and PD867676F, issued by the UK Home Office. Experiments conducted at Princeton University were approved under licence 1876-20 by the Institutional Animal Care and Use Committee (IACUC). At Cold Spring Harbor Laboratory, approvals were granted under licences 1411117 and 19.5 by the institutional IACUC. The University of California at Los Angeles granted approval through IACUC licence 2020-121-TR-00. Additional approvals were obtained from the University Animal Welfare Committee of New York University (licence 18-1502), the IACUC at the University of Washington (licence 4461-01), the IACUC at the University of California, Berkeley (licence AUP-2016-06-8860-1) and the Portuguese Veterinary General Board (DGAV) for experiments conducted at the Champalimaud Foundation (licence 0421/0000/0000/2019).

### Animals

Mice were housed under a 12–12-h light–dark cycle (normal or inverted depending on the laboratory) with food and water available ad libitum, except during behavioural training days. Electrophysiological recordings and behavioural training were performed during either the dark or light phase of the cycle depending on the laboratory. The data from $n = 139$ adult mice (C57BL/6; 94 male and 45 female, obtained from either Jackson Laboratory or Charles River) were used in this study. Mice were aged 13–178 weeks (mean 44.96 weeks, median 27.0 weeks) and weighed 16.1–35.7 g (mean 23.9 g, median 23.84 g) on the day of electrophysiological recordings. We did not attempt to standardize other variables such as temperature, humidity and environmental sound, but we regularly documented and measured them[23].

### Headbar implant surgery

A detailed account of the surgical methods for the headbar implant is provided in appendix 1 of ref. 23. In brief, mice were anaesthetized with isoflurane and head-fixed in a stereotaxic frame. The fur was then removed from their scalp, which was subsequently removed along with the underlying periosteum. Once the skull was exposed, Bregma and Lambda were marked. The head was positioned along the anterior–posterior and left–right axes using stereotaxic coordinates. The head bar was then placed in one of three stereotactically defined locations and cemented (Super-Bond C&B) in place. Future craniotomy positions were marked on the skull relative to Bregma. The exposed skull was then covered with cement and clear UV curing glue (Norland Optical Adhesives).

### Materials and apparatus

For detailed parts lists and installation instructions for the training rigs, see appendix 3 of ref. 23; for the electrophysiology rigs, see appendix 1 of ref. 25.

Each laboratory installed a standardized electrophysiological rig, which differed slightly from the apparatus used during behavioural training[23]. The structure of the rig was constructed from Thorlabs parts and was placed on an air table (Newport, M-VIS3036-SG2-325A) surrounded by a custom acoustic cabinet. A static headbar fixation clamp and a 3D-printed mouse holder were used to hold a mouse such that its forepaws rested on the steering wheel (86652 and 32019, Lego)[23]. Silicone tubing controlled by a pinch valve (225P011-21, NResearch) was used to deliver water rewards to the mouse. Visual stimuli were displayed on an LCD screen (LP097Q ×1, LG). To measure the timing of changes in the visual stimulus, a patch of pixels on the LCD screen flipped between white and black at every stimulus change, and this flip was captured with a photodiode (Bpod Frame2TTL, Sanworks).

Ambient temperature, humidity and barometric air pressure were measured using a Bpod Ambient module (Sanworks), and the wheel position was monitored with a rotary encoder (05.2400.1122.1024, Kubler). Videos of mice were recorded from 3 angles (left, right and body) with USB cameras (CM3-U3-13Y3M-CS, Point Grey) sampling at 60, 150 and 30 Hz, respectively (for details, see appendix 1 of ref. 25). A custom speaker (Hardware Team of the Champalimaud Foundation for the Unknown, v.1.1) was used to play task-related sounds, and an ultrasonic microphone (Ultramic UM200K, Dodotronic) was used to record ambient noise from the rig. All task-related data were coordinated using a Bpod State Machine (Sanworks). The task logic was programmed in Python, and the visual stimulus presentation and video capture were handled by Bonsai[93] and the BonVision package[94].

Neural recordings were made using Neuropixels probes, either v.1.0 (3A or 3B2, $n = 109$ and $n = 586$ insertions, respectively) or v.2.4 ($n = 4$ insertions) (Imec[13]), which were advanced into the brain using a micromanipulator (Sensapex, uMp-4). Typically, the probes were tilted at a 15° angle from the vertical line. Data were acquired using an FPGA (for 3A probes) or PXI (for 3B and 1.0 probes, National Instruments) system using SpikeGLX, and stored on a PC.

### Habituation, training and experimental protocol

For a detailed protocol on animal training, see the methods in refs. 23,25. In brief, at the beginning of each trial, the mouse was required to not move the wheel for a quiescence period of 400–700 ms. After the quiescence period, a visual stimulus (Gabor patch) appeared on either the left or right (±35° azimuth) of the screen, with a contrast randomly sampled from a predefined set (100, 25, 12.5, 6 or 0%). A 100-ms tone (5-kHz sine wave) was played at stimulus onset. Mice had 60 s to move the wheel and make a response. Stimuli were yoked to the rotation of the response wheel, such that a 1-mm movement of the wheel moved the stimulus by 4 visual degrees. A response was registered if the centre of the stimulus crossed the ±35° azimuth line from its original position. If the mouse correctly moved the stimulus 35° to the centre of the screen, it immediately received a 3-µl reward; if it incorrectly moved the stimulus 35° away from the centre, it received a time out. If the mouse responded incorrectly or failed to reach either threshold within the 60-s window, a white-noise burst was played for 500 ms and the inter-trial interval was between 1 and 1.5 s. In trials for which the visual stimulus contrast was set to 0%, the mouse had to respond as for any other trial by turning the wheel in the correct direction (assigned according to the statistics of the prevailing block) to receive a reward, but the mouse was not able to perceive whether the stimulus was presented on the left or right side of the screen. The mouse also received feedback (noise burst or reward) on 0% contrast trials.

Each session started with 90 trials in which the probability of a visual stimulus appearing on the left or right side was equal. Specifically, the 100%, 25%, 12.5% and 6% contrast trials were each presented 10 times on each side, and the 0% contrast was presented 10 times in total (that is, the ratio of the 100, 25, 12.5, 6 and 0% contrasts were set at 2, 2, 2, 2 and 1, respectively). The side (and thus correct movement) for the 0% contrast trials was chosen randomly between the right and left with equal probability. This initial block of 90 trials is referred to as the unbiased block (50:50).

After the unbiased block, trials were presented in biased blocks: in right-bias blocks, stimuli appeared on the right on 80% of the trials, whereas in left-bias blocks, stimuli appeared on the right on 20% of the trials. The ratio of the contrasts remained as above (2:2:2:2:1). Whether the first biased block in a session was left or right was randomly chosen, and blocks were then alternated. The length of a block was drawn from an exponential distribution with scale parameter of 60 trials, but truncated to lie between 20 and 100 trials.

The automated shaping protocol for training[23] involved two collections of sessions. In the first session, the animals started performing a version of the task without biased blocks and were then progressively

introduced to harder stimuli with weaker contrasts as they became progressively more competent. They also experienced a debiasing protocol, which was intended to dissuade them from persisting with just one of the choices. Once they were performing sufficiently well on all non-zero contrasts, they were faced with the biased blocks. When, in turn, performance on those was adequate (including on 0% contrast trials, which are informed by the block), they graduated to recording. Supplementary Figure 8a shows a joint histogram of the number of sessions the mice took in the first and second collections (these were not correlated). Supplementary Figure 8b shows a joint histogram of the number of sessions the mice took in the second collection and the performance during the recording sessions. These were also not correlated.

## Electrophysiological recording using Neuropixels probes

For details on the craniotomy surgery, see appendix 3 of ref. 25. In brief, on the first day of electrophysiological recording, the animal was anaesthetized using isoflurane and surgically prepared. The mouse was subcutaneously administered with analgesics (typically Carprofen). The UV cured glue was removed (typically using a biopsy punch (Kai Disposable Biopsy Punches (1 mm)) or a drill), exposing the skull over the planned craniotomy site (or sites). A test was made to check whether the implant could hold liquid; the bath was then grounded either through a loose or implanted pin. One or two craniotomies (approximately 1 × 1 mm) were made over the marked locations. The dura was left intact, and the brain was lubricated with artificial cerebrospinal fluid. A moisturising sealant was applied over the dura (typically DuraGel (Cambridge NeuroTech) covered with a layer of Kwikcast (World Precision Instruments). The mouse was left to recover in a heating chamber until locomotor and grooming activity were fully recovered.

Mice were head-fixed for recording after a minimum recovery period of 2 h. Once a craniotomy was made, up to four subsequent recording sessions were made in that same craniotomy. Once the first set of craniotomy was fully recorded from, a mouse could undergo another craniotomy surgery in accordance with the institutional licence. Up to two probes were implanted in the brain on a given session. CM-Dil (V22888, Thermo Fisher) was used to label probes for subsequent histology analyses.

## Serial section two-photon imaging

Mice were given a terminal dose of pentobarbital and perfuse-fixed with PBS followed by 4% formaldehyde solution (Thermo Fisher 28908) in 0.1 M PB pH 7.4. The whole mouse brain was dissected and post-fixed in the same fixative for a minimum of 24 h at room temperature. Tissue samples were washed and stored for up to 2–3 weeks in PBS at 4 °C before shipment to the Sainsbury Wellcome Centre for image acquisition. For full details, see appendix 5 of ref. 25.

For imaging, brains were equilibrated with 50 mM PB solution and embedded into 5% agarose gel blocks. The brains were imaged by serial section two-photon microscopy[29,95]. The microscope was controlled with ScanImage Basic (Vidrio Technologies) and BakingTray, a custom software wrapper for setting up the imaging parameters[96]. Image tiles were assembled into 2D planes using StitchIt[97]. Whole brain coronal image stacks were acquired at a resolution of 4.4 × 4.4 × 25.0 μm in *xyz*, with a two-photon laser wavelength of 920 nm and approximately 150 mW at the sample. The microscope cut 50-μm sections but imaged two optical planes in each slice at depths of about 30 μm and 55 μm from the tissue surface. Two channels of image data were simultaneously acquired using multialkali PMTs (green at 525 ± 25 nm; red at 570 nm low pass).

Whole brain images were downsampled to 25-μm isotropic voxels and registered to the adult mouse Allen Common Coordinate Framework[6] using BrainRegister[98], which is an elastix-based[99] registration pipeline with optimized parameters for mouse brain registration. For full details, see appendix 7 of ref. 25.

## Probe track tracing and alignment

Neuropixels probe tracks were manually traced to produce a probe trajectory using Lasagna[100], a Python-based image viewer equipped with a plugin tailored for this task. Traced probe track data were uploaded to an Alyx server[101] (a database designed for experimental neuroscience laboratories). Neuropixels channels were then manually aligned to anatomical features along the trajectory using electrophysiological landmarks with a custom electrophysiology alignment tool[102,103]. For full details, see appendix 6 of ref. 25.

## Spike sorting

The spike-sorting pipeline used at IBL is described in detail in ref. 28. In brief, spike sorting was performed using a modified version of the Kilosort 2.5 algorithm[14]. We found that it was necessary to improve the original code in several aspects (scalability, reproducibility and stability, as discussed in ref. 25); therefore, we developed an open-source Python port (the code repository is provided in ref. 104).

## Inclusion criteria

We applied a set of inclusion criteria to sessions, probes and neurons to ensure data quality. Supplementary Table 1 lists the consequences of these criteria for the number of sessions and probes that passed the criteria.

**Sessions and insertions.** Each Neuropixels insertion was repeated in at least two laboratories, with reproducibility of outcomes across laboratories verified with extensive analyses that we have previously reported[25].

Sessions were included in the data release if the mice performed at least 250 trials, with a performance of at least 90% correct on 100% contrast trials for both left and right blocks, and, to be able to analyse the feedback variable, if there were at least 3 trials with incorrect choices (after applying the trial exclusions below). Furthermore, sessions were included in the release only if they reached threshold on a collection of hardware tests (definitions are available from GiHub (https://int-brain-lab.github.io/iblenv/_autosummary/ibllib.qc.task_metrics.html)).

Insertions were excluded if the neural data failed the whole recording per visually assessed criteria of the 'Recording Inclusion metrics and Guidelines for Optimal Reproducibility' (RIGOR) from ref. 25, by presenting major artefacts (see examples in ref. 28) or if the probe tract could not be recovered during the histology procedure. Furthermore, only insertions for which alignments had been resolved (see appendix 6 of ref. 25 for definitions) were used in this study.

After applying these criteria, a total of 459 sessions, 699 insertions and 621,733 neurons remained, constituting the publicly released dataset.

**Trials.** For the analyses presented here, trials were excluded if one of the following trial events could not be detected: choice, probabilityLeft, feebackType, feeback times, stimON times and firstMovement times. Trials were further excluded if the time between stimulus onset and the first movement of the wheel (the first wheel-movement time) were outside the range of 0.08–2.00 s.

**Neurons and brain regions.** Neurons generated by the spike-sorting pipeline were excluded from the analyses presented here if they failed one of the three criteria described in ref. 28 (the single unit computed metrics of RIGOR[25]): amplitude > 50 μV; noise cut-off < 20 μV; and refractory period violation. Neurons that passed these criteria were termed well-isolated neurons (or often just 'neurons') in this study. Out of the 621,733 units collected, 75,708 were considered well-isolated neurons. Final analyses were additionally restricted to regions that were designated grey matter in the adult mouse Allen Common Coordinate

framework[6], contained at least five well-isolated neurons per session and were recorded from in at least two such sessions.

## Video analysis

We briefly describe the video analysis pipeline (full details can be found in ref. 105). The recording rigs contained three cameras: one called 'left' at full resolution (1,280 × 1,024) and 60 Hz, filming the mouse from one side; one called 'right', filming the mouse symmetrically from the other side at half resolution (640 × 512) and 150 Hz; and one called 'body' at half resolution and 30 Hz, filming the body of the mouse from above. We developed several quality-control metrics to detect raw video issues such as poor illumination (as infrared light bulbs broke) or accidental misplacement of the cameras[105].

We computed the motion energy (the mean across pixels of the absolute value of the difference between adjacent frames) of the whisker pad areas in the 'left' and 'right' videos (Fig. 1d). The whisker pad area was empirically defined using a rectangular bounding box anchored between the nose tip and the eye, both found using DeepLabCut[106] (DLC; see more below). This metric quantifies motion in the whisker pad area and has a temporal resolution of the respective camera.

We also performed markerless pose estimation of body parts using DLC[31], which is used in a fully automated pipeline in IBL (v.2.1) to track various body parts such as the paws, nose, tongue and pupil (Fig. 1d). In all analyses using DLC estimates, we excluded predictions with likelihood < 0.9. Furthermore, we developed several quality-control metrics for the DLC traces[105].

## RF mapping

At the end of the behavioural task session, for most of the recordings (504 out of 699 insertions), we performed an RF mapping experiment for 5 min. During the RF mapping phase, visual stimuli were random square pixels in a 15 × 15 grid occupying 120° of visual angle both horizontally and vertically. There were three possible colours for each pixel: white, grey and dark. The colour of pixels randomly switched at the frame rate of 60 Hz, with an average duration of around 100 ms (Supplementary Fig. 5a).

To compute the RF, we identified the moments when colour switching occurred for each pixel. We defined the moments when colour brightness increased as the on stimulus onset time, which included the transition from dark to grey and grey to white. Similarly, we defined the moments when colour brightness decreased as the off stimulus onset time, which included the transition from grey to dark and white to grey. We then computed the average spike rate aligned with on and off stimulus onset for each pixel, from 0 to 100 ms. We defined two types of RFs, on and off, as the average on and off spike rates, respectively, across pixels.

To estimate the significance of the RF, we fitted the RF to a 2D Gaussian function then compared the variance explained to the fitting of a randomly shuffled receptive field (200 shuffles) and computed the $P$ value of significance. We defined a neuron as having a significant RF if either an on or off RF had $P < 0.01$.

## Assessing significance

In this work, we studied the neural correlates of task and behavioural variables. To assess the significance of these analyses, we needed to properly account for spurious correlations. Spurious correlations can be induced in particular by slow continuous drift in the neurophysiological recordings due to various factors, including movement of the Neuropixels probes in the brain. Such slow drifts can create temporal correlations across trials. Because standard correlation analyses assume that all samples are independent, they can produce apparently significant nonsense correlations even for signals that are completely unrelated[107,108].

Null distributions were generated, which we used to test the significance of our results. Specifically, we used distinct null distributions for each of the three types of variables we considered: a discrete behaviour-independent variable (the stimulus side); discrete behaviour-dependent variables (for example, reward and choice); and continuous behaviour-dependent variables (for example, wheel speed and wheel velocity). For the rest of the section, we denote the aggregated neural activity across $L$ trials and $N$ neurons by $S \in \mathbb{R}^{L \times N}$, and denote the vector of scalar targets across all trials by $C \in \mathbb{R}^L$.

For the discrete behaviour-independent variable, we generated the null distribution from so-called pseudo-sessions. These are sessions generated from the same generative process as the one used for the mice. This process ensured that the time series of trials in each pseudo-session shared the same summary statistics as the ones used in the experiment. We generated $M$ (typically $M = 200$) pseudo-targets $\widetilde{C}_i, i \in [1, M]$, and performed the given analysis on the pair $(S, \widetilde{C}_i)$ and obtained a fit score $\widetilde{F}_i$. In pseudo-sessions, the neural activity $S$ should be independent of $\widetilde{C}_i$ as the mouse did not see $\widetilde{C}_i$ but rather $C$. Any predictive power from $\widetilde{C}_i$ to $S$ (or from $S$ to $\widetilde{C}_i$) would arise, for instance, from slow drift in $S$ unrelated to the task itself. These pseudo-scores $\widetilde{F}_i$ were compared with the actual score $F$ obtained from the neural analysis on $(S, C)$ to assess significance.

For discrete behaviour-dependent variables (such as choice or reward), we could not use the pseudo-session procedure above as we did not know the underlying generative process in the mouse. We therefore used 'synthetic' sessions to create a null distribution. These depended on a generative model of the process governing the choices of the animals. In turn, this required a model of how the animals estimated the prior probability that the stimulus appears on the right or left side of the screen, along with a model of its response to different contrasts given this estimated prior. In a companion paper on the subjective prior[24], we found that the best model of the prior across all animals uses a running average of the past actions as a subjective prior of the side of the next stimulus, which we refer to as the 'action-kernel' model. The subjective prior $\pi_t$ follows the update rule:

$$\pi_{t+1} | \pi_t, a_t, \alpha = (1 - \alpha) \cdot \pi_t + \alpha \cdot \mathbb{I}(a_t > 0)$$

with $a_t \in \{-1, 1\}$ {(left, right)} the action performed by the mouse on trial $t$ and $\alpha$ the learning rate, which we fitted on a session-by-session basis. This effectively modelled how mice use information from previous trials to build a subjective prior of where the stimulus is going to appear at the next trial. The details of how this prior is integrated with the stimulus to produce a decision policy is described in the companion paper[24].

We fit the parameters of this model of the mouse's decision-making behaviour separately for each session and then created 'synthetic' targets $\widetilde{C}_i$ for that session by applying the model (with those fitted parameter values) to stimuli generated from pseudo-sessions to obtain time series of choice and reward. Then, as for the pseudo-sessions above, we obtained pseudo-scores $\widetilde{F}_i$ based on $(S, \widetilde{C}_i)$ and assessed significance by comparing the distribution of pseudo-scores to the actual score $F$ obtained from the neural analysis on $(S, C)$.

For the third type of variable—continuous behaviour-dependent variables such as wheel speed—generating synthetic sessions was harder as we did not have access to a reasonable generative model of these quantities. We instead used what we call 'imposter' sessions, which were generated from the continuous behaviour-dependent variable from another mouse on another session. In detail, an imposter session for an original session of $L$ trials was generated by performing the following steps:

1. Concatenating trials across all sessions analysed in this study (leaving out the session under consideration).
2. Randomly selecting a chunk of $L$ consecutive trials from these concatenated sessions.
3. Returning the selected chunk, the imposter session.

The continuous behaviour-dependent variable could then be extracted from the imposter session. As with the pseudo-sessions and

the synthetic sessions, we obtained pseudo-scores $\widetilde{F}_i$ from a collection of imposter sessions and assessed significance by comparing the distribution of pseudo-scores to the actual score $F$ obtained from the neural analysis on $(S, C)$.

To apply the linear shift method[79,109] to compare spiking with movement variables, we first truncated both the movement and spiking time series by removing $n = 20$ samples from both ends of both time series. We computed the Pearson's correlation coefficient of the central segments and compared the square of this coefficient to a null distribution obtained by repeatedly shifting the spiking time series linearly from the beginning to the end of the full behavioural time series. Significance was assessed using the approximate criterion, rejecting the null with significance $\alpha = 0.05$ if the unshifted correlation was in the top $\alpha(2n + 1)$ of the shifted values.

Additional information about assessing significance for individual analyses are detailed in the analysis-specific sessions below. For decoding, single-cell and population trajectory analyses, the results come in the form of per-region $P$ values. We used the FDR to correct for comparisons across all the regions involved in each analysis (201 for the main figures) at a level of $q = 0.01$. We used the Benjamini–Hochberg procedure[110] as we expected substantial independence among the tests. As noted, we were not able to assess significance for the encoding analysis because of a lack of a convenient null distribution.

## Overview of decoding

We performed a decoding analysis to measure how much information the activity of populations of neurons contained about task variables such as stimulus side and choice. To do this we, used cross-validated, maximum-likelihood regression with L1 regularization (to zero out the contribution from noisy neurons). The neural regressors were defined by binning the spike counts from each neuron in each session in a given region in a specific time window on each trial. The duration of the time window, the number of bins in that time window (that is, the bin size) and the trial event to which it was aligned depended on the variable that is the target of our regression (Supplementary Table 2). These factors are discussed further below and include a variety of behavioural and task variables: stimulus side, choice, feedback and wheel speed and velocity. Although a session may have included multiple probe insertions, we did not perform decoding on these probes separately because they are not independent. Instead, neurons in the same session and region were combined across probes for our decoding analysis. Decoding was cross-validated and compared with a null distribution to test for significance. A given region may have been recorded on multiple sessions; therefore, in the main figures (Figs. 4–7) the region $P$ value was defined by combining session $P$ values using Fisher's combined probability test, and the region effect size was defined by subtracting the median of the null distribution from the decoding score and reporting the median of the resulting values across sessions. The $P$ values for all regions were then subjected to FDR correction for multiple comparisons at $q = 0.01$.

## Decoding target variables

Stimulus side, choice and feedback were treated as binary target variables for logistic regression. For stimulus side, trials that had zero contrast were excluded. We used the LogisticRegression module from scikit-learn[111] (v.1.1.2) with 0.001 tolerance, 20,000 maximum iterations, "l1" penalty, "liblinear" solver and "fit_intercept" set to True. We balanced decoder classes by weighting samples by the inverse of the class frequency, $1/(2P_{i,\mathrm{class}})$. Decoding performance was evaluated using the balanced accuracy of classification, which is the average of the recall probabilities for the two classes. Supplementary Figure 9 shows histograms of the regression coefficients for all the variables.

Wheel values (speed and velocity) change over the course of a trial, unlike the previous decoding targets, and we therefore had to treat these target variables differently. We averaged wheel values in

nonoverlapping 20-ms bins, starting 200 ms before first wheel-movement time and ending at 1,000 ms after first wheel-movement time. Spike counts were similarly binned. The target value for a given bin (ending at time $t$) was decoded from spikes in a preceding (causal) window spanning $W$ bins (ending at times $t, …, t\text{-}W$). Therefore, if decoding from $n$ neurons, there were $(W + 1)n$ predictors of the target variable in a given bin. In practice, we used $W = 10$. To decode these continuous-valued targets, we performed linear regression using the Lasso module from scikit-learn[111] (v.1.1.2) with 0.001 tolerance, 1,000 maximum iterations and "fit_intercept" set to True. Decoding performance was evaluated using the $R^2$ metric.

## Decoding cross-validation

We performed all decoding using nested cross-validation. Each of five outer folds was based on a training and validation set comprising 80% of the trials and a test set of the remaining 20% of trials. We selected trials at random in an interleaved manner. The training and validation set of an outer fold was itself split into five inner folds, again using an interleaved 80:20% partition. When logistic regression was performed, the folds had to be selected such that the trials used to train the decoder included at least one example of each class. Because both outer and inner folds were selected at random, it was possible that this requirement was not met. In those circumstances, we re-sampled the outer or inner folds. Likewise, we disallowed pseudo and synthetic sessions that had too few class examples. We fit regression models on the 80% training set of the inner fold using regularization coefficients ($10^{-5}, 10^{-4}, 10^{-3}, 10^{-2}, 10^{-1}, 10^{0}$ and $10^{1}$) for logistic regression (input parameter $C$ in sklearn) and ($10^{-5}, 10^{-4}, 10^{-3}, 10^{-2}$ and $10^{-1}$) for linear regression (input parameter $\alpha$ in sklearn). We then used each model to predict targets on the remaining 20% of the trials of the inner fold (that is, the validation set). We repeated this procedure such that each trial in the original training and validation set of the outer fold was used once for the validation set and four times for the training set. We then took the regularization coefficient that performed best across all validation folds and retrained a regression model using all trials in the training and validation set of the outer fold. This final model was used to predict the target variable on the 20% of trials in the test set of the outer fold. We repeated the above train–validate–test procedure five times, each time holding out a different 20% of test trials such that, after the five repetitions, each trial had been included in the test set exactly once and included in the training and validation set exactly four times. The concatenation of all test set predictions, covering 100% of the trials, was used to evaluate the decoding score.

We found that for some regions and sessions, the resulting decoding score was sensitive to the precise assignment of trials to different folds. Therefore, to provide additional robustness to this procedure, we repeated the full fivefold cross-validation over multiple separate runs, each of which used a different random seed for selecting the interleaved training, validation and test splits. We then took the average decoding score across all runs as the final reported decoding score. When decoding stimulus side, choice and feedback, we performed ten runs, and for decoding wheel speed and wheel velocity, we used two runs owing to the added computational burden of decoding the wheel values, which included multiple bins per trial.

To further reduce the sensitivity of decoding scores due to fold allocation, the companion prior paper[24] used a minimum of 250 trials to perform decoding of a given session. We waived that requirement for the decoding analyses in this study to match the same neurons used in the other analyses. We found that relaxing this requirement only affected the significance of a small number of regions for each target variable (Supplementary Fig. 10).

## Decoding significance testing with null distributions

We assessed the significance of the decoding score that resulted from the multirun cross-validation procedure by comparing it to those of

a bespoke null distribution of decoding scores. To construct appropriate null distributions, we fixed the regressor matrices of neural activity and generated new vectors of target values that followed similar statistics (Supplementary Table 2), as described above. Once the new target values were generated, we carried out the full multirun cross-validation procedure described above to obtain a new decoding score. This was repeated multiple times to produce a null distribution of decoding scores: stimulus side, choice and feedback were repeated 200 times, whereas wheel speed and velocity were repeated 100 times to reduce the computational burden (Supplementary Table 3).

The null distribution was used to define a $P$ value for each region–session pair, in which the $P$ value was defined as $1 - \rho$ where $\rho$ was the percentile relative to the null distribution. Each brain region was recorded in $\geq 2$ sessions, and we used two different methods for summarizing the decoding scores across sessions: (1) the median-corrected decoding score among sessions, which was used as the effect size in the main figures (the values were corrected by subtracting the median of the decoding score of the null distribution); and (2) the fraction of sessions in which decoding was significant, that is if the $P$ value was less than $\alpha = 0.05$, which is shown in Extended Data Fig. 6. We combined session-wide $P$ values using Fisher's combined probability test (also known as the Fisher's method[32,33]) when computing a single statistic for a region. Finally, the combined $P$ value for a region was subjected to a FDR correction for multiple comparisons at $q = 0.01$. We note that the combined $P$ value may be significant but the computed effect size may be negative. This is because many sessions used for decoding in that region may have been insignificant, thereby driving the effect size down, whereas a small number of sessions may have been significant, thereby causing the Fisher's combined probability test to produce a significant combined $P$ value.

## Single-cell correlates of sensory, cognitive and motor variables

We quantified the sensitivity of single neurons to three task variables: visual stimulus (left versus right location of the visual stimulus); choice (left versus right direction of wheel turning); and feedback (reward versus non-reward). We computed the sensitivity metric for each task variable using the condition combined Mann–Whitney $U$-statistic[1,112,113] (Supplementary Fig. 2a,c). Specifically, we compared the firing rates from those trials with one task-variable value $V_1$ (for example, trials with the stimulus on the left side) to those with the other value $V_2$ (for example, with the stimulus on the right side) while holding the values of all other task variables fixed. In this way, we could isolate the influence of individual task variables on neural activity. To compute the $U$-statistic, we first assigned numeric ranks to the firing rate observations in each trial. We then computed the sum of ranks $R_1$ and $R_2$ for the observations coming from $n_1$ and $n_2$ trials associated with the task-variable values $V_1$ and $V_2$, respectively. The $U$-statistic is defined as:

$$U = \min\left[ R_1 - \frac{n_1(n_1 + 1)}{2}, R_2 - \frac{n_2(n_2 + 1)}{2} \right]. \quad (1)$$

The probability that the firing rate on $V_1$ trials is different (greater or smaller) from the firing rate on $V_2$ trials is computed as $1 - P$, where $P$ is given by

$$P = \frac{U}{n_1 n_2}, \quad (2)$$

which is equivalent to the area under the receiver operating characteristic curve[114,115]. The null hypothesis is that the distributions of firing rates on $V_1$ and $V_2$ trials are identical.

To obtain a single probability across conditions, we combined observations across different trial conditions $j$ by a sum of $U$-statistic in these conditions[1]:

$$P = \frac{\sum_j U_j}{\sum_j n_{1,j} n_{2,j}}. \quad (3)$$

Here $n_{1,j}$ and $n_{2,j}$ are the numbers of $V_1$ and $V_2$ trials, respectively, in the condition $j$.

For the visual stimulus, we compared firing rate in trials with the stimulus on the left versus stimulus on the right during the time window 0–100 ms aligned to the stimulus-onset time. For choice, we compared firing rates in trials with the left versus right choice during the time window −100 to 0 ms aligned to the first wheel-movement time. For the feedback, we compared firing rate in trials with reward versus non-reward during the time window of 0–200 ms aligned to the feedback-onset time.

To estimate significance, we used a permutation test in which trial labels for one task variable were randomly permuted 3,000 times in each subset of trials with fixed values of all other task variables, and the Mann–Whitney $U$-statistic was computed for each permutation. We computed the $P$ value for each task variable as the fraction of permutations with the statistic $P$ greater than in the data. This approach controlled for correlations among task variables and allowed us to isolate the sensitivity of the neuron to a stimulus that is not due to sensitivity to block and choice and vice versa. Random permutations, however, do not control for spurious correlations that can arise owing to autocorrelations in the time series of the firing rate and task variable[107]. To control for spurious correlations, we used a within-block permutation test to simultaneously control for both temporal correlations and correlations among task variables. Specifically, we generated the null distribution by randomly permuting trial labels with fixed values of all other task variables in each individual block, which effectively reduced the serial dependencies of task variables at the time scale of block duration.

The combined condition Mann–Whitney $U$-statistic is known to have a relatively high false-positive rate owing to the limited number of trials in each condition. To obtain a sufficient number of trials, we also computed a simple Mann–Whitney $U$-statistic without separating different conditions. We defined $P < 0.001$ ($\alpha_{MW} = 0.001$) as the criterion of significance for the simple Mann–Whitney $U$-statistic, and $P < 0.05$ ($\alpha_{CCMW} = 0.05$) for the combined condition Mann–Whitney $U$-statistic. We defined neurons that were significant in both tests to be sensitive neurons for a specific task variable.

To quantify the overall responsiveness of single neurons to the behavioural task, we used the Wilcoxon rank-sum test to compare firing rates between the baseline (−200 to 0 ms window aligned to the stimulus onset) and the following different task periods: 50–150 ms and 0–400 ms aligned to the stimulus onset; −100 to 50 ms and −50 to 200 ms aligned to the first wheel-movement time; and 0–150 ms aligned to the reward delivery. These time windows are selected on the basis of the test of responsiveness in previous work on large-scale neural coding with a similar task structure[1].

To measure the behavioural movement correlates of single neurons in the entire recording sessions, we computed zero time-lag Pearson's correlation coefficients between time series of spike counts in 50-ms bins and time series of four behavioural variables (nose, pupil, paw and tongue) each extracted from videos of the mouse using DLC software[31]. To assess the significance of these correlations, we applied a time-shift test[79] and computed $2K = 40$ time-shifted correlations, varying the offset between time series of spiking activity and behavioural variables from 50 to 1,000 ms (both positive and negative offsets). We then counted the number of times $m$ where the absolute value of time-shifted correlation exceeded that of zero time-lag correlation and assigned the $P$ value as the fraction of the absolute value of permuted correlations greater than in the data $P = m/(2K + 1)$. We then assigned each neuron as being significantly responsive relative to a particular threshold on this $P$ value.

We then computed the fraction of neurons in each brain region that were significantly responsive to the behavioural task, movement, visual stimulus, choice and feedback, and identified brain regions that were most responsive to these conditions. Specifically, for each region, we computed the $P$ value of the fraction of neurons ($f_i$) in $i$-th session by comparing the fraction to a binomial distribution of fractions due to false-positive events: Binomial($N_i, \alpha$), where $N_i$ is the number of neurons in $i$-th session, and $\alpha$ is the false-positive rate:

$$\alpha = \alpha_{MW} \times \alpha_{CCMW} = 0.001 \times 0.05, \text{ for stimulus, choice, and feedback} \qquad (4)$$

We defined the $P$ value $P_i$ as the probability of the fraction $f_i$ that is larger than the distribution Binomial($N_i, \alpha$). Next, we used Fisher's combined probability test to compute a combined $P$ value of each brain region by combining the $P$ values of all sessions ($i = 1, 2, \ldots m$).

After computing combined $P$ values of each brain region, these $P$ values were then subjected to the FDR procedure (Benjamini–Hochberg) at $q = 0.01$ to correct for multiple comparisons. We defined a list of regions to be significant on the basis of this FDR procedure.

**Population trajectory analysis methods**
We examined how responsive different brain regions were to a task variable $v$ of interest. To do so, we constructed a pair of variable-specific supersessions ($s_v, s_v'$): We partitioned all the IBL data into two, corresponding to the opposing pair of conditions for the variable (for example, for stimulus discrimination, we split the trials into the left and right stimulus conditions) and replaced the trial-by-trial responses of each cell in the condition and in each session with one trial-averaged response (Fig. 3e). These trials were aligned to a variable-specific reference time (for example, the stimulus-onset time for stimulus discrimination). We used the canonical time windows shown in Fig. 3a around the alignment time for the main figures unless stated otherwise (for example, for feedback, we used a longer time window in the temporal evolution plot to illustrate licking), time bins of length 12.5 ms and stride of 2 ms. The supersessions $S_v, S_v'$ had a number of rows equalling the number of IBL sessions that passed quality control for that variable condition times the number of cells per session; columns corresponded to time bins.

We then subdivided the supersessions by brain region $r$ ($S_{v,r}, S_{v,r}'$). These defined a pair of across-IBL response trajectories (temporal evolution of the response) to the pair of variable $v$ conditions for each brain region.

We next computed the time-resolved difference in response of brain region $r$ to the opposing conditions of task variable $v$. We restricted our analyses to regions with ≥20 rows in ($S_{v,r}, S_{v,r}'$) for all analyses. Our primary distance metric, which we call $d_{v,r}(t)$, was computed as a simple Euclidean distance in neural space, normalized by the square root of the number of cells in the given region.

Given a time-resolved distance curve, we computed the maximum and minimum distances along the curve to define a variable-specific and region-specific modulation amplitude:

$$A_{v,r} = \max_t[d_{v,r}(t)] - \min_t[d_{v,r}(t)]. \qquad (5)$$

We obtained a variable-specific and region-specific response latency by defining it as the first time $t$ at which $d_{v,r}(t) = \min_t[d_{v,r}(t)] + 0.7(\max_t[d_{v,r}(t)] - \min_t[d_{v,r}(t)])$. Using modulation amplitude as a measure of effect size, we then quantified the combined modulation amplitude and latency of regions as a function of task variable.

To generate a significance measure for the variable-specific and region-specific distance measures, we used a pseudo-trial method for generating null distance distributions, as described below. Distances were significant if they were greater in size than the corresponding null distance distribution with $P < 0.01$. Although the significance of regions was therefore controlled for the effects of

other task variables, note that the distance amplitudes and latencies were not.

Below we list the three task variables examined and the associated null distributions:
- Stimulus supersession: $S_v, S_v'$ corresponded to trials with the stimulus on the left or right, respectively, aligned by the stimulus-onset time and including 0 ms before to 150 ms after onset. To generate pseudo-trials, we permuted the stimulus side labels among trials that shared the same block and choice side.
- Choice supersession: $S_v, S_v'$ corresponded to trials with the animal's response (wheel movement) to the left or right, respectively, aligned by the first wheel-movement time and including 0 ms before to 150 ms after onset. To generate pseudo-trials, we permuted the choice labels among trials with the same block and stimulus side.
- Feedback supersession: $S_v, S_v'$ corresponded to trials in which the animal's response was correct (recall that the feedback was water delivery) or incorrect (recall that the feedback was tone and time out delivery), respectively, aligned by feedback onset and including 0 ms before to 150 ms after onset. To generate pseudo-trials, we permuted the choice labels among trials with the same block and stimulus side and then compared these pseudo-choices with the true stimulus sides to obtain pseudo-feedback types.

For each $S_{v,r}, S_{v,r}'$ pair, we repeated the pseudo-trial process $M$ (1,000) times, then followed the same distance computation procedures described above to obtain a null distribution of $M$ modulation amplitude scores. We obtained a $P$ value by counting $n$ (as the number of pseudo-scores that were greater than the true score for this region) as: $P = \frac{n+1}{M+1}$.

For regions with significant and large effect sizes to a given variable, we generated visualizations of the population dynamics by projecting the trajectories in $S_{v,r}, S_{v,r}'$ into a low-dimensional subspace defined by the first three principal components of the pair $S_{v,r}, S_{v,r}'$. In addition to the main figure results, population trajectory results on the maximal dataset are shown in Extended Data Fig. 10.

**Multiple linear regression model of single-neuron activity**
We fit linear regression models to single-neuron activity, measured as spikes binned into 20-ms intervals. These models aimed to express $\{s_{lt}\}$, the neural activity in time bin $t \in [1, T]$ on trial $l \in [1, L]$ based on $D$ time-varying task-related regressors $X \in \mathbb{R}^{L,T,D}$. We first represented the regressors across time using a basis of raised cosine 'bump' functions in log space[116]. Each basis function was associated with a weight in the regression model, with the value of the basis function at time $t$ described by $\cos\left(\frac{2(t-\tau)\pi}{2w} + \frac{1}{2}\right)$. The basis functions were computed in log space and then mapped into linear time to more efficiently capture both fast neuronal responses in the <100-ms range and slow changes beyond that time (Fig. 3c). The width $w$ and centre $\tau$ of each basis were chosen to ensure even coverage of the total duration of the kernel. In an example kernel with three bases, three separate weights would be fit to the event in question with weights describing early, middle and late activity predicted by the event. These bases were convolved with a vector describing the effects of each regressor. In the case of timing events, the bases were convolved with a Kronecker delta function, which resulted in a copy of the kernel at each time when the event occurred. We describe the simple case that each regressor has the same number $B$ of basis functions. This produced a new regression tensor $\hat{X} \in \mathbb{R}^{L,T,D,B}$.

We then sought regression weights $\beta \in \mathbb{R}^{D,B}$ such that, as closely as possible, $s_{lt} = \beta_0 + \sum_{d,b} \beta_{db} \hat{x}_{ltdb}$, where $\{\beta_{db}\}$ are linear regression weights. Each single-neuron model used regressors for stimulus onset (left and right separately), first wheel-movement time (left or right), correct feedback, incorrect feedback, value of the block probability, movement initiation and wheel speed. Fitting was performed using an L2-penalized objective function (as implemented in the scikit-learn

Python ecosystem as $\|\mathbf{s} - \beta_0 - \hat{X} \cdot \beta\|_2^2 + \alpha \times \|\beta\|_2^2$), with the weight of the regularization $\alpha$ determined through cross-validation. Note that the intercept of the model is not included in the regularization to capture fully the mean of the distribution of $\mathbf{s}$.

We used a kernel composed of five basis functions to parameterize left and right stimulus onset, and correct and incorrect feedback. These bases spanned 400 ms and corresponded to 5 weights per regressor for each of these 4 regressors in the model.

Previous work has shown that difficulty in perceptual decision-making tasks[117], along with neural responses, does not change linearly with contrast. To account for this, we modulated the height of the stimulus-onset kernels as a function of contrast $c$ with height $h = \frac{\tanh 5c}{\tanh 5}$. The resulting kernels would produce a response that was lower at low contrasts for the same set of weights $\{\beta_{db}\}$.

To capture statistical dependencies between wheel movements and spiking, we used anticausal kernels (in which the convolution of signal and kernel produces a kernel peak before peaks in the signal) describing the effect of first wheel-movement time for leftward and rightward movements. These kernels described 200 ms of activity preceding first movement using 3 basis functions. We also used an additional anticausal kernel of 3 bases covering 300 ms describing the effect of wheel speed, and was convolved with the trace of wheel speed for each trial. With these regressors, we aimed to capture preparatory signals that preceded movements related to the wheel.

Models were fit on a per-neuron basis with the L2 objective function using fivefold cross-validation. Trials for cross-validation were chosen from a uniform distribution, and not in contiguous blocks. Models were then fit again using a leave-one-out paradigm, with each set of regressor weights $\beta_{d1}...\beta_{dB}$ being removed as a group and the resulting model fit and scored again on the same folds. The change between the base model score $R_{\text{full}}^2$ and the omission model $R_{-\text{regressor}}^2$ was computed as $\Delta R_{\text{regressor}}^2 = R_{\text{full}}^2 - R_{-\text{regressor}}^2$. Moreover, the sensitivity for several pairs of associated regressors, such as left or right stimulus onset and correct and incorrect feedback, were defined as $\log|\Delta R_A^2 - \Delta R_B^2|$. This computation was applied to the following pairs: right and left stimulus, right and left first wheel-movement time, and correct and incorrect feedback.

## Granger analysis across simultaneously recorded regions

Granger causality has been suggested as a statistically principled technique to estimate directed information flow from a pair of time series[118]. We used nonparametric spectral Granger causality[119], implemented in Python[120], to compute a Granger score for all simultaneously recorded region pairs in the IBL's brain-wide dataset.

For a given session, binned spikes (12.5-ms bin size) from both probes were averaged across regions to obtain a firing rate time series for the complete recording (excluding regions with fewer than ten neurons per recording). These series (typically 1.5-h long) were then divided in nonoverlapping 10-s segments (irrespective of task contingencies or alignment), which resulted in a data input of shape no. of regions × no. of segments × no. of observations from which a Granger score as a function of frequency was computed for each directed region pair with the Spectral Connectivity Python package[120]. We obtained a single Granger score per directed region pair by averaging across frequencies[121].

Significance for a Granger score and region pair for a given session was established using a permutation test. That is, a null distribution of pseudo Granger scores was obtained by randomly swapping the two region labels across segments. A total of 1,000 of these pseudoscores were computed, and a $P$ value was obtained by counting the number of pseudoscores that were greater than the true Granger score and dividing this count by the number of pseudoscores plus 1. $P$ values across all Granger scores were corrected for multiple comparison using the Benjamini–Yekutieli method. Measurements were combined across sessions by taking the mean Granger score and using Fisher's combined probability test to combine the $P$ values.

## Visualization and comparison of results across neural analyses

To facilitate comparisons of neural analyses across brain regions, for each task variable, we visualized effect sizes in a table (for example, Fig. 4f), specifying the effect size for each analysis and brain region. Cells of the table were coloured according to effect size using the same colour map as in the corresponding flatmap. Before summing, the effect sizes for each analysis were normalized to lie in the interval from 0 to 1. This method highlights regions with large effects across all analyses and indicates the extent to which the analyses agree. For a direct comparison of analyses scores, see flatmaps in Extended Data Fig. 2 and scatter plots of scores for analysis pairs in Extended Data Fig. 3.

## Reporting summary

Further information on research design is available in the Nature Portfolio Reporting Summary linked to this article.

## Data availability

Instructions for downloading the data used in this Article are available online (https://int-brain-lab.github.io/iblenv/notebooks_external/data_release_brainwidemap.html). The data can also be browsed online at the IBL website (https://viz.internationalbrainlab.org). The following resources are available from Figshare: a white paper for the released data, with additional details about quality control and metrics (https://figshare.com/articles/preprint/Data_release_-_Brainwide_map_-_Q4_2022/21400815)[22]; the protocol used to train mice (https://figshare.com/projects/A_standardized_and_reproducible_method_to_measure_decision-making_in_mice/74373)[122]; and the pipeline used to perform the electrophysiology recordings and histology validations (https://figshare.com/projects/Reproducible_Electrophysiology/138367)[123].

## Code availability

The code used to produce the results and figures presented in this Article is available from GitHub (https://github.com/int-brain-lab/paper-brain-wide-map).

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

**Acknowledgements** This work was supported by grants from the Wellcome Trust (209558 and 216324), the Simons Foundation, The National Institutes of Health (NIH U19NS12371601), the National Science Foundation (NSF 1707398), the Gatsby Charitable Foundation (GAT3708), the Max Planck Society and the Humboldt Foundation. Part of the data analyses for this project was performed using Stanford University's Sherlock cluster. Another part was performed at the University of Geneva on 'Baobab' and 'Yggdrasil' high-performance computing clusters. We also acknowledge computing resources from the Columbia University's Shared Research Computing Facility project, which is supported by NIH Research Facility Improvement grant 1G20RR030893-01, and associated funds from the New York State Empire State Development, Division of Science Technology and Innovation (NYSTAR) contract C090171. We thank staff at Stanford University and the Stanford Research Computing Center for providing computational resources and support that contributed to these research results; and P.Latham, T.Mrsic-Flogel and IBL colleagues for comments on the manuscript. The production of all IBL platform papers is led by a task force, which defines the scope and composition of the paper, assigns and/or performs the required work for the paper and ensures that the paper is completed in a timely fashion. The task force members for this platform paper include authors A.P., B.G., B.B., C.F., C.L., D.B., F.H., G.A.C., I.R.F., J.M.H., K.D.H., K.Z.S., M.R.W., M.C., M.S., N.J.M., N.A.S., O.W., P.D., T.A.E. and Y.S.

**Author contributions** Detailed author contributions are provided in Supplementary Table 4.

**Competing interests** The authors declare no competing interests.

**Additional information**
**Correspondence and requests for materials** should be addressed to International Brain Laboratory.

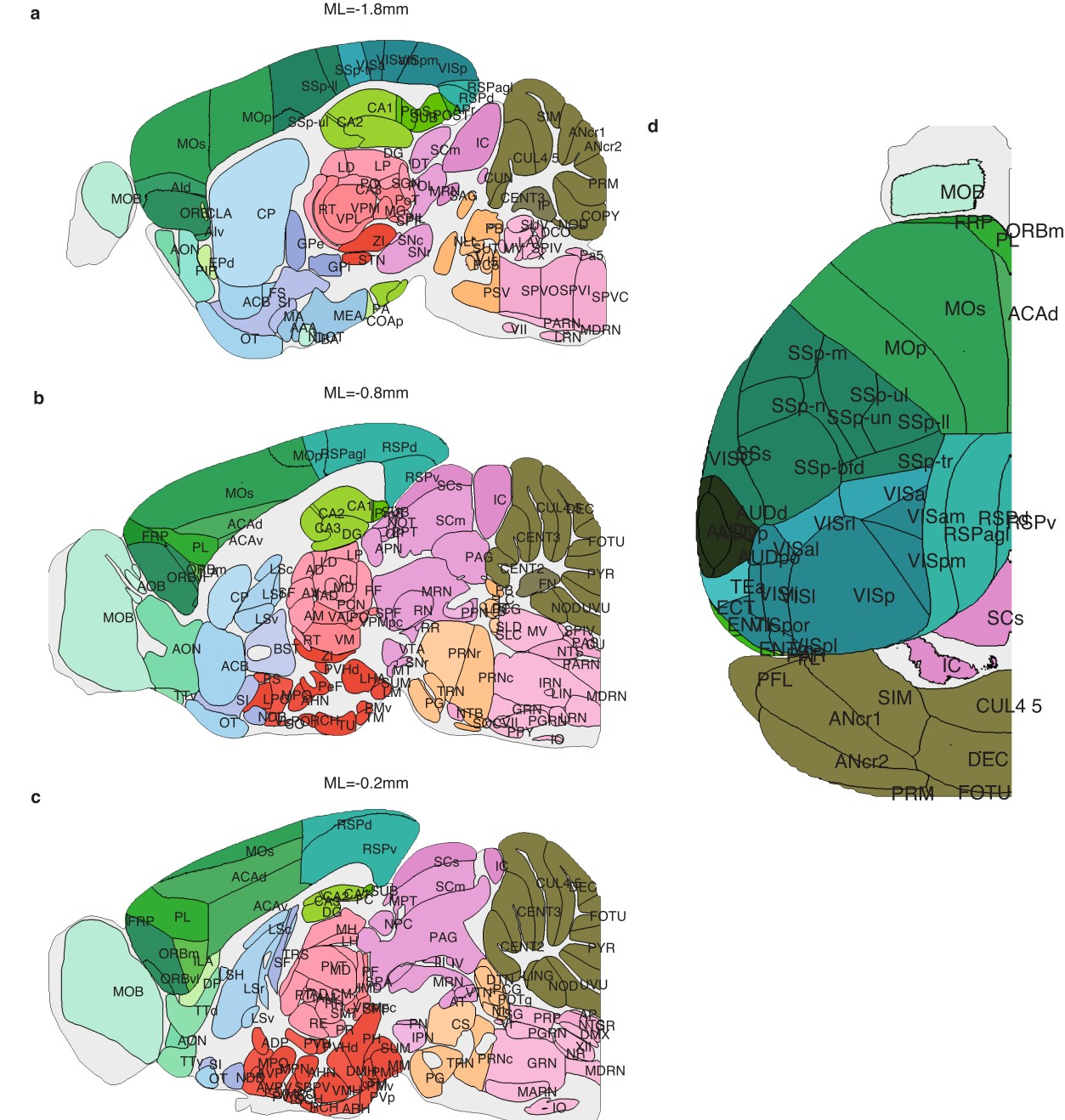

**Extended Data Fig. 1 | 2d-brain slices maps annotated with region acronyms. a)** Region acronyms for sagittal slices with coordinates: ML=−1.8 mm, **b)** ML=−0.8 mm, **c)** ML=−0.2 mm. **d)** Region acronyms for the top view of the dorsal cortex.

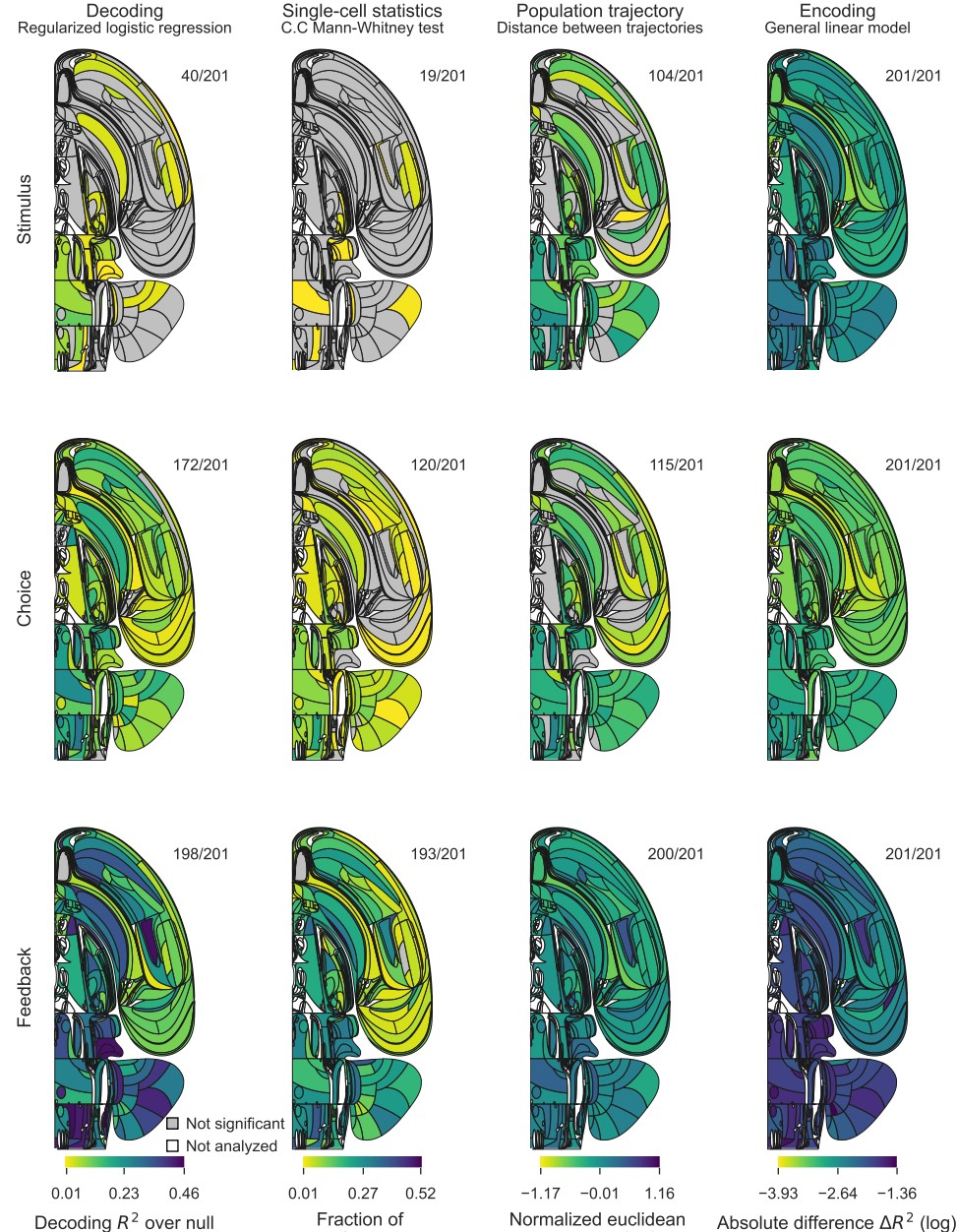

**Extended Data Fig. 2 | Comparison of effect sizes across task variables.** Each column corresponds to a particular neural analysis and each row a task variable. For each analysis, the colour scale is fixed across all variables to enable comparison of effects between variables. For most analyses, the feedback variable has the largest effect amongst all task variables. The numbers at the top right indicate the fraction of significant regions across all analysed regions.

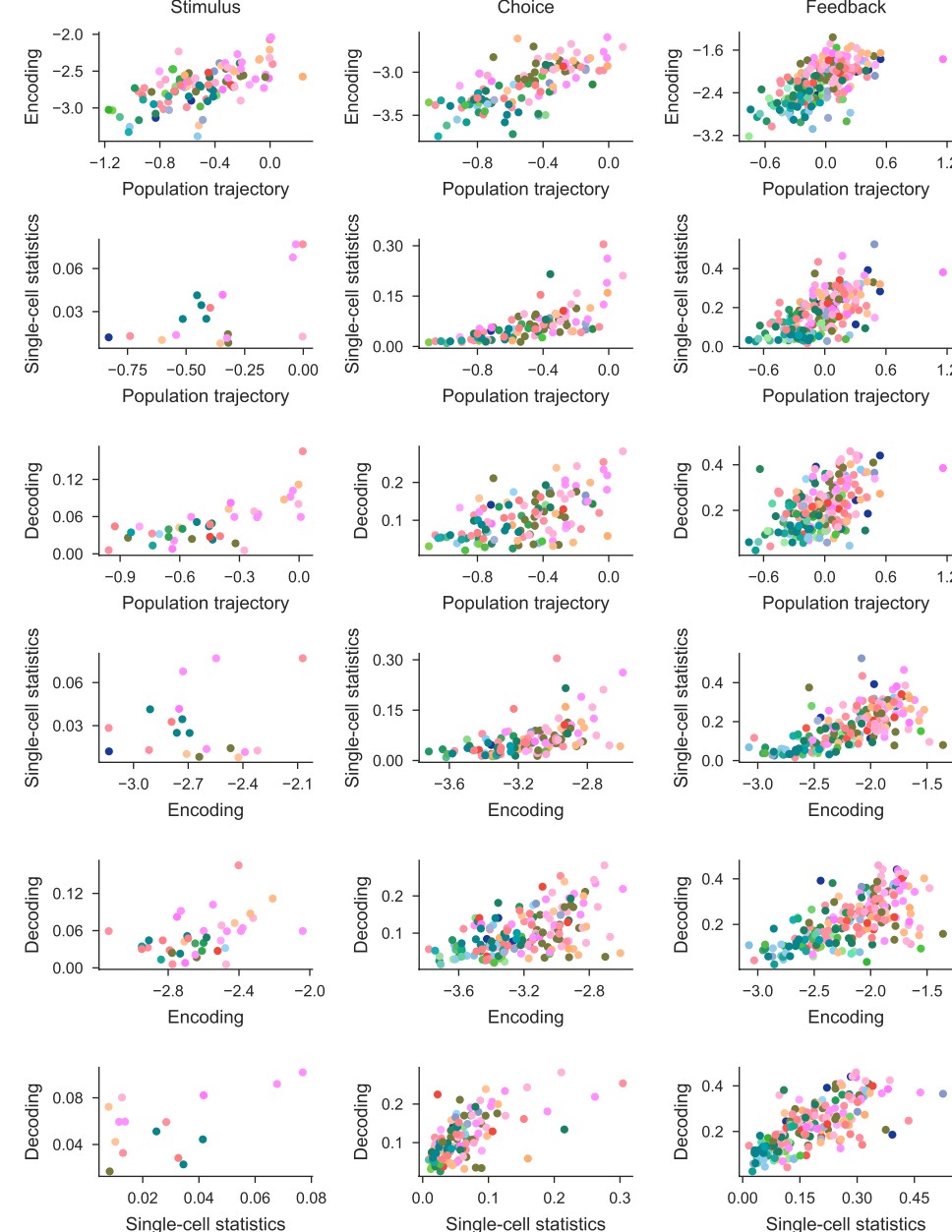

**Extended Data Fig. 3 | Amplitudes of analysis pairs for the three main variables.** For a given analysis pair, say encoding and population trajectory, and a variable, say stimulus, all regions for which both analyses were significant are shown as dots in a scatter plot with the amplitudes as coordinates, colored using our canonical region coloring. There are 6 possible analysis pair combinations (rows) and 3 main variables (columns).

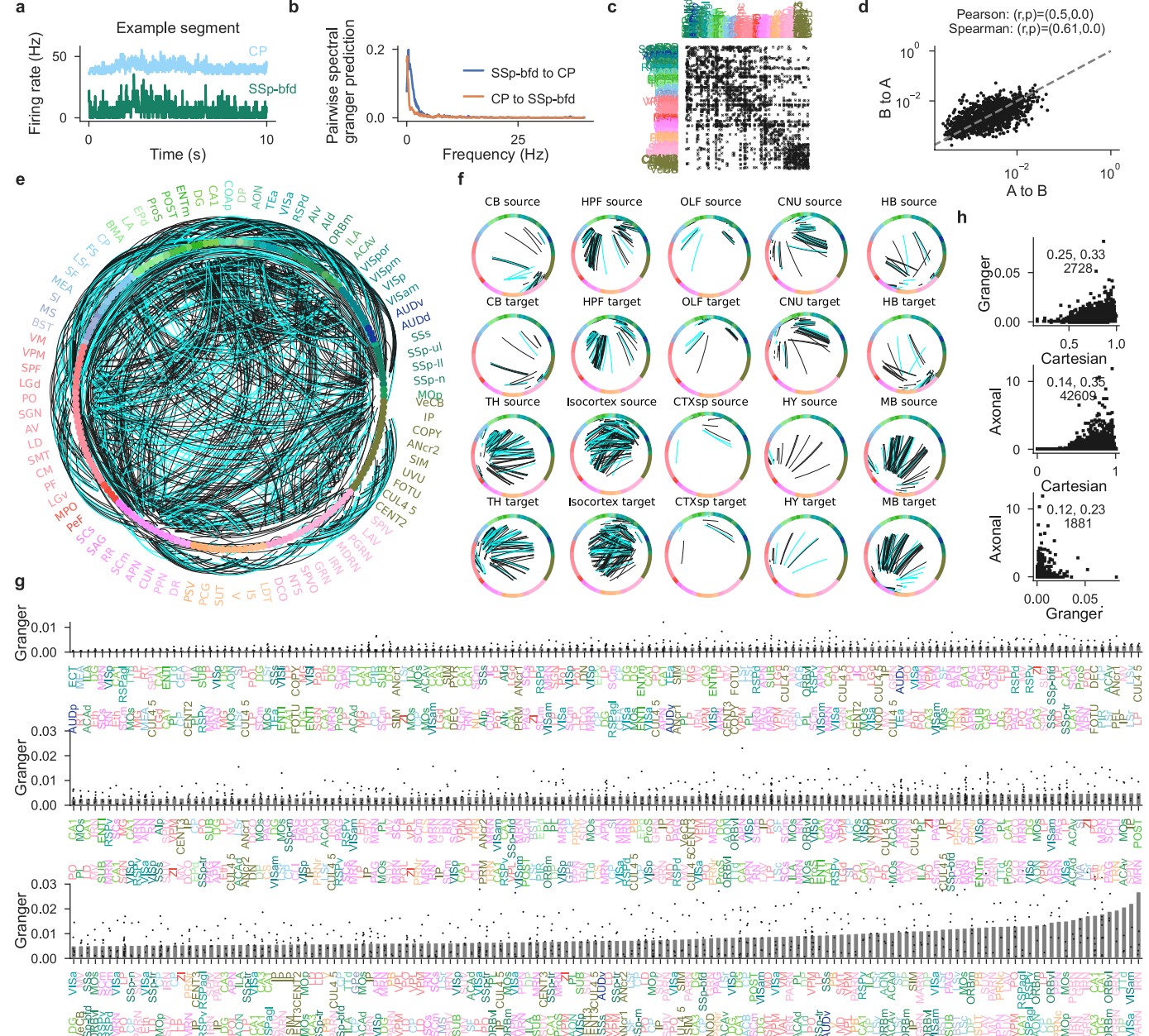

**Extended Data Fig. 4 | Granger scores for simultaneously recorded region pairs. a)** Firing rates in two regions (CP and MOp) for an example session (eid = af55d16f-0e31-4073-bdb5-26da54914aa2); first 10 sec of recording. **b)** Directed spectral Granger prediction for an example region pair from this example session as a function of frequency. This is the average across consecutive 10 sec windows of the whole recording, irrespective of trial-structure. The mean Granger prediction across frequencies is the Granger score, used in all other panels. **c)** Binarised significant Granger score adjacency matrix, canonical region ordering (as in circular graph plot). Note the near-symmetry. **d)** Symmetry of Granger scores for all significant region pairs, log scale. Correlation scores in panel title. **e)** Granger scores for region pairs as averages across recordings, edge width proportional to Granger score, black if

significant. Only region pairs with at least 2 recordings are shown. **f)** Graph of **e)** restricted to incoming/outgoing Granger scores for subsets of regions (Cosmos hierarchical level). **g)** Significant Granger scores for all region pairs, black dots are individual recordings, gray bars are mean across recordings, ordered by mean. Only region pairs with at least 3 recordings are shown. **h)** Granger scores in relation to two other connectivity metrics: axonal (axonal projection tracing, Fig. 3 in[124]) and cartesian (inverse Euclidean distance between centroids of region pairs). Weak but significant correlations (Pearson, Spearman, on top of panels, together with number of directed region pairs for the plot) are found for cartesian/Granger (.25, .33), cartesian/axonal (.14, .35) and Granger/axonal (.12, .23). All results are further listed in this online table.

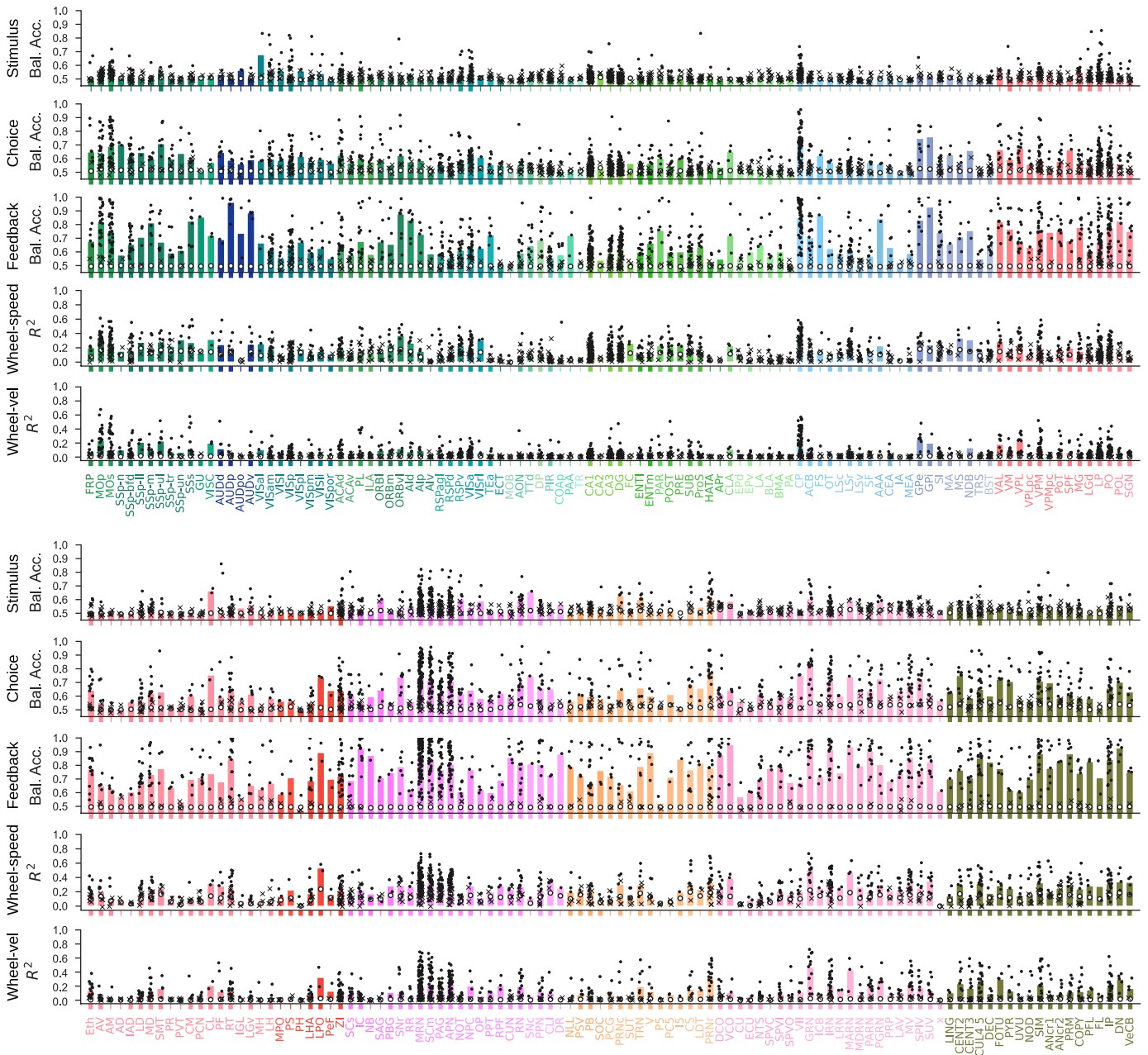

**Extended Data Fig. 5 | Decoding performance per region with per session results.** Decoding analysis as performed for stimulus in Fig. 4, choice in Fig. 5, feedback in Fig. 6, and wheel-speed and wheel-velocity in Fig. 7. No FDR correction has been applied in the bar plots, but the bold ticks indicate those regions that survive $FDR_{0.01}$ (and are shown in the main figures). Black dots and x's indicate decoding performance on individual sessions; dots are significant at $\alpha = 0.05$ and x's are insignificant. The bar height is the median of all sessions within that region, and the white dot is the across-session median of the null distribution medians.

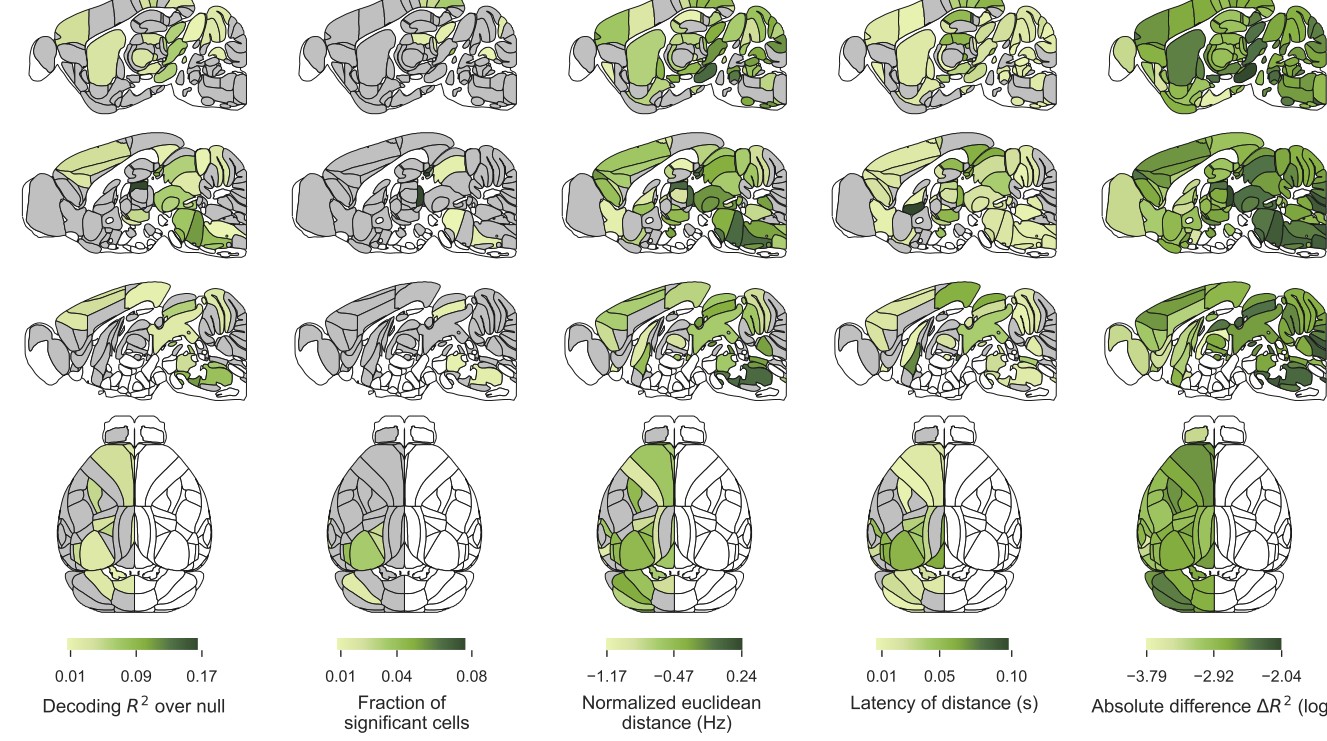

**a** Decoding

Fraction of significant sessions (%)

Zero significant sessions (%)
Not analyzed

NB
TRN
PRNc

**b**

| Decoding | Single-cell statistics | Population trajectory | Population trajectory | Encoding |
|---|---|---|---|---|
| Regularized logistic regression | C.C Mann-Whitney test | Distance between trajectories | Time near peak | General linear model |

Decoding $R^2$ over null
0.01  0.09  0.17

Fraction of significant cells
0.01  0.04  0.08

Normalized euclidean distance (Hz)
−1.17  −0.47  0.24

Latency of distance (s)
0.01  0.05  0.10

Absolute difference $\Delta R^2$ (log)
−3.79  −2.92  −2.04

**Extended Data Fig. 6 | Representation of the stimulus variable. a)** Fraction of sessions with significant decoding performance for the stimulus variable relative to the null. **b)** 2d-brain slices of analysis results for the stimulus variable in Fig. 4a–e. Instead of Swanson flat map, here we use 3 sagittal slices with coordinates ML=−1.8 mm, −0.8mm, −0.2mm, and the top view of the dorsal cortex to visualize the representation of task variables across the brain. The locations of sagittal brain slices are optimised to display 252 brain regions. The region acronyms for these slices are listed in Extended Data Fig. 1.

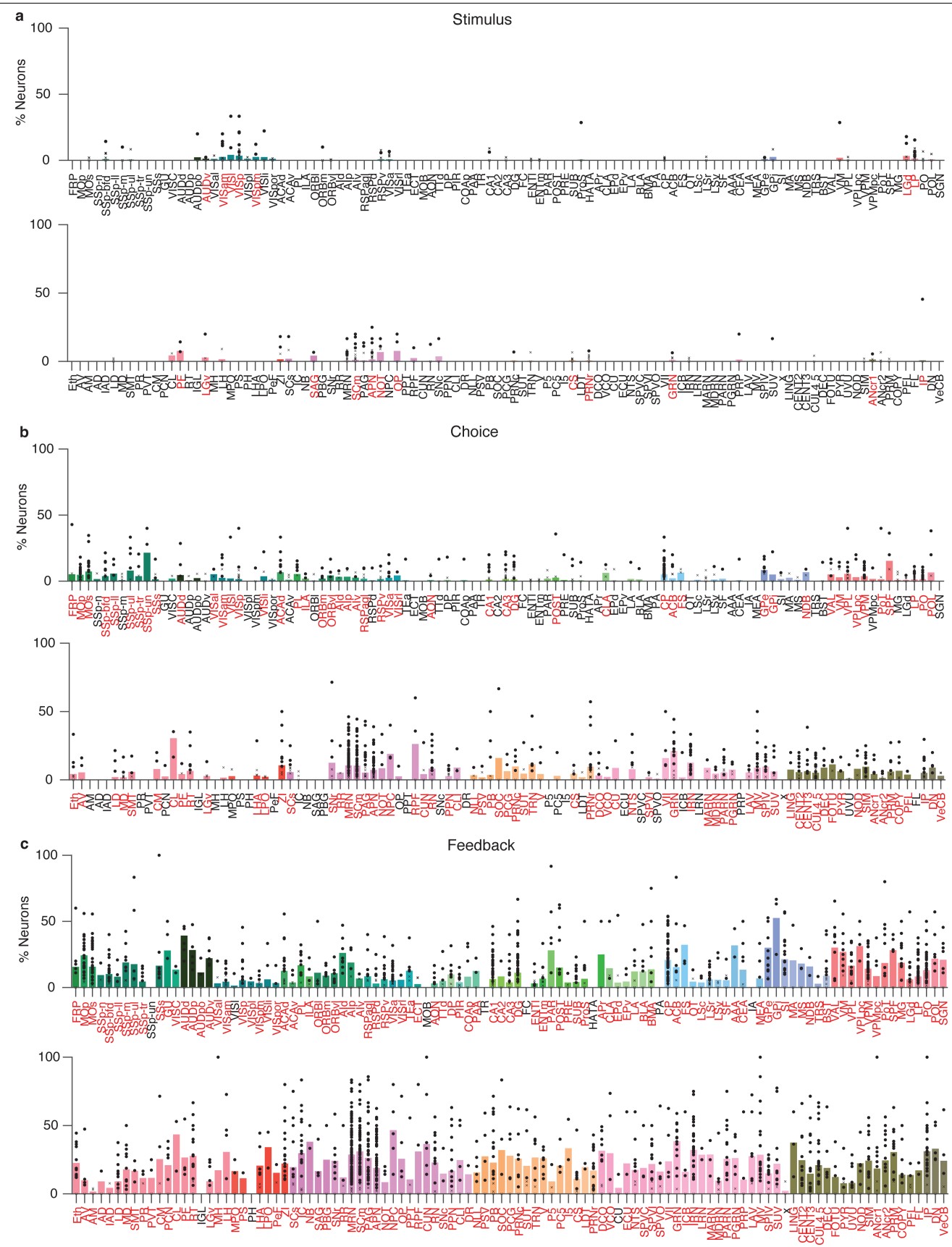

**Extended Data Fig. 7 | Fraction of significant cells per region in single-cell analysis.** Summary of single-cell analysis for stimulus in Fig. 4, **b)** choice in Fig. 5, **c)** feedback in Fig. 6. No FDR correction has been applied in the bar plots; but the red colour labels indicate those regions that survive $FDR_{0.01}$ (and are shown in the figures in the main paper). Black dots and x's indicate single-cell analysis is done on individual sessions where dots are significant at $\alpha = 0.05$ and x's are insignificant. The bar height is the mean of all sessions within that region.

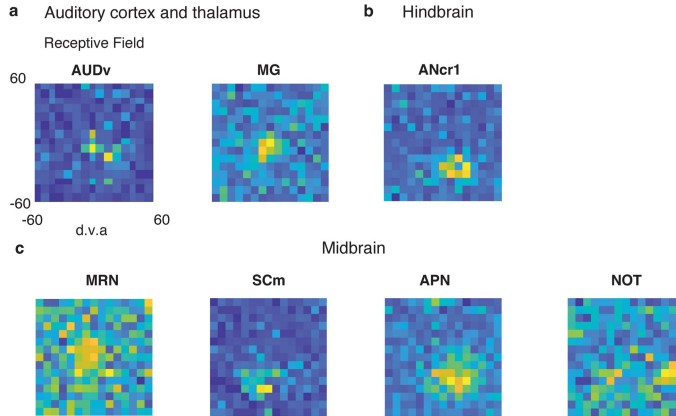

**a** Auditory cortex and thalamus

Receptive Field

**AUDv**　　　　**MG**

60

-60
　-60　　　　60
　　d.v.a

**b** Hindbrain

**ANcr1**

**c** Midbrain

**MRN**　　**SCm**　　**APN**　　**NOT**

**Extended Data Fig. 8 | Example of significant receptive fields of single neurons in auditory areas, hindbrain, and midbrain. a)** Example of receptive fields in auditory cortex (AUDv) and auditory thalamus (MG) (d.v.a. stands for degrees of visual angle). Each pixel in the receptive field denotes 8 × 8 d.v.a. The receptive field is computed by averaging spike rate aligned with On and Off stimulus onset for each pixel, from 0 to 100 ms (Methods). **b)** Example of receptive fields of single neurons in hindbrain. **c)** Example of receptive fields of single neurons in midbrain.

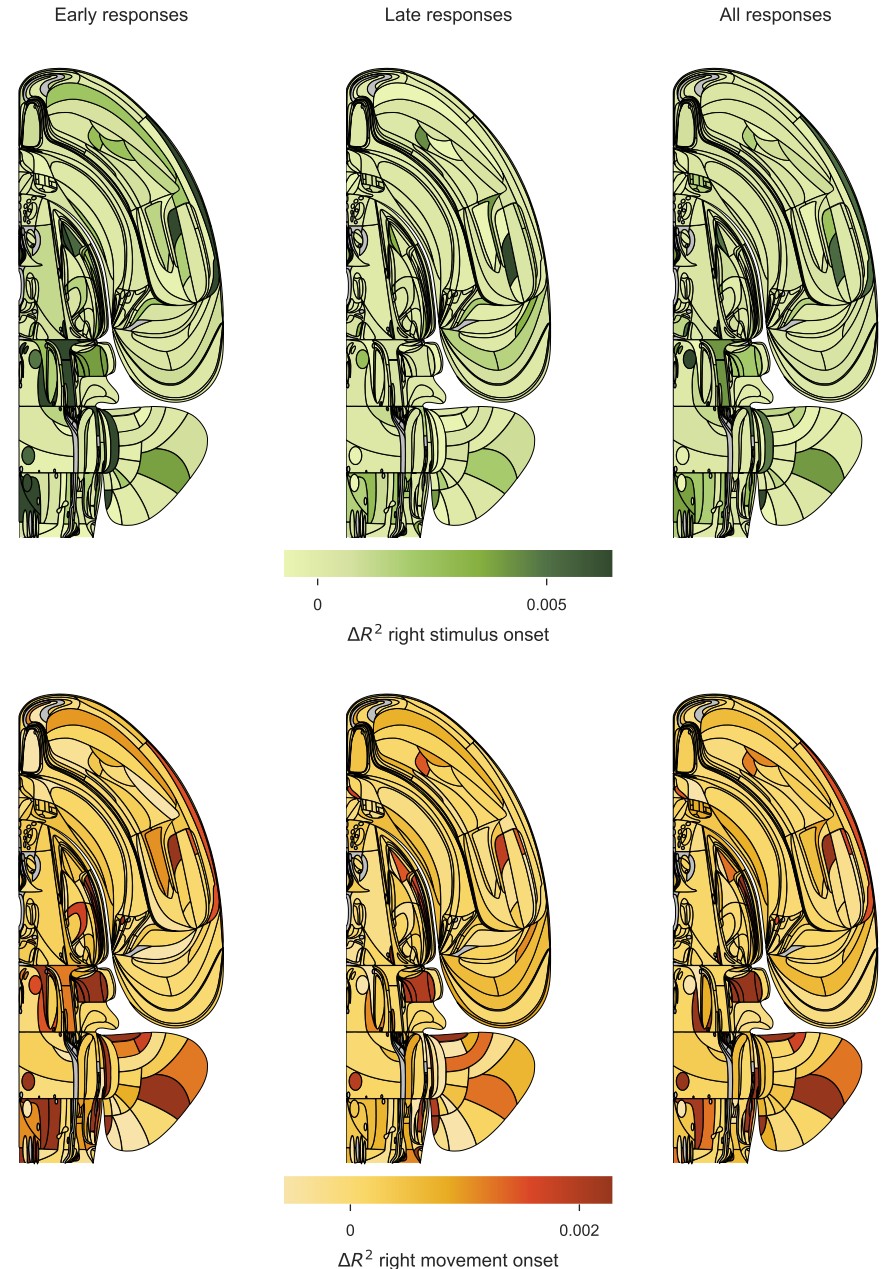

**Extended Data Fig. 9 | Variance explained by stimulus and choice kernels in GLMs fit to early (below median), late (above median), and all RT trials.**
**a)** Mean $\Delta R^2$ from the right stimulus onset kernel per region in trials with response time below median (left), above median (middle), and all trials (right). **b)** Mean $\Delta R^2$ from the right first wheel movement time kernel per region in trials with response time below median (left), above median (middle), and all trials (right).

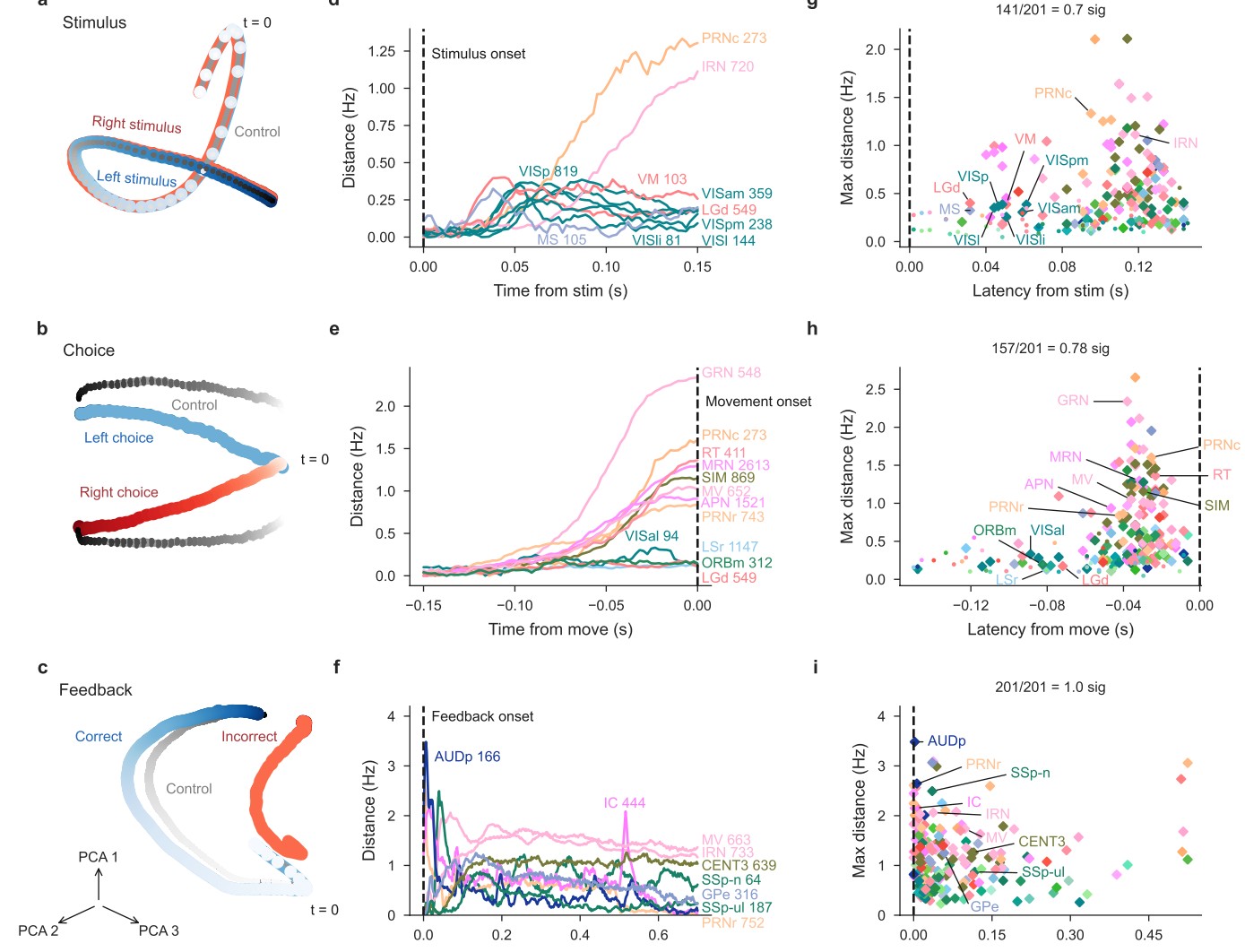

**Extended Data Fig. 10 | Population trajectories across the brain on the full dataset.** Using all well-isolated units and considering regions with at least 20 neurons after pooling across sessions, results in about 446 more neurons (in 9 more regions) than in the canonical set of cells that are used across analyses and shown in the main figures. **a-c)** Visualizations (through low-dimensional PCA-embedding) of whole-brain population dynamics (combined across all cells, all sessions, all regions) for three task variables (left versus right stimulus, left versus right choice, correct versus wrong feedback. Blue/red dots represent one time-bin of the population response for left/right (or correct/wrong) trials; colour gradient indicates temporal evolution (darker is later). Grey dots: pseudo-trials. **d-f)** Quantification of the time-resolved distance between opposite trajectories for each variable, based on Euclidean distance (in spikes/second) in the full-dimensional space (dimension = number of cells) for example brain regions, selected based on response magnitude and to illustrate different response profiles. Curves are annotated by region name and number of cells. Scalebars in all panels represent spikes/s/cell. **g-i)** Summary of variable discriminability for stimulus side, choice side, and feedback type, respectively, by magnitude and latency of response across all recorded brain regions. Diamonds indicate all regions that have statistically significant discrimination ($p < 0.01$ relative to pseudo-trial controls), and line plot examples are labelled by region name. Dots indicate responses of non-significant regions.

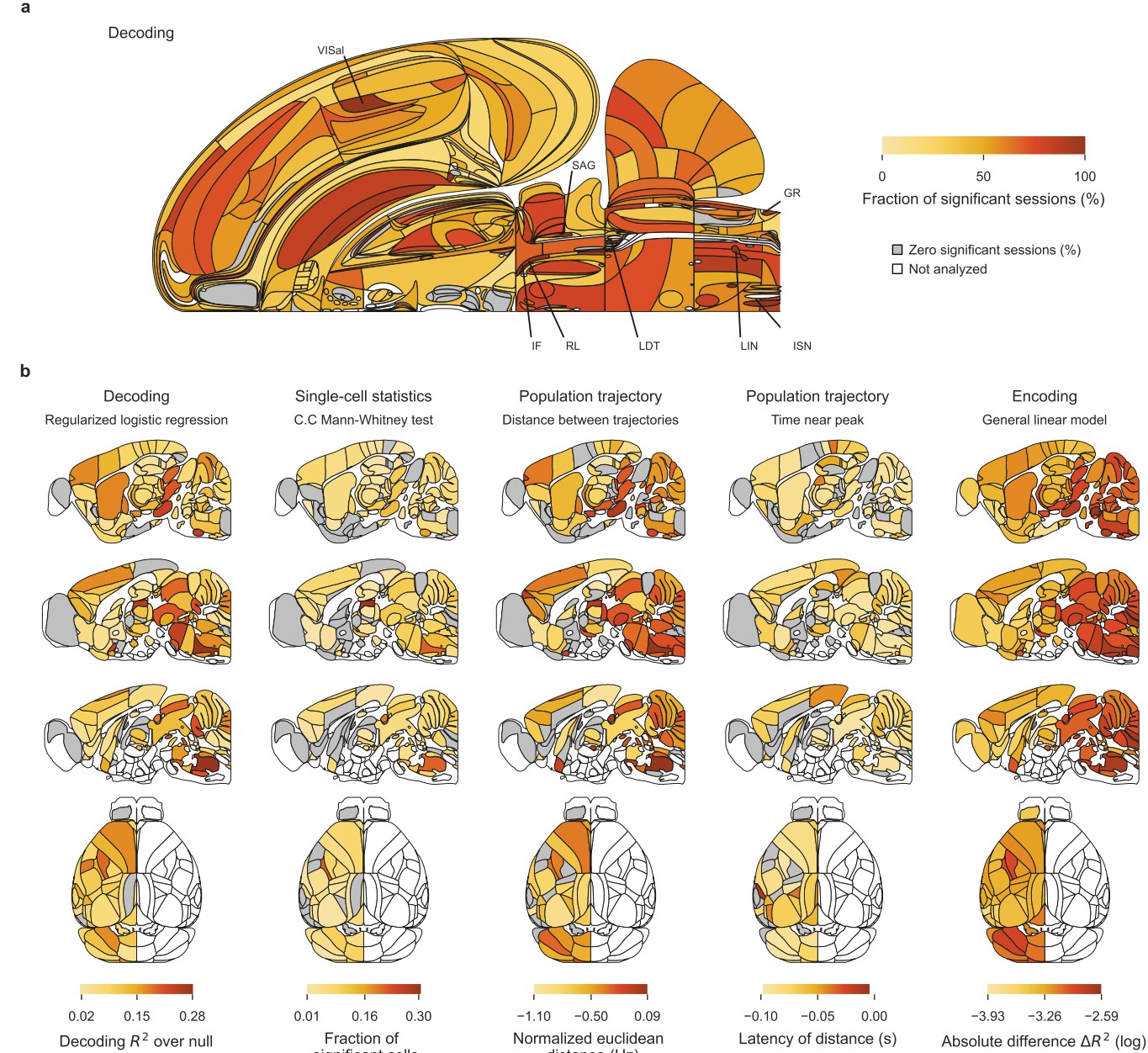

**Extended Data Fig. 11 | Representation of the choice variable.** Analysis of the choice variable, with conventions as in Extended Data Fig. 6.

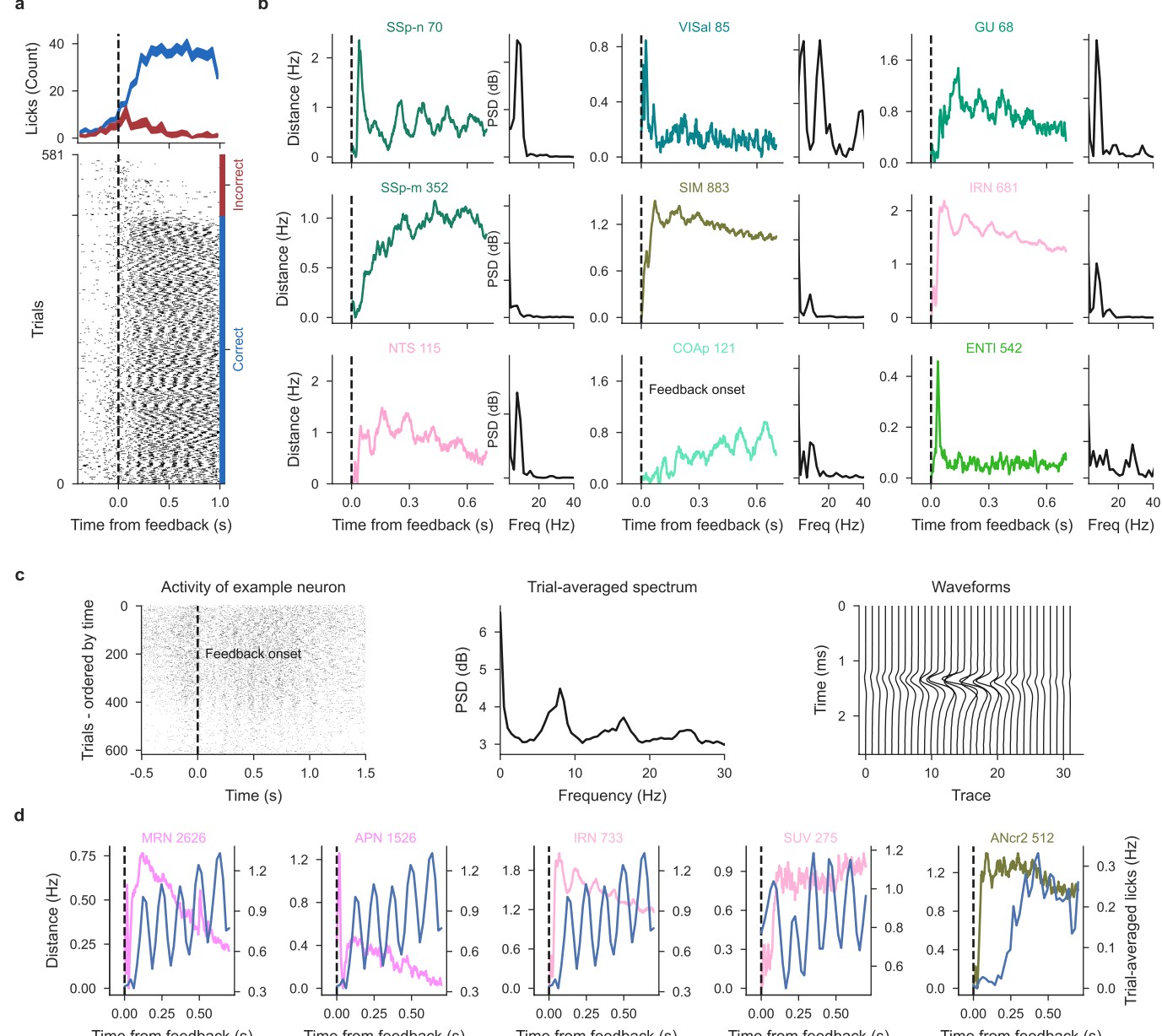

**Extended Data Fig. 12 | Neural correlates of licking. a)** Example lick activity for a single session, top trial-averaged, bottom per trial. Animals lick more for correct trials (blue) with a clear rhythm around 10 Hz. Licks were detected using tongue tracking via DLC from side videos. **b)** Population trajectory distance between correct and incorrect trials for example regions selected manually for visible oscillations, with the number of cells (pooled across sessions) next to the region acronym in the title, aligned to feedback. Right to each panel is the power spectral density of the distance curve, all having a peak around 10 Hz, correlating with licking. **c)** One example neuron's activity (pid = '3b729602-20d5-4be8-a10e-24bde8fc3092', region VPL) to show activity is physiological and not an artefact. Left panel, raster per trial with rhythmic 10 Hz activity, also shown in the middle panel by the power spectral density of

the raster, averaged across trials. Right panel, waveforms of this neuron across adjacent traces, illustrating that the spikes we counted are physiological rather than being caused by an electrical artefact. Artefacts could arise, for example, from current flowing through the drinking spout into the Neuropixels probe, which would result in all traces having a strong waveform. We exclude saturated segments prior to analysis and after this found no evidence for such artefacts when sampling various neurons and inspecting the waveforms. **d)** Single-session population trajectory distance for select regions with trial-averaged lick activity in blue on top. E.g. in MRN a clear correlation with licking was found when restricting the analysis to a single session, while much less so when considering the session-averaged results (not shown).

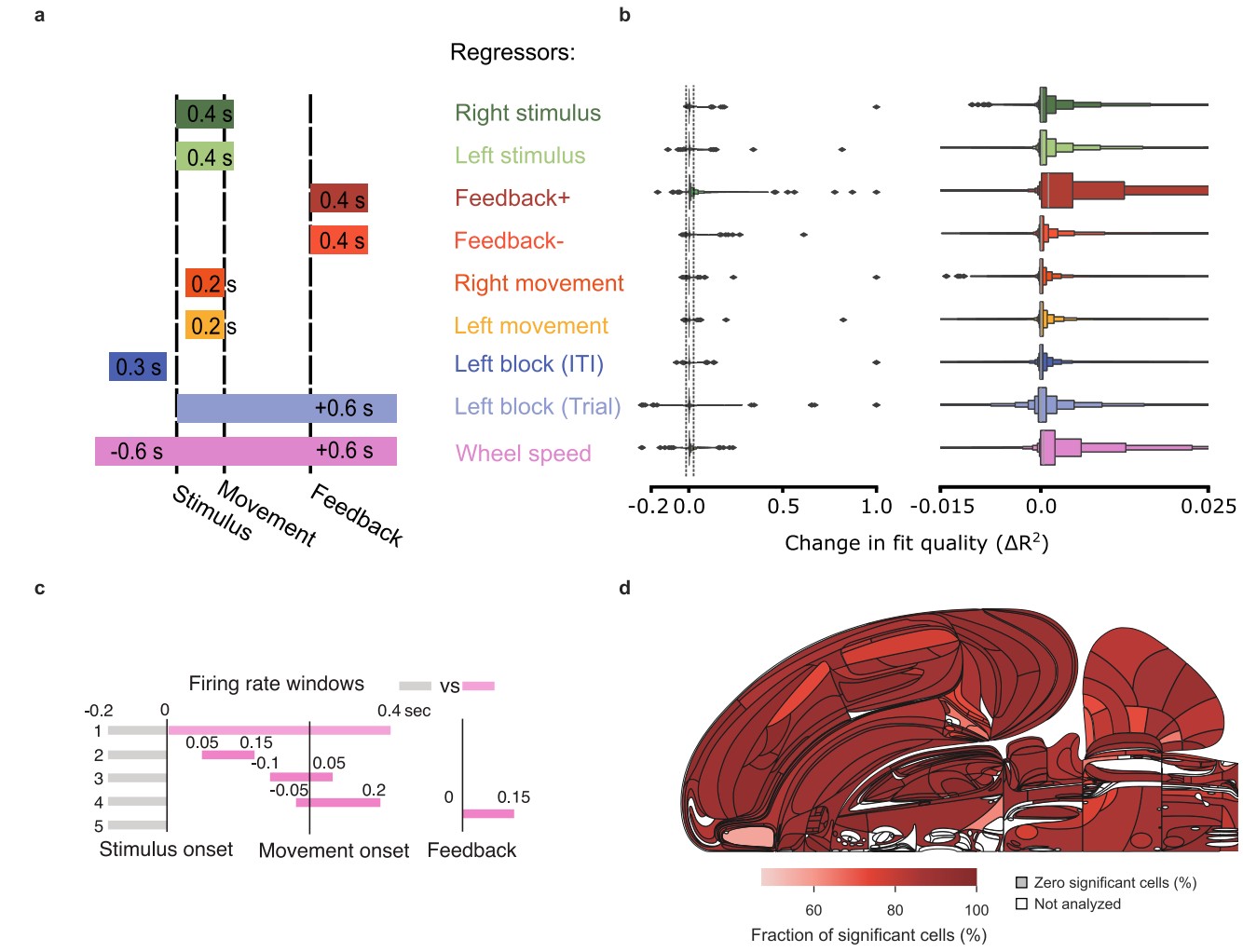

**Extended Data Fig. 13 | Regressor windows and variance explained in linear encoding model and neural correlates of the task across the brain.** **a)** Schematic of within-trial windows in which different regressors in the encoding model apply to firing predictions. **b)** Additional variance explained in a leave-one-out paradigm by each regressor for the full distribution (left) and zoomed-in to the medians of the distributions (right). Note that the range on the right panel is depicted on the left via dotted lines. **c)** Statistical tests to measure responsiveness in different task windows. The schematics show the summary of all tests, superimposed on the task timeline. Each row represents a separate Wilcoxon rank-sum test comparing firing rates in two different periods over which firing rates were estimated. **d)** The flat brain map of the fraction of neurons that show significant task response during at least one of the task epochs (test of responsiveness: **c**), using $FDR_{0.01}$ to correct for multiple comparisons.

**a**

Decoding

NI LDT

BST PVH CS V

Fraction of significant sessions (%)

0　　　　50　　　　100

☐ Zero significant sessions (%)
☐ Not analyzed

**b**

| Decoding | Single-cell statistics | Population trajectory | Population trajectory | Encoding |
|---|---|---|---|---|
| Regularized logistic regression | C.C Mann-Whitney test | Distance between trajectories | Time near peak | General linear model |

Decoding $R^2$ over null
0.02　0.24　0.46

Fraction of significant cells
0.02　0.27　0.52

Normalized euclidean distance (Hz)
−0.76　0.20　1.16

Latency of distance (s)
0.00　0.08　0.17

Absolute difference $\Delta R^2$ (log)
−3.22　−2.29　−1.36

**Extended Data Fig. 14 | Representation of the feedback variable.** Analysis of the feedback variable, with conventions as in Extended Data Fig. 6.

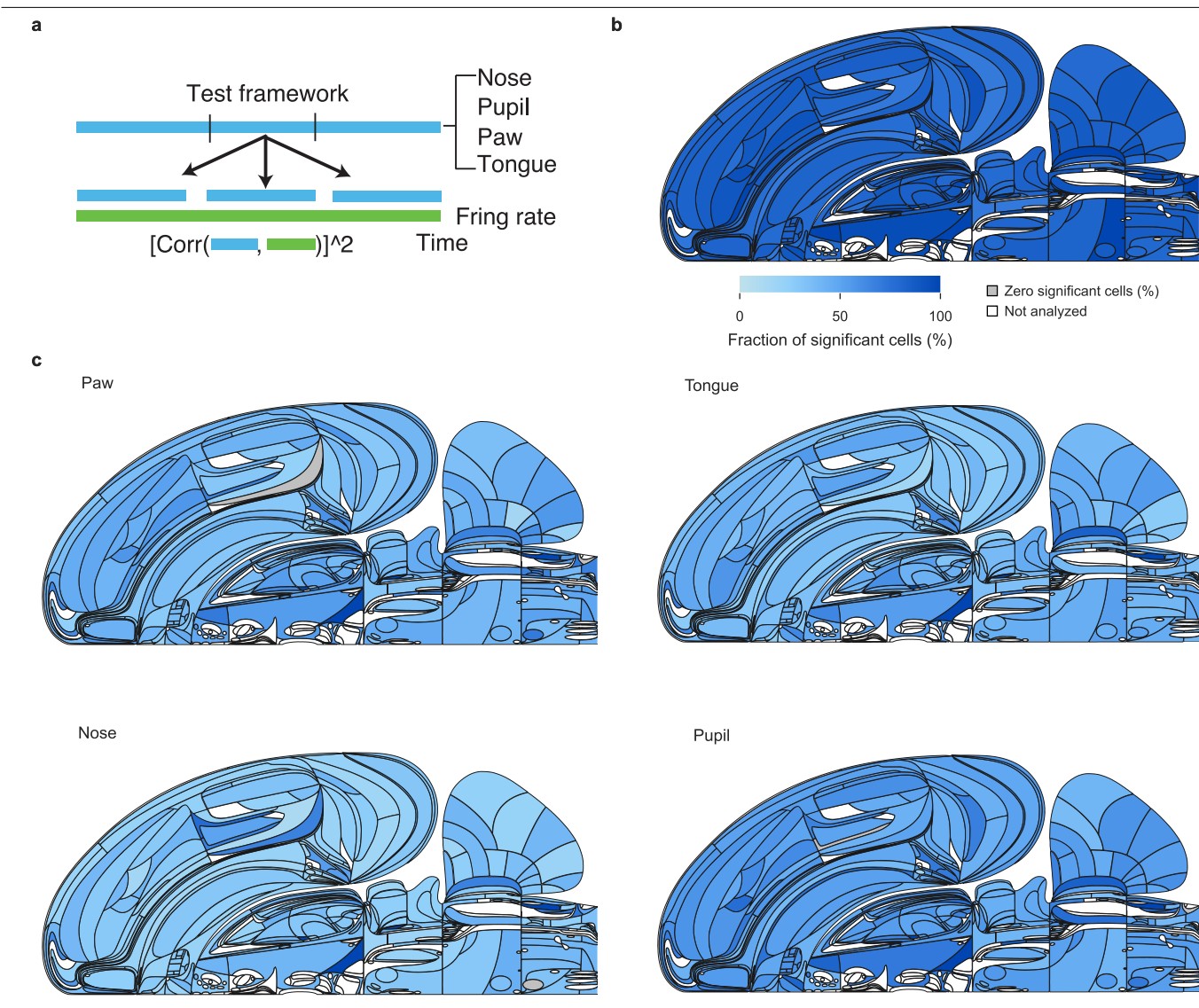

**Extended Data Fig. 15 | The behavioural correlates of single-neuron activity across the brain. a)** Statistical tests to measure the behavioural correlates of single neurons across all sessions. We compute the Pearson correlation coefficient between the time series of neural activity and five behavioural variables (nose position, pupil diameter, paw position, and licks, extracted from behaviour video by using DLC; see Methods). The significance of correlation is estimated by a time-shift test[79] (Methods), using $FDR_{0.01}$ to correct for multiple comparisons. **b)** The flat brain map of the fraction of neurons significantly correlates with at least one of the movement variables. **c)** The flat brain map of the fraction of neurons that significantly correlate with one of the movement variables: nose, pupil, paw, tongue.

# Reporting Summary

## Statistics

For all statistical analyses, confirm that the following items are present in the figure legend, table legend, main text, or Methods section.

| n/a | Confirmed | |
|---|---|---|
| ☐ | ☒ | The exact sample size (*n*) for each experimental group/condition, given as a discrete number and unit of measurement |
| ☒ | ☐ | A statement on whether measurements were taken from distinct samples or whether the same sample was measured repeatedly |
| ☐ | ☒ | The statistical test(s) used AND whether they are one- or two-sided *Only common tests should be described solely by name; describe more complex techniques in the Methods section.* |
| ☐ | ☒ | A description of all covariates tested |
| ☐ | ☒ | A description of any assumptions or corrections, such as tests of normality and adjustment for multiple comparisons |
| ☐ | ☒ | A full description of the statistical parameters including central tendency (e.g. means) or other basic estimates (e.g. regression coefficient) AND variation (e.g. standard deviation) or associated estimates of uncertainty (e.g. confidence intervals) |
| ☐ | ☒ | For null hypothesis testing, the test statistic (e.g. *F*, *t*, *r*) with confidence intervals, effect sizes, degrees of freedom and *P* value noted *Give P values as exact values whenever suitable.* |
| ☒ | ☐ | For Bayesian analysis, information on the choice of priors and Markov chain Monte Carlo settings |
| ☐ | ☒ | For hierarchical and complex designs, identification of the appropriate level for tests and full reporting of outcomes |
| ☐ | ☒ | Estimates of effect sizes (e.g. Cohen's *d*, Pearson's *r*), indicating how they were calculated |

*Our web collection on statistics for biologists contains articles on many of the points above.*

## Software and code

Policy information about availability of computer code

| Data collection | please see http://viz.internationalbrainlab.org and https://int-brain-lab.github.io/iblenv/notebooks_external/data_release_repro_ephys.html |
|---|---|
| Data analysis | all our code is available at https://github.com/int-brain-lab/paper-brain-wide-map |

For manuscripts utilizing custom algorithms or software that are central to the research but not yet described in published literature, software must be made available to editors and reviewers. We strongly encourage code deposition in a community repository (e.g. GitHub). See the Nature Portfolio guidelines for submitting code & software for further information.

## Data

Policy information about availability of data

All manuscripts must include a data availability statement. This statement should provide the following information, where applicable:
- Accession codes, unique identifiers, or web links for publicly available datasets
- A description of any restrictions on data availability
- For clinical datasets or third party data, please ensure that the statement adheres to our policy

all the data for this paper are available via http://viz.internationalbrainlab.org and https://int-brain-lab.github.io/iblenv/notebooks_external/data_release_repro_ephys.html

April 2023

# Research involving human participants, their data, or biological material

Policy information about studies with [human participants or human data](). See also policy information about [sex, gender (identity/presentation), and sexual orientation]() and [race, ethnicity and racism]().

| | |
|---|---|
| Reporting on sex and gender | *Use the terms sex (biological attribute) and gender (shaped by social and cultural circumstances) carefully in order to avoid confusing both terms. Indicate if findings apply to only one sex or gender; describe whether sex and gender were considered in study design; whether sex and/or gender was determined based on self-reporting or assigned and methods used. Provide in the source data disaggregated sex and gender data, where this information has been collected, and if consent has been obtained for sharing of individual-level data; provide overall numbers in this Reporting Summary.  Please state if this information has not been collected. Report sex- and gender-based analyses where performed, justify reasons for lack of sex- and gender-based analysis.* |
| Reporting on race, ethnicity, or other socially relevant groupings | *Please specify the socially constructed or socially relevant categorization variable(s) used in your manuscript and explain why they were used. Please note that such variables should not be used as proxies for other socially constructed/relevant variables (for example, race or ethnicity should not be used as a proxy for socioeconomic status). Provide clear definitions of the relevant terms used, how they were provided (by the participants/respondents, the researchers, or third parties), and the method(s) used to classify people into the different categories (e.g. self-report, census or administrative data, social media data, etc.) Please provide details about how you controlled for confounding variables in your analyses.* |
| Population characteristics | *Describe the covariate-relevant population characteristics of the human research participants (e.g. age, genotypic information, past and current diagnosis and treatment categories). If you filled out the behavioural & social sciences study design questions and have nothing to add here, write "See above."* |
| Recruitment | *Describe how participants were recruited. Outline any potential self-selection bias or other biases that may be present and how these are likely to impact results.* |
| Ethics oversight | *Identify the organization(s) that approved the study protocol.* |

Note that full information on the approval of the study protocol must also be provided in the manuscript.

# Field-specific reporting

Please select the one below that is the best fit for your research. If you are not sure, read the appropriate sections before making your selection.

☒ Life sciences ☐ Behavioural & social sciences ☐ Ecological, evolutionary & environmental sciences

For a reference copy of the document with all sections, see [nature.com/documents/nr-reporting-summary-flat.pdf]()

# Life sciences study design

All studies must disclose on these points even when the disclosure is negative.

| | |
|---|---|
| Sample size | the recording strategy is discussed extensively in the methods section for the paper |
| Data exclusions | the Methods contains an explicit statement on data exclusions |
| Replication | reproducibility for this project is discussed in: https://elifesciences.org/articles/63711 and https://www.biorxiv.org/content/biorxiv/early/2023/05/18/2022.05.09.491042.full.pdf |
| Randomization | permutation strategies are discussed in detail in the Methods |
| Blinding | blinding was not possible in this study |

# Reporting for specific materials, systems and methods

We require information from authors about some types of materials, experimental systems and methods used in many studies. Here, indicate whether each material, system or method listed is relevant to your study. If you are not sure if a list item applies to your research, read the appropriate section before selecting a response.

## Materials & experimental systems

| n/a | Involved in the study |
|-----|----------------------|
| ☒ | ☐ Antibodies |
| ☒ | ☐ Eukaryotic cell lines |
| ☒ | ☐ Palaeontology and archaeology |
| ☐ | ☒ Animals and other organisms |
| ☒ | ☐ Clinical data |
| ☒ | ☐ Dual use research of concern |
| ☒ | ☐ Plants |

## Methods

| n/a | Involved in the study |
|-----|----------------------|
| ☒ | ☐ ChIP-seq |
| ☒ | ☐ Flow cytometry |
| ☒ | ☐ MRI-based neuroimaging |

# Animals and other research organisms

Policy information about studies involving animals; ARRIVE guidelines recommended for reporting animal research, and Sex and Gender in Research

| Laboratory animals | we used C57BL/6 laboratory mice |
|---|---|
| Wild animals | n/a |
| Reporting on sex | both sexes were used; and are reported |
| Field-collected samples | n/a |
| Ethics oversight | All procedures and experiments were carried out in accordance with the local laws and following approval by the relevant institutions: the Animal Welfare Ethical Review Body of University College London; the Institutional Animal Care and Use Committees of Cold Spring Harbor Laboratory, Princeton University, University of Washington, University of California at Berkeley and University of California at Los Angeles; the University Animal Welfare Committee of New York University; and the Portuguese Veterinary General Board. |

Note that full information on the approval of the study protocol must also be provided in the manuscript.

