## [Peer Review file · Nature]

A Brain-Wide Map of Neural Activity during Complex Behaviour

Corresponding Author: Professor Peter Dayan

Version 0:

Reviewer comments:

Referee #1

(Remarks to the Author)

This study offers an initial assessment of a brain-wide activity map based on neural recordings gathered from multiple laboratories in mice engaged in a decision-making task encompassing sensory, motor, and cognitive elements. The neurophysiological data are accompanied by a systematic assessment of behavioural readouts, including the mouse's movement (speed and velocity) on the running wheel, pupil size, and whisker activity. The study analyses the neural responses of over 30 thousand sorted single units in the context of a solid and well-established behavioural paradigm (e.g., Steinmetz, Nature 2019) that includes cognitive, sensory and motor aspects. The manuscript describes an impressive dataset that the IBL has publicly released, explaining recording and analytical strategies. It focuses on the neural representations of the visual stimulus, behavioural choice, feedback, choice/stimulus expectation and wheel movement. Except for object prior expectations (- but see the comments below), the neural correlates of these variables were widespread in the brain.

The study is particularly significant because it represents much more than a simple manuscript, but rather the result of a network of labs that pooled and analyzed hundreds of experiments, producing a publicly available dataset highly valuable to the global community. This effort resulted in an outstanding brain-wide neural activity map. The dataset is highly relevant because, when fully shared and detailed, it will have the potential to inspire hypothesis-based experiments targeting specific subsets of the brain regions analyzed in this manuscript.

The data, methods, and statistics are solid and well-presented. The manuscript highlights the widespread brain representations of visual stimuli, motor actions, choices, and reward delivery/consumption, supporting globally distributed information coding. This study capitalizes on a large body of work previously done by the IBL team that has carefully established reproducible and standardized measurements of behavioural and electrophysiological recordings. The abstract, introduction and conclusions are well-written and clearly explained. However, some comments below require attention in order to increase the clarity and interpretability of the main findings and datasets.

Figure 1 and its legend are missing. Please include them in the manuscript.

The recordings described in this study were collected by 11 labs across different countries and continents. The IBL has done significant work, over the past years, ensuring the reproducibility of its results across research facilities (e.g., IBL, Elife, 2021). However, for this specific study, it would be informative to indicate the actual learning rates and performances, across laboratories, for the 115 mice included in this manuscript.

The behavioural task described in this study is, at its core, a visual task, and a high number of recordings were acquired from visual areas. For the neurons recorded from these visual areas, were any visual receptive field mapping experiments performed? Were the receptive field properties of each neuron considered in any of the presented analyses? Besides the pupil size, were the X-Y eye positions taken into account?

In Figure 5 (Representation of the Visual Stimulus), Panel H compares peri-event time histograms (PETHs) of spiking activity for left and right stimuli for an example neuron. Why are the PETHs from the top panel different from the ones on the bottom panel? (The difference is evident in the red standard errors of the means.)

Different animals, across different recording sessions, will inevitably have their own unique behavioural strategies and neural responses. From the results/methods of this manuscript, it appears that most recordings were done with multiple simultaneous probe insertions. These data are particularly insightful for future studies because they can truly provide information on how simultaneously recorded brain regions responded during the task on a trial-by-trial basis. It would be helpful to indicate whether the simultaneous recordings are clearly identified on the publicly released data set. It would be useful to show a table illustrating the number and identity of areas simultaneously recorded in at least 2 or 3 mice. So that future studies (or even this one, if feasible, on a small sample of variables) could restrict the analysis of the neural correlates of the main task/variable events to only simultaneously recorded areas (and, for example, verify if they confirm the whole population data)

One of the conclusions of the abstract is: "Representations of objective prior expectations were weaker, found in sparse sets of neurons from restricted regions." This conclusion is in contrast with the main finding of a related paper released by the same group on bioRxiv (Findling, IBL, "Brain-wide representations of prior information in mouse decision-making": e.g. "This widespread representation of the prior is consistent with a neural model of Bayesian inference involving loops between areas, as opposed to a model in which the prior is incorporated only in decision making areas"). While the reasons for this discrepancy are well explained in both the results and discussion sections of this manuscript, the abstract conclusions could be misleading, and the sentence reported above should be either rephrased or explained in more detail for consistency.

In the Methods, "Neurons and Brains Region" paragraph: "amplitude >50 μ V ; noise cut-off < 20 μ V; refractory period violation." How long is the refractory period time window (in ms), and, for a "good unit", what is the percentage of accepted refractory period violations over the total number of spikes?

In Figure 4, panel C, should the colour code (orange/amber) of the movement onset in the graph and the temporal kernel functions be the same?

Sincerely,

Riccardo Beltramo

Referee #2

(Remarks to the Author)

In this pair of companion studies, the authors (members of an international consortium for experimental neuroscience called the International Brain Lab; IBL) present an unprecedented dataset, collected across 11 labs, comprised of 547 high-density neural recordings sampled across the entire mouse brain during a standardized decision-making task. In one study, the dataset (curated and now made publicly available) and methodology are described, and then used to perform a battery of correlational analyses across the brain to identify encoding of simple task variables during decision-making (including sensory, choice, outcome, movement, and prior information). Based on this investigation and application of rigorous statistical methods, the authors conclude that different task variables are encoded in different ways (some are represented more widely across the brain, while others are more localized), confirming the results of countless previous reports. In the second study, the authors investigate more deeply the encoding of prior information using a more refined set of analytical tools and combining their electrophysiology dataset with another large calcium imaging dataset collected by IBL participants. The main conclusion of this second study is that, in contrast to the initial conclusion of the first study, which suggested that prior information is encoded in a restricted set of brain regions, subjective priors are actually encoded more widely across cortical, subcortical, and midbrain regions. Further, this more widespread subjective prior is partially embodied in the animal's behavior and is driven by action history (as opposed to sensory and/or outcome history exclusively).

The dataset collected here, and particularly the effort to generate a dataset of such comprehensive scale with rigorous control and standardization of task conditions, preprocessing parameters, etc. is obviously noteworthy and is inarguably a highly unique and important contribution to the field. The presented impetus for this enormous collaborative effort is that our capacity to really gain insight into how the brain processes sensory information, computes decisions, and generates behavioral outputs (i.e. how the brain works) is substantially hindered by the limitations of individual lab groups to survey brain structures comprehensively under consistent conditions, and the idiosyncrasies with which different groups design and implement their experiments that limit our capacity to draw accurate comparisons and generalizations. The assumption underlying and framing the effort is that, with a big enough dataset, collected with careful methodological standardization, we should be able to mine the data to gain novel insight into how the brain works. It is, in some sense, the extreme terminus of "data-driven" approaches.

Somewhat unfortunately, the studies, even taken together, fall woefully short of this laudable goal. In the first study, there are

in many cases no explicit hypotheses given to ground the questions the authors ask of the data, or if there are, they are so vague as to be almost useless (e.g. visual information may be encoded to some degree beyond classically defined visual areas). We are presented with what amounts to a brain-wide “screen” of task variable neural correlates, and learn, in the end, essentially nothing new about how the brain works – visual stimuli are predominantly represented in visual cortex and thalamus; movement correlates are widespread across the brain, as are responses to salient rewards and punishment. And, at least in this particular case, this is doubly a problem because the specific framing of the knowledge gap the work is intended to fill is itself defeated – the need for this type of approach totally obviated – by the apparent lack of novel insight. If recording from 300,000 neurons sampled from the entire mouse brain during a well-designed decision-making task does not yield any novel insight, or change our understanding of the brain in any substantial way, then what is the value of such an approach at all?

In the second companion paper, the authors go a bit further. They are able to draw some interesting, useful conclusions from the work that might have been difficult (but this is not totally clear) to fully capture with more limited, fragmented efforts. In particular, they show that while “objective” prior information is only rarely encoded in restricted brain regions (in the first paper), “subjective” prior information is, in fact, much more robustly encoded, and across several levels of brain structure – and this prior takes the form of a partially-embodied action prior, as opposed to other formulations. In the context of value-based decision, this is indeed an important finding, but in the specific context of these two studies, it is still limited in scope for at least two reasons: First, it somewhat contradicts the conclusions of their first study – the authors make a big deal in study 1 about how the restricted nature of prior encoding is interesting because it stands in contrast to how the other task variables are encoded, but then proceed to explain this difference by arguing that this is probably just because animals do not have access to an objective prior, and in fact, encode subjective priors quite robustly. If this is the case, why not just present the analysis of subjective priors alongside other task variables in the first study? Second, it is not clear to what extent the positive conclusions they are able to draw about subjective priors really depend on the nature of the dataset they have compiled. Could this have not been discovered with much more limited methodology?

On balance, while this work is definitely noteworthy even just by virtue of the unprecedented scale of the accompanying dataset and its public availability, in order to merit publication in *Nature*, we would need to be shown more convincingly that such an approach has the potential to expand our understanding of how the brain works. Because as it stands right now, the biggest conclusion that one walks away from reading these papers is, ironically, that generating and then investigating enormous datasets with correlative analytics and without grounding hypotheses is a fairly impotent approach that is likely not worth the effort.

Fortunately, I think this is not actually the case, and there are probably countless ways that the authors could spend a bit more time with this dataset to make some genuine, likely impactful discoveries. Without going deep into specifics, I will provide a few ideas for avenues that the authors could pursue to achieve this (avenues which are likely already being explored).

1) The dataset contains neurophysiological and behavioral measures from 115 mice performing the exact same, standardized task. Presumably then, there is some degree of individual variability in task performance (objective as well as idiosyncratic variability in task strategy). This seems like a perfect opportunity to investigate, even in a coarse manner, how such differences in behavioral variability may be related to underlying differences in neural dynamics and mechanisms. For example, are there specific brain regions, or clusters of regions that, beyond encoding task variables to whatever degree, are highly predictive of task performance, or are correlated with the extent to which animals use a particular cognitive strategy (such as how much they depend on estimates of priors?).

2) While restricted because of the geometry of single probe insertions, the authors have access to a wealth of information about simultaneous neural dynamics recorded across different brain regions (at least those that are dorsoventral neighbors). Despite the repeated mention of “loops” and “widespread, distributed activity,” there is astonishingly no effort here to leverage the dataset to understand anything about inter-areal communication during decision-making. It would be nice to get at least some insight into this knowledge gap that the authors allude to, certainly have the capacity to address, but then say nothing substantial about.

3) Given the unprecedented nature of the dataset, it would also be interesting to see some quantitative assessment of how useful enormous amounts of neurophysiological data really are. The authors are in a unique position to address this. For example, how strongly do conclusions about task variable coding maps hold as a function of the number of neurons recorded, the number of sessions, mice, etc.? If it turns out to be the case that most of the main conclusions could be drawn with much more restricted coverage, i.e. that there are dramatic diminishing returns on scaling up the sheer quantity of neurophysiological data, then this would seem to be an important conclusion to spell out so that future efforts can be better directed toward more fruitful avenues.

Finally, there is a potential problem in the task design. In most trials, whether animals are rewarded or not is determined by the correctness of the choice based on the sensory cue. However, in the 0% contrast trials, where there is no “correct” choice, the probabilities of reward on the left and right sides were set to 20% or 80% according to the prior probability of left and right stimuli. This means that, in principle, mice can use these differential reward probabilities in 0% contrast trials to bias their choice. In other words, the bias in reward probability during 0% contrast trials, rather than the prior probabilities of left and right trials, could have contributed to or been the main cause of the choice bias. It is important to exclude this possibility to support the overall conclusions.

Overall, my sense is that this work could ultimately be impactful and unique enough to merit publication in Nature, but definitely not in its current form. The authors must do a better job of convincing us that this approach is worthwhile in terms of real intellectual and scientific return – the dataset alone, while obviously impressive (and exciting), is not enough.

Referee #3

(Remarks to the Author)

A. Key results:

The paper “A Brain-Wide Map of Neural Activity during Complex Behaviour” presents a brain-wide analysis of neural activity during a standardized visuomotor decision-making task. By pooling large-scale electrophysiological data acquired by multiple laboratories and ensuring high reproducibility, the activity of a large number of neurons covering most brain regions was acquired, allowing analysis of the decision-making task at an unprecedented coverage and at high temporal resolution. The authors used strict quality control metrics and standardized analysis pipelines to uncover the brain regions that contain neural activity linked to different aspects of the decision-making process, such as the visual stimulus, the choice, the feedback, the objective prior, and the wheel movement. The data shows widespread engagement of brain regions by the wheel movement and the feedback (reward and noise/time-out), a large but smaller set of regions driven by choice, and a sparse set of regions encoding the visual stimulus or block prior. An important finding is that visual stimuli appear encoded first in classical visual areas and later in a collection of mid and hindbrain regions that also strongly encode choices. Also, interestingly, choice encoding was more substantial in these subcortical regions than in classically described cortical regions. As previously reported, movement (uninstructed or instructed, in that case, wheel movement) seems to be encoded throughout the brain. One should interpret the regions encoding feedback cautiously because the task design does not allow decorrelating reward from actions linked to reward. Similarly, results on the block prior are unclear as the authors used the objective prior unknown to the mouse; a companion paper addressed this issue. Beyond the results, it should be noted that this paper presents a new, unprecedented resource: a large publicly available dataset easily searchable and useable by other neuroscientists.

B. Originality and significance: This paper by the International Brain Laboratory (IBL) represents a unique collaborative effort, the first of this type and scale in systems neuroscience, at least to the reviewers' knowledge. Therefore, this paper is one of a kind and hopefully the start of a paradigm shift in systems neuroscience. The high significance of such effort comes from the fact that neural correlates of behavior are distributed throughout the brain. Small-scale studies focusing on a predefined set of brain regions have taught us a lot and have value, but they do not capture the big picture of where different variables are encoded in the brain. The only way to answer this question is through large-scale, reproducible studies like the one presented in this paper. The dataset is of high quality and provides unprecedented spatial coverage of single-neuron activity at high temporal resolution. Moreover, the provided data and the technical information accompanying it are easily accessible, searchable, and useable for other researchers, meaning that more discoveries linked to this data may come. Of course, the present results are not necessarily generalizable and are specific to one task. Nonetheless, these results provide clear answers about the regions involved in such a task. For example, this unbiased screen demonstrates that the representation of visual stimulus and choice is prominent in mid- and hindbrain structures for this type of decision-making task, which has been overlooked in the literature – often “cortico-centric”. This result paves the way for further studies to investigate whether it holds for more complex tasks and how the encoding arises through learning. In summary, this paper is highly original and significant and will significantly impact neuroscience.

C. Data & Methodology: The manuscript is well-written and well-structured. The methods are detailed and provide all the information for successful reproduction. The data is publicly available, and an intuitive API allows easy data browsing.

D. Appropriate use of statistics and treatment of uncertainties: The authors use appropriate statistical methods and carefully discuss potential uncertainties. The authors took great care to choose appropriate null distributions whenever possible. Statistics are of the utmost importance for this paper, which pools data from different sessions, mice, and laboratories. Method sections give ample details, which is highly appreciated.

E. Conclusions: robustness, validity, reliability. As the data was collected in multiple laboratories (each recording location was performed in at least two different laboratories) and the reproducibility was carefully investigated in another publication (IBL 2022), the authors ensured very high robustness and reliability of the described findings. The overall conclusion that neuronal activity during many aspects of the task is widespread throughout the brain is well supported.

F. Suggested improvements: Overall, we do not see significant flaws in the manuscript, but we would like to suggest improvements. We understand the difficulty of analyzing brain-wide data and the beauty in using the same analysis pipeline throughout the paper to present comparable brain-wide maps across the different aspects of the task (vision, feedback, movement, etc.). However, in some places, it seems that the chosen analysis is better suited for certain time periods and variables (i.e. choice) than for others (i.e. visual stimulation or prior encoding) - see points below - and that additional analysis could help extract additional information from this valuable dataset.

G/H References, Clarity and context: lucidity of abstract/summary, appropriateness of abstract, introduction and conclusions : We found the paper very factual, not overselling, and the presentation of the context appropriate.

Specific comments:

1. Major point: We find the section on the representation of prior more confusing than helpful. As the authors point out, the variable studied here - the true block prior – is not accessible to the mouse. The results show very few regions encoding that variable. The companion paper addresses that issue and finds that the subjective block prior is actually encoded in a large number of regions. Why keeping that section? Either readers have not read the other paper and will be confused, thinking that the prior is sparsely encoded, or they have read the other paper and will think that this part is conceptually wrong. We suggest to remove it and point towards the other paper in the discussion instead.

2. Comments on the representation of the visual stimulus:

- a. How do you interpret the large difference between the single-cell analysis and the decoding/manifold analyses for that variable? This is puzzling compared to other variables. What do the authors conclude about these midbrain/hindbrain regions when considering all analyses? Do they encode the stimulus or not? A clear statement – even if it is a speculation given the evidence provided – would be useful. Generally, the conclusion of this part is vague, uninformative for the reader and could be improved.
- b. The latency information extracted from the manifold analysis could be supplemented with more classical analysis of the latency directly from the PSTH.
- c. What about the latency in the superior layers of the superior colliculus, which also receive direct input from the retina? The authors only comment on visual and thalamic areas. If the latency is also high in SCs, it would contradict the argument made by the authors about the “classical areas” responding fast.
- d. Could we see some example neurons/PSTH from these hindbrain regions locked on the stimulus?

3. Comments on the representation of choice:

- a. In figure S2, one can clearly see how the average PSTH per brain region for visual stimulation and choice are shifted version of each other. This is not surprising as the mice have fast reaction times, however it would be interesting if the authors can elaborate more on this and potentially provide additional analysis in Figure 6, i.e. how the choice map changes when selecting trials with short versus long reaction times?
- b. Similarly, looking at the example unit in GRN as well as the average PSTHs in Figure S2 one wonders how the signal develops beyond the 50ms after movement onset shown. If possible, the authors could extend these time windows allowing a better understanding of the temporal dynamics of the ramping up regions.

4. Comments on the representation of wheel movement:

- a. In figure 9 it does not fully become clear why the authors present the analysis for both speed and velocity. Not surprisingly, the results are extremely similar. The velocity could be moved to supplementary material. Analogous analysis of other DLC outputs from the nose, paw etc. could be more interesting in the main figure
- b. Why did the authors choose the linear-shift method rather than “imposter sessions” (taking the behavioral data from other trials/sessions) in the case of these behavioral variables? Could the authors spell that out for the reader?

5. Additional comments:

- a. Do the authors have any data in neuromodulatory regions (dorsal raphe, locus coeruleus, basal forebrain, ventral tegmental area) and could they analyze this data to evaluate the impact of global neuromodulatory signals across the brain during this task? If not, maybe it would be worth discussing about targeting such regions, as the authors speculate on the role of such signals several times in the paper.
- b. In the paragraph of the discussion about the brain coverage (page 31), it would be interesting to know what type of brain regions are systematically missed (i.e. nuclei below a certain size? lateral regions?).

Minor points:

1. Figure 1 and one panel of Figure 5 (j) disappeared from the provided pdf. We had to use the preprint version of the paper to assess these parts.
2. Throughout the paper when the results of the encoding model for a single unit are presented (i.e. 5h, 6h, 7h), a subtitle would be helpful to understand the difference between the upper and lower graphs without reading the legend
3. Similarly, it would be great to add an inset next to the brain maps stating the proportion of significant regions. It is difficult to assess only with the color map particularly when faint. It could also be more precise in the text (example: page 27 “a wealth of areas in the brain”).
4. The two last paragraphs of the results section could be moved to the discussion.

Version 1:

Reviewer comments:

Referee #1

(Remarks to the Author)

I have reviewed the revised manuscript and the authors' responses to my previous comments. The authors have carefully considered my feedback and made substantial improvements to the manuscript. They have addressed all the concerns raised in the initial review, and provided detailed clarifications and justifications for the raised points. They included a much larger dataset from about 700 probe insertions and improved the presentation of key data and analyses. I appreciate the effort the authors have put into these revisions. I have no further comments, and I believe the manuscript is now significantly improved.

Riccardo Beltramo

Referee #2

(Remarks to the Author)

The authors have performed novel analyses to reinforce their arguments and present novel findings that were only possible with the unique dataset. Overall, this revision has substantially improved the manuscript. The team effort to collect an extensive dataset spanning many brain regions is significant and offers a valuable resource for the neuroscience community to build upon and explore future research directions.

Referee #3

(Remarks to the Author)

In this revised version of the manuscript entitled "A Brain-Wide Map of Neural Activity during Complex Behavior", the authors have thoroughly addressed all of the reviewer's comments, leading to a significant improvement in the manuscript's clarity. The revised version clearly articulates the key take-home messages. Removing the analysis of prior has further enhanced the manuscript's overall focus and coherence.

The reviewer would like to highlight the paradigm-shifting nature of this work, which serves as an exemplary contribution to systems neuroscience. Furthermore, the authors' initiative to make this dataset accessible and easily searchable provides a valuable service to the scientific community. The article successfully delivers a detailed and unbiased depiction of the representation of stimulus, choice, feedback, and movement in a visuomotor decision-making task, revealing the sparser localization of stimulus-related signals compared to those related to reward, choice, and movement.

Minor comments/typos:

- One suggestion for further enhancement is to include a dynamic representation of the population trajectory on the Swanson map or a 3D render of the brain for each studied variable. For instance, combining Fig. 1c and Fig. 1d into a movie could give readers a more intuitive understanding of both the localization and dynamics of visual stimulus encoding.
- Line 102 – Reference missing.
- Line 286 – Parenthesis missing.
- Line 285 – Adding an introductory sentence to this paragraph would improve readability.

Dear Editor,

Thank you very much for your message and the reviews. We are very grateful for your and their helpful comments. We present our responses below (comments in blue; reply in black; excerpts from changes in the manuscript in green; figures that exist in this rebuttal only are marked as Figure R(n)).

All the reviewers appreciated the unique contribution of the dataset and the attention we have paid to reproducibility and statistical rigour. The major changes that they requested, and we have duly made, include removing all analyses of the block prior in this paper (leaving it to the more sophisticated and detailed analyses that are possible in the companion paper, which we are resubmitting at the same time), and adding quite a number of new supplementary figures, including a Granger causality analysis of simultaneously recorded areas. Perhaps the most major difference is that we now report on the activity of around three times the original number of neurons, and 50% more brain areas. This is as a result of data from a total of 24 extra mice and 152 extra probe insertions, along with significant improvements in our spike sorting.

We have substantially rewritten the manuscript in response to the reviewers' comments. We have therefore not highlighted every alteration in the text. However, the response is, hopefully, comprehensive.

We hope that you and the reviewers will be satisfied with the changes, and look forward to your further thoughts.

Best wishes,

Before we address individual points of the reviewers, we want to illustrate the main source of changes due to nearly double the amount of data as a result of the new data and the improved spike sorting. This plot shows the number of extra neurons on which we are now reporting on in detail, by area. The revised total count is 75708, of which 62857 are included in all the main figures (the reduction coming from the intersection of constraints associated with each analysis type). Those figures now cover 201 areas of the brain.

Reviewer 1

This study offers an initial assessment of a brain-wide activity map based on neural recordings gathered from multiple laboratories in mice engaged in a decision-making task encompassing sensory, motor, and cognitive elements. The neurophysiological data are accompanied by a systematic assessment of behavioural readouts, including the mouse's movement (speed and velocity) on the running wheel, pupil size, and whisker activity. The study analyses the neural responses of over 30 thousand sorted single units in the context of a solid and well-established behavioural paradigm (e.g., Steinmetz, Nature 2019) that includes cognitive, sensory and motor aspects. The manuscript describes an impressive dataset that the IBL has publicly released, explaining recording and analytical strategies. It focuses on the neural representations of the visual stimulus, behavioural choice, feedback, choice/stimulus expectation and wheel movement. Except for object prior expectations (- but see the comments below), the neural correlates of these variables were widespread in the brain.

The study is particularly significant because it represents much more than a simple manuscript, but rather the result of a network of labs that pooled and analyzed hundreds of experiments, producing a publicly available dataset highly valuable to the global community. This effort resulted in an outstanding brain-wide neural activity map. The dataset is highly relevant because, when fully shared and detailed, it will have the potential to inspire hypothesis-based experiments targeting specific subsets of the brain regions analyzed in this manuscript.

The data, methods, and statistics are solid and well-presented. The manuscript highlights the widespread brain representations of visual stimuli, motor actions, choices, and reward delivery/consumption, supporting globally distributed information coding. This study capitalizes on a large body of work previously done by the IBL team that has carefully established reproducible and standardized measurements of behavioural and electrophysiological recordings.

Thank you for your generous assessment of our paper and for your comments. We have addressed them all.

The abstract, introduction and conclusions are well-written and clearly explained. However, some comments below require attention in order to increase the clarity and interpretability of the main findings and datasets.

Figure 1 and its legend are missing. Please include them in the manuscript.

We apologise: this figure disappeared in the conversion process implemented by *Nature*. We will try to ensure that it does appear in the revision.

The recordings described in this study were collected by 11 labs across different countries and continents. The IBL has done significant work, over the past years, ensuring the reproducibility of its results across research facilities (e.g., IBL, Elife, 2021). However, for this specific study, it would be informative to indicate the actual learning rates and performances, across laboratories, for the 115 mice included in this manuscript.

That's a good idea. We have added a supplementary figure (S25) to document this process. Before the recordings, we train the mice in two key stages: first, they learn the task and get to experience all the contrasts, but stimuli appear with equal probability on left vs. right (non-biased training sessions); then, they experience the biased blocks (biased sessions), and then there are recording sessions. The new figure (S25) shows a scatter plot of the number of sessions it took the mice to complete the two training stages, and also the (lack) of correlation between these numbers of training sessions and the subsequent performance of the mice during the recording sessions. Coloring the animals by lab shows that there is no systematic lab bias.

Figure S25. Training and performance statistics for each mouse, colored by lab, and their correlations across animals. **Left:** Scatter-plot of the number of biased versus unbiased training sessions of each animal. There is no significant correlation between these stages of training. **Right:** Scatter-plot of the number of biased training sessions of each mouse, against the overall percentage of correct trials across all recording sessions of that animal. There is again no significant correlation. The correlation between the number of pre-bias sessions and performance during recording is $r = -0.142$, $p = 0.094$ (not shown).

Please note that the number of mice and recording sessions is now higher. We added recordings from 24 mice so that we now have a total of 139 mice in 459 sessions (and 699 Neuropixels insertions).

The behavioural task described in this study is, at its core, a visual task, and a high number of recordings were acquired from visual areas. For the neurons recorded from these visual areas, were any visual receptive field mapping experiments performed? Were the receptive field properties of each neuron considered in any of the presented analyses? Besides the pupil size, were the X-Y eye positions taken into account?

Thank you for this suggestion. We performed receptive field (RF) mapping at the end of most of the recording sessions (504/699 insertions). In the revised manuscript, we report on our analyses of this dataset, including RFs of neurons across the brain, including areas traditionally designated as visual and non-visual (204 regions): L274:

'At the end of the decision-making task, we performed receptive field (RF) mapping for most of the recording sessions (504/699 insertions). We thereby computed the visual RFs of neurons across the brain (total number of brain regions covered: 204 regions), including classical visual areas and beyond. We estimated the significance of the receptive field of each neuron by fitting the receptive field to a 2d-gaussian function, and comparing the variance explained to the fitting of 200 random shuffles of each RF. Overall, we found a relatively small fraction of cells have significant receptive fields (Fig. S13). Those regions with large fractions tend to be classical visual regions (VISp, VISl, SCs). We also observed non-zero fractions in diverse areas beyond classical visual regions, including auditory cortex (AUDv), auditory thalamus (MG), parts of midbrain (MRN, SCm, APN, NOT), and hindbrain (ANcr1) (Fig. S13 and S14), which further supports the results of neural analysis on coding of visual stimuli during the task.'

We summarised this result in the main text (representation of visual stimulus, L274) and Supplementary Figures S13, S14:

"Taken together, the decoding, single-cell statistics, and manifold analyses reveal a largely consistent picture of visual responsiveness that includes large and short-latency responses in classical areas but also extends to diverse other regions". (L292)

We did not take eye position into account in these RF measurements as mice typically move their eyes little during passive stimulus viewing, and the fact that we measured examples of tightly-localized spatial receptive fields in areas such as the example in LGd (Fig S13d). However, in the analysis of neural correlates with behavioural movement, we did include X-Y eye position as the indicator of "pupil" variable and studied the neural correlates of eye position (Fig. S23).

Figure S13: Receptive field mapping of single-cell across the brain a) Examples of the sequence of visual stimulus for receptive field mapping. Frame rate is 60 Hz. White/grey/dark pixels indicate white/grey/dark stimulus, respectively. (d.v.a. stands for degrees of visual angle) b,c) Fraction of neurons with significant receptive field in each brain region. In panel c), the hollow bars indicate regions containing fewer than 3 neurons with significant RFs, while the filled bars indicate regions containing at least 3 neurons with significant RFs. d) PSTH during the task, the shape of the receptive fields, and the peak response of the receptive fields aligned to stimulus onset for example single cells with a significant receptive field. The peak response of the receptive fields is defined as the PSTH of the pixel in the receptive field with a maximal average spike rate.

Fig. S14: Example of significant receptive fields in auditory areas, hindbrain, and midbrain. a) Example of receptive fields in auditory cortex (AUDv) and auditory thalamus (MG). (d.v.a. stands for degrees of visual angle) b) Example of receptive fields in hindbrain. c) Example of receptive fields in midbrain.

In Figure 5 (Representation of the Visual Stimulus), Panel H compares peri-event time histograms (PETHs) of spiking activity for left and right stimuli for an example neuron. Why are the PETHs from the top panel different from the ones on the bottom panel? (The difference is evident in the red standard errors of the means.)

Apologies. This was a problem in the composition of the figure. It has been corrected.

Figure 5h) Upper panel: Comparison of peri-event time histogram (PETH) of spiking activity for left and right stimuli for the example neuron in panel g conditioned on stimulus onset, along with associated predictions of the full encoding model. The width of the PETH traces reflects standard error of the mean. Lower panel: The same PETHs, but with predictions produced by an encoding model in which the stimulus onset-aligned regressors were omitted. Error bars represent 1 SEM about the mean rate at each time point.

Different animals, across different recording sessions, will inevitably have their own unique behavioural strategies and neural responses. From the results/methods of this manuscript, it appears that most recordings were done with multiple simultaneous probe insertions. These data are particularly insightful for future studies because they can truly provide information on how simultaneously recorded brain regions responded during the task on a trial-by-trial basis. It would be helpful to indicate whether the simultaneous recordings are clearly identified on the publicly released data set. It would be useful to show a table illustrating the number and identity of areas simultaneously recorded in at least 2 or 3 mice. So that future studies (or even this one, if feasible, on a small sample of variables) could restrict the analysis of the neural correlates of the main task/variable events to only simultaneously recorded areas (and, for example, verify if they confirm the whole population data)

That's a very good idea. We have improved the published dataset by adding information about which neurons were recorded on the same probe and the same session (there were typically two probes per session). We have also created a new online table that lists the paired recordings (with pointers to the original data):

<https://github.com/int-brain-lab/paper-brain-wide-map/blob/develop/brainwidemap/meta/granger.csv>

In addition, we now analyze aspects of the paired recordings, looking at Granger causality as a way of assessing functional connectivity (figure S8 [which is too large to include here]). We find high Granger interaction scores for region pairs from all major brain regions, weakly correlated

with anatomical distance, and most such interactions are equally strong in both directions. Our companion paper on the block prior includes further analyses of Granger scores for region pairs computed based on decoding the block variable.

One of the conclusions of the abstract is: "Representations of objective prior expectations were weaker, found in sparse sets of neurons from restricted regions." This conclusion is in contrast with the main finding of a related paper released by the same group on bioRxiv (Findling, IBL, "Brain-wide representations of prior information in mouse decision-making": e.g. "This widespread representation of the prior is consistent with a neural model of Bayesian inference involving loops between areas, as opposed to a model in which the prior is incorporated only in decision making areas"). While the reasons for this discrepancy are well explained in both the results and discussion sections of this manuscript, the abstract conclusions could be misleading, and the sentence reported above should be either rephrased or explained in more detail for consistency.

We appreciate this comment, as it is an issue that we had discussed extensively ourselves. We ultimately decided that the best way to proceed is, as suggested by reviewer 3, to remove the analysis of the objective prior from the current paper, and leave it to the more extensive and detailed presentation that is possible in the companion paper - where, for instance, the relationship between objective and subjective priors can be described and justified in a detail that is not viable for the current paper. That paper also relates electrophysiological results about the representation of the prior to results from wide field imaging - something that we could only collect for dorsal cortical regions.

We have therefore removed what was figure 8 in the previous paper, and have adjusted the abstract, results and discussion accordingly.

In the Methods, "Neurons and Brains Region" paragraph: "amplitude >50 μ V ; noise cut-off < 20 μ V; refractory period violation." How long is the refractory period time window (in ms), and, for a "good unit", what is the percentage of accepted refractory period violations over the total number of spikes?

Given the complexity of the project, we had created a series of white papers describing the various technical components, including:

- Spike sorting pipeline for the International Brain Laboratory (<https://doi.org/10.6084/m9.figshare.19705522.v4>).
- Video hardware and software for the International Brain Laboratory (<https://doi.org/10.6084/m9.figshare.19694452.v1>)
- Data release - Brainwide map (<https://doi.org/10.6084/m9.figshare.21400815.v7>)

The first of these contains an explanation of how we dealt with the refractory period:

"We developed a metric which estimates whether a neuron is contaminated by refractory period violations (indicating potential overmerge problems in the clustering step) without assuming the length of the refractory period. For each of many possible refractory period lengths (ranging from 0.5 ms to 10 ms, in 0.25 ms bins), we compute the number of spikes (refractory period

violations) that would correspond to some maximum acceptable amount of contamination (chosen as 10%). We then compute the likelihood of observing fewer than this number of spikes in that refractory period under the assumption of Poisson spiking. For a neuron to pass this metric, this likelihood, or the confidence that our neuron is less than 10% contaminated, must be larger than 90% for any one of the possible refractory period lengths.”

Over our recorded units, 41.6% of units have a 90% confidence in having less than 10% contaminated spikes.

In Figure 4, panel C, should the colour code (orange/amber) of the movement onset in the graph and the temporal kernel functions be the same?

Apologies. These were just meant to be random colours to distinguish the kernel components. We have fixed it (it is now figure 4e)

Figure 4e) The encoding model uses multiple linear regression of task- and behaviourally-defined temporal kernels on the activity.

Reviewer 2

In this pair of companion studies, the authors (members of an international consortium for experimental neuroscience called the International Brain Lab; IBL) present an unprecedented dataset, collected across 11 labs, comprised of 547 high-density neural recordings sampled across the entire mouse brain during a standardized decision-making task. In one study, the dataset (curated and now made publicly available) and methodology are described, and then used to perform a battery of correlational analyses across the brain to identify encoding of simple task variables during decision-making (including sensory, choice, outcome, movement, and prior information). Based on this investigation and application of rigorous statistical methods, the authors conclude that different task variables are encoded in different ways (some are represented more widely across the brain, while others are more localized), confirming the results of countless previous reports. In the second study, the authors investigate more deeply the encoding of prior information using a more refined set of analytical tools and combining their electrophysiology dataset with another large calcium imaging dataset collected by IBL participants. The main conclusion of this second study is that, in contrast to the initial conclusion of the first study, which suggested that prior information is encoded in a restricted set of brain regions, subjective priors are actually encoded more widely across cortical, subcortical, and midbrain regions. Further, this more widespread subjective prior is partially embodied in the animal's behavior and is driven by action history (as opposed to sensory and/or outcome history exclusively).

The dataset collected here, and particularly the effort to generate a dataset of such comprehensive scale with rigorous control and standardization of task conditions, preprocessing parameters, etc. is obviously noteworthy and is inarguably a highly unique and important contribution to the field. The presented impetus for this enormous collaborative effort is that our capacity to really gain insight into how the brain processes sensory information, computes decisions, and generates behavioral outputs (i.e. how the brain works) is substantially hindered by the limitations of individual lab groups to survey brain structures comprehensively under consistent conditions, and the idiosyncrasies with which different groups design and implement their experiments that limit our capacity to draw accurate comparisons and generalizations. The assumption underlying and framing the effort is that, with a big enough dataset, collected with careful methodological standardization, we should be able to mine the data to gain novel insight into how the brain works. It is, in some sense, the extreme terminus of "data-driven" approaches.

Somewhat unfortunately, the studies, even taken together, fall woefully short of this laudable goal. In the first study, there are in many cases no explicit hypotheses given to ground the questions the authors ask of the data, or if there are, they are so vague as to be almost useless (e.g. visual information may be encoded to some degree beyond classically defined visual areas). We are presented with what amounts to a brain-wide "screen" of task variable neural correlates, and learn, in the end, essentially nothing new about how the brain works – visual stimuli are predominantly represented in visual cortex and thalamus; movement correlates are widespread across the brain, as are responses to salient rewards and punishment. And, at least in this particular case, this is doubly a problem because the specific framing of the knowledge

gap the work is intended to fill is itself defeated – the need for this type of approach totally obviated – by the apparent lack of novel insight. If recording from 300,000 neurons sampled from the entire mouse brain during a well-designed decision-making task does not yield any novel insight, or change our understanding of the brain in any substantial way, then what is the value of such an approach at all?

In the second companion paper, the authors go a bit further. They are able to draw some interesting, useful conclusions from the work that might have been difficult (but this is not totally clear) to fully capture with more limited, fragmented efforts. In particular, they show that while “objective” prior information is only rarely encoded in restricted brain regions (in the first paper), “subjective” prior information is, in fact, much more robustly encoded, and across several levels of brain structure – and this prior takes the form of a partially-embodied action prior, as opposed to other formulations. In the context of value-based decision, this is indeed an important finding, but in the specific context of these two studies, it is still limited in scope for at least two reasons: First, it somewhat contradicts the conclusions of their first study – the authors make a big deal in study 1 about how the restricted nature of prior encoding is interesting because it stands in contrast to how the other task variables are encoded, but then proceed to explain this difference by arguing that this is probably just because animals do not have access to an objective prior, and in fact, encode subjective priors quite robustly. If this is the case, why not just present the analysis of subjective priors alongside other task variables in the first study? Second, it is not clear to what extent the positive conclusions they are able to draw about subjective priors really depend on the nature of the dataset they have compiled. Could this have not been discovered with much more limited methodology?

On balance, while this work is definitely noteworthy even just by virtue of the unprecedented scale of the accompanying dataset and its public availability, in order to merit publication in Nature, we would need to be shown more convincingly that such an approach has the potential to expand our understanding of how the brain works. Because as it stands right now, the biggest conclusion that one walks away from reading these papers is, ironically, that generating and then investigating enormous datasets with correlative analytics and without grounding hypotheses is a fairly that is likely not worth the effort.

Thank you for your careful reading and thoughtful critique of the two papers.

The design of the whole project was to create a single task with just sufficient complexity in as many respects as possible, to do as complete recordings in a fair and unbiased manner as practically possible, to share quality-controlled data openly, and to present both a broad overall survey of methods and results and detailed, deep, targeted, hypothesis-driven analyses. The two papers are paradigm examples of these two different sorts of communication, and we suggest that they are both valuable.

In particular, the current Brainwide Map paper is the bedrock of the whole project - it would not be possible to test hypotheses at scale without it and the data that it is reporting. In order to do due diligence to the presentation of these data, we had to favour breadth over depth. We agree that the results described in the Brainwide Map paper are primarily a screen for regions containing neurons that correlate with task features. However we would push back on the idea

that from this information we learn essentially nothing about how the brain works. Specifically, while our results in forebrain areas are indeed consistent with previous work, our work suggests that specific subcortical regions whose role in such tasks had not been previously appreciated, such as the NOT (nucleus of the optic tract), APN (anterior pretectal nucleus), GRN (gigantocellular reticular nucleus), CA (Ammon's horn in hippocampus), and PRNr (pontine reticular nucleus) are involved in many aspects of this quintessentially cognitive task. It is particularly surprising that hindbrain regions such as GRN - which would traditionally be thought to be involved only in low-level motor control - contain visual sensory information just shortly after the classical visual areas, even once other task variables are controlled for (Fig. 5b,f), as well as some of the lowest choice latencies and strongest population-level choice representations.

Our data therefore suggest that these regions may play a key role in all parts of the sensory-motor decision-making processes, a result that we would not have found if we had stuck to recording from the “usual suspect” areas such as cortex and basal ganglia. The fact that we have identified which specific hindbrain regions contain task correlates will allow future recordings to target only these regions, but this would not have been possible had we not performed the systematic scan. We also hope that this finding might help alleviate the bias in the literature that favours recording from the cortex in perceptual decision-making tasks (with other areas such as the basal ganglia being studied less, and most subcortical regions barely studied at all). We discuss these issues in the text (lines 309-317, 475-485)

To make the separation even clearer, and to address the point you and other reviewers raised about the apparent contradiction between the Brainwide Map and Prior papers, we have now removed the analysis of the objective prior from the Brainwide Map paper, and leave it to the more extensive and detailed presentation that is possible in the companion paper - where, for instance, the relationship between objective and subjective priors can be described and justified in a detail that is not viable for the current paper. We have therefore removed what was figure 8 in the previous paper, and have adjusted the abstract, results and discussion accordingly.

Of course, you are quite right that there are very many more depth-based analyses that can be accomplished with this dataset. Teams within IBL are engaged in many of those analyses (as partly evidenced by posters and preprints); and our intent in making the data widely available (in fact, even before submission of the current papers), and teaching courses on how to get access to and use the data (including via Neuromatch) is that people outside IBL will use the data to do new and exciting research of which we haven't even dreamt.

Fortunately, I think this is not actually the case, and there are probably countless ways that the authors could spend a bit more time with this dataset to make some genuine, likely impactful discoveries. Without going deep into specifics, I will provide a few ideas for avenues that the authors could pursue to achieve this (avenues which are likely already being explored).

Thank you very much for these suggestions. Many of them are indeed under active investigation within IBL. In the revised version of the current manuscript, we now present the ones that fit under the rubric of the broad coverage of the paper.

1) The dataset contains neurophysiological and behavioral measures from 115 mice performing the exact same, standardized task. Presumably then, there is some degree of individual variability in task performance (objective as well as idiosyncratic variability in task strategy). This seems like a perfect opportunity to investigate, even in a coarse manner, how such differences in behavioral variability may be related to underlying differences in neural dynamics and mechanisms. For example, are there specific brain regions, or clusters of regions that, beyond encoding task variables to whatever degree, are highly predictive of task performance, or are correlated with the extent to which animals use a particular cognitive strategy (such as how much they depend on estimates of priors?).

This is an interesting suggestion. Indeed, one of the early publications from IBL was Ashwood et al. (*Nature Neurosci*, 2022) which provides a method for discovering latent structure in behavioural time-series, using the first 90 trials of each session (before the biased blocks begin); and a recent preprint (Bruijns et al, *bioRxiv*, 2023) extends this to look across the initial acquisition of good behaviour. As we now point out in Discussion, however, we are somewhat victims of our own success in our rigorous training protocol - the difference in the asymptotic behaviour of the mice is surprisingly modest, making it hard to determine the physiological correlates.

In the discussion (L523): “Furthermore, our behavioural training protocol is aimed at reducing individual differences in performance, which impedes a complete analysis of the relationship between neural activity in particular regions and factors such as the reward rate.”

It is also the case that within-session differences are dominated by factors such as satiation/fatigue that pose interpretational and technical difficulties (for an instance of the latter, the effect of Neuropixels drift).

As a simple test of the potential predictability of task performance, we correlated a sliding window of within-session reward rate with overall firing rate in all the regions concerned, finding no systematic correlation, Fig. R1.

Figure R1: The left panel shows a sample insertion with two regions. For each, the scatter plot is firing rate in the inter-trial period versus performance, for chunks of 20 consecutive trials. Pearson’s and Spearman’s correlation are given. The right plot shows these correlations for the whole data set, gray dot per recording when $p_{\text{pearson}} > 0.05$, else black.

2) While restricted because of the geometry of single probe insertions, the authors have access to a wealth of information about simultaneous neural dynamics recorded across different brain regions (at least those that are dorsoventral neighbors). Despite the repeated mention of “loops” and “widespread, distributed activity,” there is astonishingly no effort here to leverage the dataset to understand anything about inter-areal communication during decision-making. It would be nice to get at least some insight into this knowledge gap that the authors allude to, certainly have the capacity to address, but then say nothing substantial about.

Thank you for this excellent suggestion. We have performed several new analyses along these lines. Specifically, in this paper (figure S8, which is too large to copy into this reply) and the companion paper (Prior paper figure S11) we now report functional connectivity analyses (using Granger causality). We find strong interconnectedness across most region pairs and sub-division into 10 “Cosmos” brain regions shows the isocortex to be most broadly connected to all other regions. Correlating Granger scores with structural connectivity (based on axonal tract tracing from a different study) and with Cartesian distance in anatomical space resulted only in moderate scores.

Furthermore, to enable further analyses along these lines the published dataset now contains information about which clusters were recorded on the same probe and the same session (since there are normally two probes). We have also created a new online table

(<https://github.com/int-brain-lab/paper-brain-wide-map/blob/develop/brainwidemap/meta/granger.csv>) which contains an explicit list of the paired recordings (with pointers to the original data).

3) Given the unprecedented nature of the dataset, it would also be interesting to see some quantitative assessment of how useful enormous amounts of neurophysiological data really are. The authors are in a unique position to address this. For example, how strongly do conclusions about task variable coding maps hold as a function of the number of neurons recorded, the number of sessions, mice, etc.? If it turns out to be the case that most of the main conclusions could be drawn with much more restricted coverage, i.e. that there are dramatic diminishing returns on scaling up the sheer quantity of neurophysiological data, then this would seem to be an important conclusion to spell out so that future efforts can be better directed toward more fruitful avenues.

First, we note that our main aim was comprehensive brainwide coverage, which requires multiple recordings from each area to identify which ones correlate with task variables. It would not have been possible to draw these conclusions without brainwide coverage - and so more restricted recordings would not have achieved the IBL intent. (If we find no evidence that a brain region encodes a particular variable that does not mean even in retrospect that we did not need to record that brain region.)

The primary goal of our targeting strategy was to deliver two recordings in as many areas as possible with as few total penetrations, along with a single site targeted in all mice for purposes of reproducibility. The fact that we have very many recordings in some regions is partly a function of the latter site (which is the subject of a different paper: <https://www.biorxiv.org/content/10.1101/2022.05.09.491042v6>), partly due to the large size of some regions making them appear in many recordings; and partly incidental happenstance of the former targeting.

Supplementary figures S9 and S12 show session-specific decoding and single-cell analyses for all the relevant variables to provide a global measure of the variability across sessions. We expect that a good part of the variability comes from heterogeneity within the regions and of the precise location of the Neuropixels probes within those regions. The logic of the population trajectory (formerly called manifold) analysis makes it most suitable to look at the effects of the numbers of neurons recorded - and we now have a new supplementary figure (S5) that reports on this, showing that for both, sampling neurons or sampling sessions, and most regions, the mean of the sampled scores visually matches the scores of the full dataset, showing that the strongest regional differences are also present for subsets of the data, however with more variance.

Fig S5: Maximal Euclidean distances for example regions with random data subsets. Shown are maximal population trajectory distances for the same highlighted regions as in the main figures (5I, 6I, 7I). The distances are computed after 15 subsamplings from half or a quarter of all neurons (top row of panels) or sessions (bottom row of panels). Grey dots indicate control scores for the sampled data, from trial randomization as in the main population trajectory analysis. For both, sampling neurons or sampling sessions, and most regions, the mean of the sampled scores visually matches the scores of the full dataset, showing that the strongest regional differences are also present for subsets of the data, however with more variance.

Finally, there is a potential problem in the task design. In most trials, whether animals are rewarded or not is determined by the correctness of the choice based on the sensory cue. However, in the 0% contrast trials, where there is no “correct” choice, the probabilities of reward on the left and right sides were set to 20% or 80% according to the prior probability of left and right stimuli. This means that, in principle, mice can use these differential reward probabilities in 0% contrast trials to bias their choice. In other words, the bias in reward probability during 0% contrast trials, rather than the prior probabilities of left and right trials, could have contributed to or been the main cause of the choice bias. It is important to exclude this possibility to support the overall conclusions.

Thank you for pointing out this possible confound. This is addressed in the response from the Prior paper where it is shown that the subjects are not just using the 0% contrast trials to infer the prevailing block. Note that analysis of the block prior has been removed from the Brainwide Map paper.

Overall, my sense is that this work could ultimately be impactful and unique enough to merit publication in Nature, but definitely not in its current form. The authors must do a better job of

convincing us that this approach is worthwhile in terms of real intellectual and scientific return – the dataset alone, while obviously impressive (and exciting), is not enough.

We hope that we have been able to convince you that the results in the present paper make it a significant contribution in and of itself.

Reviewer 3

A. Key results:

The paper “A Brain-Wide Map of Neural Activity during Complex Behaviour” presents a brain-wide analysis of neural activity during a standardized visuomotor decision-making task. By pooling large-scale electrophysiological data acquired by multiple laboratories and ensuring high reproducibility, the activity of a large number of neurons covering most brain regions was acquired, allowing analysis of the decision-making task at an unprecedented coverage and at high temporal resolution. The authors used strict quality control metrics and standardized analysis pipelines to uncover the brain regions that contain neural activity linked to different aspects of the decision-making process, such as the visual stimulus, the choice, the feedback, the objective prior, and the wheel movement. The data shows widespread engagement of brain regions by the wheel movement and the feedback (reward and noise/time-out), a large but smaller set of regions driven by choice, and a sparse set of regions encoding the visual stimulus or block prior. An important finding is that visual stimuli appear encoded first in classical visual areas and later in a collection of mid and hindbrain regions that also strongly encode choices. Also, interestingly, choice encoding was more substantial in these subcortical regions than in classically described cortical regions. As previously reported, movement (uninstructed or instructed, in that case, wheel movement) seems to be encoded throughout the brain. One should interpret the regions encoding feedback cautiously because the task design does not allow decorrelating reward from actions linked to reward. Similarly, results on the block prior are unclear as the authors used the objective prior unknown to the mouse; a companion paper addressed this issue. Beyond the results, it should be noted that this paper presents a new, unprecedented resource: a large publicly available dataset easily searchable and useable by other neuroscientists.

B. Originality and significance: This paper by the International Brain Laboratory (IBL) represents a unique collaborative effort, the first of this type and scale in systems neuroscience, at least to the reviewers’ knowledge. Therefore, this paper is one of a kind and hopefully the start of a paradigm shift in systems neuroscience. The high significance of such effort comes from the fact that neural correlates of behavior are distributed throughout the brain. Small-scale studies focusing on a predefined set of brain regions have taught us a lot and have value, but they do not capture the big picture of where different variables are encoded in the brain. The only way to answer this question is through large-scale, reproducible studies like the one presented in this paper. The dataset is of high quality and provides unprecedented spatial coverage of single-neuron activity at high temporal resolution. Moreover, the provided data and the technical information accompanying it are easily accessible, searchable, and useable for other researchers, meaning that more discoveries linked to this data may come. Of course, the

present results are not necessarily generalizable and are specific to one task. Nonetheless, these results provide clear answers about the regions involved in such a task. For example, this unbiased screen demonstrates that the representation of visual stimulus and choice is prominent in mid- and hindbrain structures for this type of decision-making task, which has been overlooked in the literature – often “cortico-centric”. This result paves the way for further studies to investigate whether it holds for more complex tasks and how the encoding arises through learning. In summary, this paper is highly original and significant and will significantly impact neuroscience.

C. Data & Methodology: The manuscript is well-written and well-structured. The methods are detailed and provide all the information for successful reproduction. The data is publicly available, and an intuitive API allows easy data browsing.

D. Appropriate use of statistics and treatment of uncertainties: The authors use appropriate statistical methods and carefully discuss potential uncertainties. The authors took great care to choose appropriate null distributions whenever possible. Statistics are of the utmost importance for this paper, which pools data from different sessions, mice, and laboratories. Method sections give ample details, which is highly appreciated.

E. Conclusions: robustness, validity, reliability. As the data was collected in multiple laboratories (each recording location was performed in at least two different laboratories) and the reproducibility was carefully investigated in another publication (IBL 2022), the authors ensured very high robustness and reliability of the described findings. The overall conclusion that neuronal activity during many aspects of the task is widespread throughout the brain is well supported.

F. Suggested improvements: Overall, we do not see significant flaws in the manuscript, but we would like to suggest improvements. We understand the difficulty of analyzing brain-wide data and the beauty in using the same analysis pipeline throughout the paper to present comparable brain-wide maps across the different aspects of the task (vision, feedback, movement, etc.). However, in some places, it seems that the chosen analysis is better suited for certain time periods and variables (i.e. choice) than for others (i.e. visual stimulation or prior encoding) - see points below - and that additional analysis could help extract additional information from this valuable dataset.

G/H References, Clarity and context: lucidity of abstract/summary, appropriateness of abstract, introduction and conclusions : We found the paper very factual, not overselling, and the presentation of the context appropriate.

Thank you for these very accurate summaries. You have captured the intent of the project as a whole and this particular paper perfectly.

Specific comments:

1. Major point: We find the section on the representation of prior more confusing than helpful. As the authors point out, the variable studied here - the true block prior – is not accessible to the mouse. The results show very few regions encoding that variable. The companion paper

addresses that issue and finds that the subjective block prior is actually encoded in a large number of regions. Why keeping that section? Either readers have not read the other paper and will be confused, thinking that the prior is sparsely encoded, or they have read the other paper and will think that this part is conceptually wrong. We suggest to remove it and point towards the other paper in the discussion instead.

Thank you very much for this suggestion - it was also something that we had discussed extensively ourselves. We have followed your advice to remove the analysis of the objective prior from the current paper, and leave it to the more extensive and detailed presentation that is possible in the companion paper. We have therefore removed what was figure 8 in the previous paper, and have adjusted the abstract, results and discussion accordingly.

2. Comments on the representation of the visual stimulus:

a. How do you interpret the large difference between the single-cell analysis and the decoding/manifold analyses for that variable? This is puzzling compared to other variables. What do the authors conclude about these midbrain/hindbrain regions when considering all analyses? Do they encode the stimulus or not? A clear statement – even if it is a speculation given the evidence provided – would be useful. Generally, the conclusion of this part is vague, uninformative for the reader and could be improved.

We apologise for being unclear: we do find visual responses in midbrain/hindbrain, even when controlling for other variables such as choice. With the extra recordings and the improved spike-sorting, we now have more neurons in these regions. As a result, the single-cell analysis identified 8 significant regions in midbrain/hindbrain, which is more consistent with the results of decoding/trajjectory (formerly manifold) analyses (See the section in the main manuscript on the representation of visual stimuli L280). Therefore, we suggest that our conclusions about the representation of visual stimuli are coherent. We hope that this is now clearer in the main text of the manuscript.

To strengthen these conclusions further, we performed receptive field (RF) mapping at the end of most of the recording sessions (504/699 insertions). In the revised manuscript, we report on our analyses of this dataset, including RFs of neurons across the brain, including areas traditionally designated as visual and non-visual (204 regions). We summarised this result in the main text (representation of visual stimulus, L274) and Supplementary Figure S13, S14. Specifically, we first computed the RF of each neuron in the standard way. Then, as a first-pass estimate of the significance of this description of a neuron's activity, we compared the variance explained by a 2d-Gaussian function fit to this RF to the variance explained by 2d-Gaussian function fits to 200 permutations of the same RFs. This generated an empirical p-value. Overall, we found a relatively small fraction of cells had significant receptive fields in this sense (Fig. S13). These regions with large fractions tend to be classical visual regions (VISp, VISl, SCs). However, we also observed non-zero fractions in diverse regions beyond classical visual regions, include auditory cortex (AUDv), auditory thalamus (MG), parts of midbrain (MRN, SCm, APN) and hindbrain (ANcr1) (Fig. S13, S14). This further supports the conclusion of the neural analysis of the coding of visual stimuli during the task:

Figure S13: Receptive field mapping of single-cell across the brain a) Examples of the sequence of visual stimulus for receptive field mapping. Frame rate is 60 Hz. White/grey/dark pixels indicate white/grey/dark stimulus, respectively. (d.v.a. stands for degrees of visual angle) b,c) Fraction of neurons with significant receptive field in each brain region. In panel c), the hollow bars indicate regions containing fewer than 3 neurons with significant RFs, while the filled bars indicate regions containing at least 3 neurons with significant RFs. d) PSTH during the task, the shape of the receptive fields, and the peak response of the receptive fields aligned to stimulus onset for example single cells with a

significant receptive field. The peak response of the receptive fields is defined as the PSTH of the pixel in the receptive field with a maximal average spike rate.

Fig. S14: Example of significant receptive fields in auditory areas, hindbrain, and midbrain. a) Example of receptive fields in auditory cortex (AUDv) and auditory thalamus (MG). (d.v.a. stands for degrees of visual angle) b) Example of receptive fields in hindbrain. c) Example of receptive fields in midbrain.

We conclude from this that: “Taken together, the decoding, single-cell statistics, and population trajectory analyses reveal a largely consistent picture of visual responsiveness that includes large and short-latency responses in classical areas but also extends to diverse other regions, even when controlling for correlated variables, particularly at later times relative to stimulus onset” (L292)

Nevertheless, it is certainly the case that the results from the different analysis methods are not identical. This is because the different analyses are reporting on very different aspects of the same data, making different assumptions and licensing different statistical conclusions - and so we did not expect them to agree completely. For example, when analyzing visual responses, the population trajectory analysis - which includes all cells simultaneously from all recordings - is expected to have higher statistical power than the single-cell analysis, even though both account for covariates such as choice. Accordingly, it finds a larger number of significant regions. The paper now says:

It is important to note that we do not expect the different analysis methods to agree perfectly, since they focus on different aspects of the responsivity of individual neurons and populations thereof, and even, in the case of the population trajectory analysis, combining information across multiple sessions, rather than within single sessions (which also allowed us to use this

method to test the robustness of our results by comparing findings based on subsets of the data; Fig. S5). The methods thus should be interpreted collectively. For a direct comparison of analysis scores, see flatmaps in Fig. S6 and scatter plots of scores for analysis pairs in Fig. S7. (L199)

b. The latency information extracted from the manifold analysis could be supplemented with more classical analysis of the latency directly from the PSTH.

Agreed. We had shown PSTHs across all regions in Figure S2 of the original submission. We have now created a new Figure R2 which reports latencies extracted from the PSTHs shown in R4, defined as the time until the z-scored PETH has reached 70% of its maximum. In order to add to an answer about activity after motion onset, we picked the windows of interest here all to start at the respective alignment event (stim on, first motion, feedback) and last 150 ms. This confirms the analyses of the main text manifold analysis.

Fig R2) Top: PETH-based latency for all regions and three alignment events. For each alignment event (stimulus onset, first movement onset and feedback time) the latency is shown per region, computed as the time in the window of interest (150 ms after stimulus, 150 ms after first movement, 150 ms after feedback time) when 70% of the maximum of the z-scored PETH in this window was reached. Compare with PETHs shown in fig R4. Visual areas are among the earliest peaking regions after stimulus onset; motor areas such as MOp continue ramping after motion onset (and thus late latency), while most areas for choice are falling in activity and thus peak at time 0, and most areas are immediately peaking after feedback time. These results are mainly in line with the manifold-based latencies in the main figures (5-7). Bottom: Zoom into one panel to highlight SCm, SCs, IC.

c. What about the latency in the superior layers of the superior colliculus, which also receive direct input from the retina? The authors only comment on visual and thalamic areas. If the latency is also high in SCs, it would contradict the argument made by the authors about the “classical areas” responding fast.

In Figure R2 (shown in full above), the superior and inferior colliculus respond relatively late when considering the PETH-based latency, i.e. taking the mean of the PETHs of all cells in a region, then z-scoring and quantifying the time until 70% of the max of this line was reached, using 150 ms long windows starting at stimulus onset.

We think the reason we found high latency in SCs is that we did not observe significant visual responses there during the task. We do see passive visual responses in SCs during receptive field mapping, even when they are not observed in the main task (Figure S13). This is due to the fact that, by chance, the receptive field locations of the SCs neurons we recorded did not overlap the stimulus positions used in the task. By analysing receptive field mapping data, we found 13 of 81 (16%) SCs neurons have significant RF fields. Here we show all the significant SCs receptive field locations measured along with the location of the stimuli used in the task in the following plot

Figure R3: Significant receptive field in SCs. The red circle denotes the location of visual stimulus.

d. Could we see some example neurons/PSTH from these hindbrain regions locked on the stimulus?

We have an extensive visualization website where it is possible to explore the data interactively in great detail. For example, a neuron in the midbrain region SCs (superior colliculus sensory related) has these PETHs locked to stimulus onset (besides shown other events):

Select a good cluster from the dropdown menu or by clicking directly on the unit to visualize its properties.

- SCsg — #1336
- SCsg — #908
- SCop — #910
- SCsg — #912

< >

Brain region	Cluster #	Max amplitude	N spikes
Superior colliculus superficial gray layer	912	709.48 uV	19899

This screenshot was taken directly from the website - please scroll down to the bottom of

https://viz.internationalbrainlab.org/app?dset=bwm&pid=f9b2e36b-eef8-402f-b121-786dfed58bc&tid=0&cid=912&qc=0&spikesorting=ss_2024-05-06

Here is further a list of probe ids (“pid”) that recorded neurons in SCs, which can be entered and explored on the website [pid, #neurons in SCs]:

[('530f1670-9412-44ac-afdb-935d46bcaad3', 3), ('97207d87-3fcd-4ebb-b0c7-087bdf8a95c', 2), ('53ecbf4f-e0d8-4fe6-a852-8b934a37a1c2', 8), ('b543e81e-4c8f-415e-82ec-631b177d19d2', 4), ('f083fcd5-e456-433d-8aa5-6ace0a7ac170', 15), ('ba291bec-4492-4d7f-a6aa-483ebb64b3c3', 9), ('07abb39b-063b-4ef8-b3ab-963c6a1b5cc5', 19), ('ca5404f7-297c-40f1-bbf0-5ac0a63e24f8', 13), ('d14f70e6-bf7b-4d6d-a380-bfd0a46ed7a1', 12), ('f9b2e36b-eef8-402f-b121-786dfed58bc', 21), ('43436b4b-0431-407f-b83d-d657ec22b5c6', 7), ('b939cc85-6028-404a-995d-28c8405a07db', 6), ('dac5defe-62b8-4cc0-96cb-b9f923957c7f', 7),

12), ('f0c390da-d8e3-4b5f-8df7-bd2f153ed01b', 11), ('d4291925-ad00-47fe-baaf-3fdff0991e86', 5), ('e033ef96-3590-4e52-bb01-6c6d75bab083', 21)]

Here is a hindbrain example neuron, for example from region GRN:

https://viz.internationalbrainlab.org/app?dset=bwm&pid=f9b2e36b-eef8-402f-b121-786dfed58bc&tid=0&cid=912&qc=0&spikesorting=ss_2024-05-06

Further regions can easily be found in the suggestions in the search window of this webpage:

Examples: `cortexlab` (lab), `CSHL052` (subject), `CA1` (brain region)

Brain wide map	Repeated sites	Original	2024-03-22	2024-05-06
Q 0ece5c6a-7d1e-4365-893d-ac1cc04f1d7b				
angelakilab	NYU-37	2021-01-27	CM, IAM, RE, RH, HY	66f810a8-b18d-4b54-9f28-2964dfceccad
angelakilab	NYU-37	2021-01-30	GRN, DRP, PGRNd, CENT3, chpl	7332e6cf-9847-4aca-b2e3-d864989dd0fb
angelakilab	NYU-37	2021-01-30	ANcr1, IP, arb, icp, DN...	789fd2ee-c755-46c5-9c6f-260092520216
angelakilab	NYU-37	2021-02-01	DR, PAG, SCiw, SCig, SCdg...	b7c57ce1-a75b-410a-94b6-8abee5a92c4f
angelakilab	NYU-37	2021-01-25	CP, SSp-m6a, SSp-ul5, SSp-m5, STR...	c4b5a9fa-10cb-4195-9c17-15b6a1f7719a
angelakilab	NYU-37	2021-01-26	CP, MOp2/3, MOp6a, MOp5, MOpL...	decc8d40-c74-4263-ae9d-a0cc68b47e86

Probe type 3A

3. Comments on the representation of choice:

a. In figure S2, one can clearly see how the average PSTH per brain region for visual stimulation and choice are shifted version of each other. This is not surprising as the mice have fast reaction times, however it would be interesting if the authors can elaborate more on this and potentially provide additional analysis in Figure 6, i.e. how the choice map changes when selecting trials with short versus long reaction times?

Figure S15 compares kernels for the GLMs for stimulus and choice when doing a median split on the first response times. The difference in maps between models fit on the earlier reaction trials, late reaction trials, and all trials show several findings for the choice kernels and stimulus kernels.

0.000 0.005
 ΔR^2 right stimulus onset, early responses

0.000 0.005
 ΔR^2 right stimulus onset, late responses

0.000 0.005
 ΔR^2 right stimulus onset, all responses

0.000 0.002
 ΔR^2 right movement onset, early responses

0.000 0.002
 ΔR^2 right movement onset, late responses

0.000 0.002
 ΔR^2 right movement onset, all responses

Figure S15. Variance explained by stimulus and choice kernels in GLMs fit to early (below median), late (above median), and all RT trials. a) Mean ΔR^2 from right stimulus onset kernel per region in trials with response time below median (left), above median (middle), and all trials (right). b) Mean ΔR^2 from right first wheel movement time kernel per region in trials with response time below median (left), above median (middle), and all trials (right).

Regions with high explained variance from stimulus onset on all trials mostly appear as well in the early-reaction model (e.g. RSPv, VISl, PAG, and RN) while a handful of new cortical regions in motor areas (namely MOs, ORBm and ILA to a lesser extent) seem to only be explained when fitting early trials. Late response trials show fewer regions with well-explained (greater than an 0.03 change) variance, but interestingly shows explained variance in subiculum and post subiculum which does not appear in the set of all trials.

In regions which show a high degree of variance explained by rightward movement onset RSPv again appears when fitting all trials and early trials, but not late trials. In the case of movement onset, however, secondary visual areas VISam and VISal are consistently involved in all trials along with motor areas. Notably the high variance explained in subiculum extends to hippocampal CA1 and post-subiculum only in late response trials, and does not appear at all in early response trials. Subcortical involvement seems limited to early trials in some regions like PAG which do not appear in the model fit to all trials.

b. Similarly, looking at the example unit in GRN as well as the average PSTHs in Figure S2 one wonders how the signal develops beyond the 50ms after movement onset shown. If possible, the authors could extend these time windows allowing a better understanding of the temporal dynamics of the ramping up regions.

We have duly extended the PSTH beyond the first 50ms (see figure R4 below). Of course, activity is then affected by the feedback phase of the task. GRN for example continues ramping for 150 ms after first motion onset (see third column below for choice, GRN being in the middle of pinkish hindbrain regions).

Figure R4: **Additional PETHs**. The window for “choice” is aligned here from 0 to 0.15 s after first motion, i.e. showing some more temporal development. We further show “stim0” which averages responses to stim onset for 0-contrast trials only and “stimdiff” showing the difference of these to the PETHs “stim” which are all trials averaged aligned to stim onset [0, 0.15]. Each PETH was independently z-scored.

Since the duration between movement onset and feedback onset is about 100 ms on average and varies trial by trial, the direct computation of PSTH after 50 ms of movement onset is mixed with neural response to feedback. To extend time-windows and visualise continuous temporal dynamics across different task epochs, we also computed time-warped PETHs, simultaneously aligned to stimulus, movement and feedback onsets. Specifically, we evenly divided the duration between stimulus onset to movement onset into 10 time bins, and also divided the duration between movement onset and feedback onset into 10 time bins for each trial (the length of time bin varies trial by trial). We then computed the spike rate of each time bin, and averaged across trials and sessions. This approach ensures stimulus, movement and feedback onsets are perfectly aligned across trials, therefore enabling us to visualise temporal dynamics for the entire trial duration. We added Supplementary figure S3 in the manuscript, where we plotted the time-warped PETHs across the brain. We found different types of temporal dynamics across regions. For example, GRN displays slow ramping dynamics after movement onset; while RSPv reaches the peak activity around movement onset, and decreases after that.

Figure S3: Time-warped PSTH of average neural activity across the brain. The duration between stimulus onset and first wheel movement time is divided into 10 equal-size time bins, as is the duration between onset to first wheel movement time and feedback onset for each trial (thus the length of time bin varies trial by trial). The spike rate of each time bin is computed and averaged across trials and sessions. This approach ensures stimulus, movement, and feedback onsets are perfectly aligned across trials.

4. Comments on the representation of wheel movement:

a. In figure 9 it does not fully become clear why the authors present the analysis for both speed and velocity. Not surprisingly, the results are extremely similar. The velocity could be moved to supplementary material. Analogous analysis of other DLC outputs from the nose, paw etc. could be more interesting in the main figure

Although speed and velocity look quite similar in aggregate, the sensitivity to velocity is markedly less than to speed, which is why we reported them both (and devoted supplementary figure S22 to it).

We report only selected analyses of other DLC outputs in the present paper since they are not part of the way that the subjects have to report the results of the discrimination and so are not controlled in quite the same way. In figure S10 of the companion paper, we do analyze motor correlates of the block prior (as in figure S3 of the original submission); we show neural correlates of licking in figure S19 [which is too large to include in the rebuttal], and we now

expand figure S23 to separate out the correlations with different motor variables extracted from DLC.

Figure S23. The behavioural correlates of single-neuron activity across the brain. a) Statistical tests to measure the behavioural correlates of single neurons across all sessions. We compute the Pearson correlation coefficient between the time series of neural activity and five behavioural variables (nose position, pupil diameter, paw position, and licks, extracted from behaviour video by using DLC; see Methods). The significance of correlation is estimated by a time-shift test⁷⁸ (Methods), using $FDR_{0.01}$ to correct for multiple comparisons. b) The flat brain map of the fraction of neurons significantly correlates with at least one of the movement variables. c) The flat brain map of the fraction of neurons that significantly correlate with one of the movement variables: nose, pupil, paw, tongue.

b. Why did the authors choose the linear-shift method rather than "imposter sessions" (taking the behavioral data from other trials/sessions) in the case of these behavioral variables? Could the authors spell that out for the reader?

We used the linear-shift method rather than imposter sessions where possible because it allows computing an independent p-value for each session. Because the imposter session method

uses data from multiple experiments it raises the potential for correlations in the significance values found for multiple experiments.

5. Additional comments:

a. Do the authors have any data in neuromodulatory regions (dorsal raphe, locus coeruleus, basal forebrain, ventral tegmental area) and could they analyze this data to evaluate the impact of global neuromodulatory signals across the brain during this task? If not, maybe it would be worth discussing about targeting such regions, as the authors speculate on the role of such signals several times in the paper.

These regions are certainly part of the general targeting:

region acronym, pid, number of neurons in region:

'SNc':	{'18be19f9-6ca5-4fc8-9220-ba43c3e75905':	21,
	'96c816ad-9a48-46a4-8a84-9a73cc153d69':	6}
'DR':	{'b7c57ce1-a75b-410a-94b6-8abee5a92c4f':	29,
	'46cd9c0a-39de-4aeb-90a6-86a2fda0b1a4':	13}.

Although we are working on trying to distinguish different cell types based on our extracellular recordings, we do not yet have confidence that we can do this in a reliable manner, and so are cautious to distinguish neuromodulatory neurons in these nuclei from their GABA/Glutamatergic neighbors. Thus, we are hesitant to draw conclusions about specifically neuromodulatory influences based on these recordings. We now make this clear in the paper, L112:

While information about molecular cell types can sometimes be gleaned from spike waveforms, we did not attempt this for the analyses here; for example, while we recorded from some of the main neuromodulatory regions, we do not make specific claims about which neurons release which neurotransmitter.

It is quite true that neuromodulatory signals are likely to be very important. Within IBL, there is a fibre photometry project aiming to record their activity during the task, but data collection is not yet complete.

b. In the paragraph of the discussion about the brain coverage (page 31), it would be interesting to know what type of brain regions are systematically missed (i.e. nuclei below a certain size? lateral regions?).

Including the latest data that was publicly released on the 15th February 2024, and taking account of our threshold on there being at least 5 neurons in a region, our canonical Brainwide map dataset comprises 210 out of the 308 regions composing our high-level parcellation of the Allen atlas. Note that 201 out of these 210 regions are analysed in the pooled figures, as the manifold criterion of at least 20 neurons per region is not met for these 9 regions: AD, APr, CA2, ECU, FC, IA, IGL, PBG, PH.

Within the 98 regions not in our canonical set, 7 were excluded by design from the Brainwide map targets (namely MOB, AOB, AOBgr, onl, AOBmi, void, root). The remaining

91 regions not covered are indeed small and difficult to target. Their average volume is 0.16 mm³, and their positions in the brain, as the reviewer suggests, are the most lateral ones.

Minor points:

1. Figure 1 and one panel of Figure 5 (j) disappeared from the provided pdf. We had to use the preprint version of the paper to assess these parts.

We apologise that these were missing. They somehow disappeared in the conversion process associated with the submission to *Nature*. We will ensure that it does not happen with the revision.

2. Throughout the paper when the results of the encoding model for a single unit are presented (i.e. 5h, 6h, 7h), a subtitle would be helpful to understand the difference between the upper and lower graphs without reading the legend

We now include a subtitle.

3. Similarly, it would be great to add an inset next to the brain maps stating the proportion of significant regions. It is difficult to assess only with the color map particularly when faint. It could also be more precise in the text (example: page 27 “a wealth of areas in the brain”).

We now include this statistic for all but the GLMs (for which we lack an appropriate significance test). We have attempted to make the language more rigorous.

4. The two last paragraphs of the results section could be moved to the discussion.

Thank you for this suggestion. In order to streamline this rather complex paper, we aimed to have mini-discussions associated with the reporting of each figure so that they are in close apposition to the data on which they are commenting. Otherwise, the main discussion would be very disjointed. We therefore hope that it is acceptable to leave those two paragraphs in their current location.